# Problem Distributions as *Tasks*: Repurposing Meta Learning for Generative Combinatorial Optimization towards Multi-task Pretraining and Adaptation

**Wenzheng Pan**[1]  **Jiale Ma**[1 2]  **Nuoyan Chen**[1]  **Yang Li**[1 2]  **Junchi Yan**[1 2]

## Abstract

Despite the fast progress of Neural Combinatorial Optimization (NCO) on graphs, existing solvers mainly learn a narrow task (e.g., uniform TSP) at a time and hardly handle instances over diverse distributions. This paper proposes M²GenCO, a Multi-task learning framework that pioneers the instantiation of the Meta-learning mechanism with diffusion-based Generative solving for CO Problems (COPs) on graphs, first formulating "tasks" in meta-learning as distinct problem types instead of instances of the same problem. With a tailored lightweight graph neural network, our framework performs effective joint pre-training on a variety of problem types and efficient fine-tuning to adapt for out-of-distribution scenarios. Further, we establish a benchmark comprising 5 classic graph COPs with varying scales and multiple distributions, forming 38 distinct test datasets that facilitate standard evaluation of generalizability and adaptability for NCO solvers. Empirically, M²GenCO with greedy decoder yields an overall 9.16% performance gain with an average 95.6× acceleration for inference, and achieves concrete state-of-the-arts on all test sets with simple local searchers, maintaining superior solving time against previous neural methods. The computational resource and time consumption for training are saved by up to 82% and 91%, respectively. The code is available at our Github repository.

## 1. Introduction

Combinatorial Optimization (CO) (Papadimitriou & Steiglitz, 1998) finds extensive applications over transportation, portfolio optimization, drug recommendation, and more (Veres & Moussa, 2019; Wang et al., 2023; Guan et al., 2024; Paschos, 2014; Wang & Tang, 2021; Singh & Rizwanullah, 2022; Baty et al., 2024), aiming to find the optimal (or acceptable) discrete solutions to optimize operations and minimize costs under various constraints. Recent advancements in Neural Combinatorial Optimization (NCO) (Bengio et al., 2021; Zhang et al., 2023; Guo et al., 2023; Li et al., 2025b) have favored the efficiency of data-driven problem solving, with the mainstream research goals broadly diverting into a triad of: **a)** squeezing the optimality gap for solving performance to the extreme on a specific task with in-distribution (ID) instances (Joshi et al., 2019; Li et al., 2024; Drakulic et al., 2023); **b)** developing unified representations or pipelines that consistently learns and solves diverse COPs (Li & Liu, 2023; Boisvert et al., 2024; Drakulic et al., 2025; Zhou et al., 2024); and **c)** designing meta-learners to pursue effective adaptation to varied problem types, shifted scales, and unseen data distributions, thereby enhancing the generalization capability on the out-of-distribution (OOD) instances (Qiu et al., 2022; Wang & Li, 2023; Chen et al., 2023; Dernedde et al., 2024).

For the aforementioned **a) task-specific methods**, recent diffusion models, which estimate high-quality solution distributions for individual CO tasks (Sun & Yang, 2023; Li et al., 2023), have significantly improved solving quality on graph-based problems, particularly for in-distribution cases. For instance, Fast-T2T (Li et al., 2024) achieves an optimality gap of less than 1% on TSP-1k, and DiffUCO (Sanokowski et al., 2024) closely approaches Gurobi on the Maximum Clique problem. However, these methods remain challenged by the need for high computational costs and lengthy training convergence times (Table **37**). For **b) cross-task frameworks**, one way is to create unified problem representations, e.g., encoding COPs into abstract syntax trees (Boisvert et al., 2024) or reducing them to matrix-encoded general TSP (Pan et al., 2025b), yet causing extra overhead for task reformulation. In the specific domain of Vehicle Routing Problem (VRP), the works (Zhou et al., 2024; Liu et al., 2024) propose general VRP solvers using mixture-of-experts (MoE) for intra-family variants. While studies such as (Drakulic et al., 2025; Li et al., 2025a) innovate model architectures for cross-problem learning, their

---

[1]School of Artificial Intelligence and School of Computer Science, Shanghai Jiao Tong University [2]Shanghai Innovation Institute, Shanghai, China. Correspondence to: Junchi Yan <yanjunchi@sjtu.edu.cn>.

*Proceedings of the 43ʳᵈ International Conference on Machine Learning*, Seoul, South Korea. PMLR 306, 2026. Copyright 2026 by the author(s).

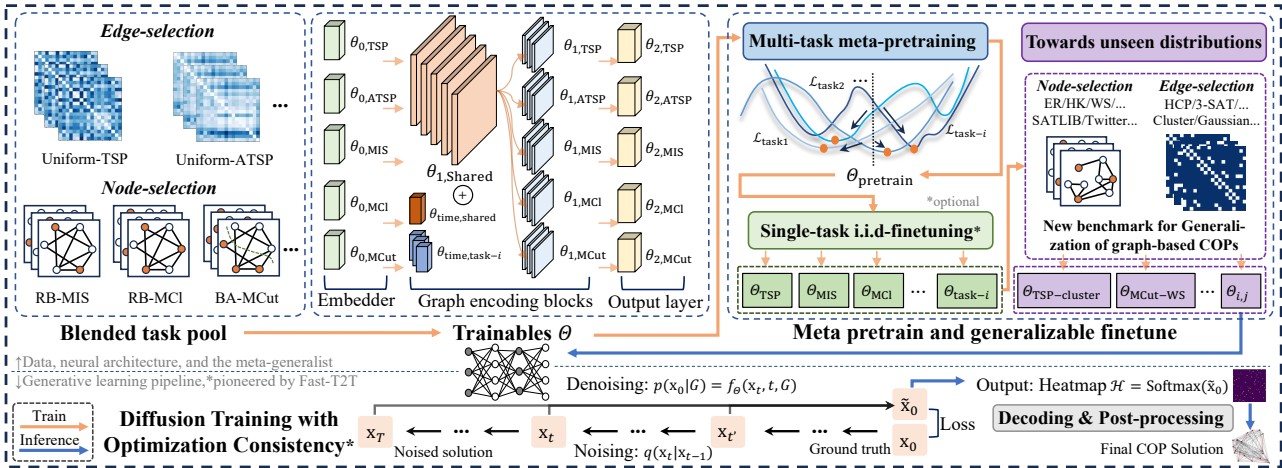

*Figure 1.* Overview of M$^2$GenCO framework: a multi-task meta-pretrain stage and a task-/distribution-specific finetune stage, trained by the established discrete diffusion-based generative supervised learning paradigm for combinatorial optimization on graphs.

sequential solution construction nature hinders efficiency and limits its compatibility with more advanced backbones. Furthermore, for the last related direction in the scope of this paper, i.e., **c) meta-learners**, DIMES (Qiu et al., 2022) first adopts the meta-learning mechanism in a neural CO solver via reinforcement learning (RL) and is evaluated on TSP and MIS only. Meta-EGN (Wang & Li, 2023), built upon (Karalias & Loukas, 2020), first adopts meta-learning for unsupervised learning (UL) on node-oriented COPs like MCl. These methods, though insightful, are limited by their choice of either RL or UL paradigm, as supervised learning (SL) is always time-consuming for high-quality labels. More fundamentally, they constrain the scope of meta-learning to task-specific optimization, i.e., treating different instances of the same problem class (rather than distinct problem types) as meta-tasks, which departs from the original multi-task principle as in Finn et al. (2017).

Naturally motivated by these observations, we take the initiative to practice the technically non-trivial integration of expressive generative modeling with the underexplored setting of multi-task meta-NCO solving. In this paper, we develop a cost-effective multi-task meta-learning framework that empirically advances cross-problem generality without explicit transformation of task representations, and can be readily instantiated with the sophisticated generative neural backbones and supervised training scheme. Specifically, we propose M$^2$GenCO, a **M**ulti-task **M**eta-learning paradigm for **G**enerative **C**ombinatorial **O**ptimization on graphs. In this paradigm, the potent diffusion modeling is leveraged to precisely capture instance-wise distribution, compensating for the relatively weak expressivity of the training paradigms employed by previous multi-task general solvers. Conversely, the meta scheme, formulated across different problem types, endows the model with an enhanced few-shot adaptability to new data distributions via lightweight

finetuning. To our best knowledge, this is among the earliest efforts to: 1) define the meta-learning *tasks* as instances from different COP types; 2) adapt the discrete diffusion backbone into a multi-task meta-solving framework.

As shown in Fig. 1, the diffusion backbone first undergoes the multi-task meta-pretraining, where each batch of instances corresponds to a specific task type in the inner loop, while the task type varies across batches in the outer loop of meta-model updates. This stage initializes the model with generalized knowledge spanning multiple COP types. Subsequently, the model is finetuned on quite few (OOD) query data for certain tasks with specific distributions, enabling rapid and effective adaptation. To instantiate such pipeline, we tailor a lightweight architecture (compared to existing single-task diffusion models for CO (Sun & Yang, 2023; Li et al., 2023)) that integrates task-specific and parameter-shared embedders, with an intermediate node-embedding-based per-layer message-passing scheme to enhance the traditional graph convolutional network (GCN) (Joshi et al., 2019), especially when node features are lacking, e.g., the Asymmetric TSP (ATSP) instances without coordinates.

Further, by identifying the absence of a standardized evaluation protocol in the literature for testing the generalization capabilities of NCO methods, we construct a systematic benchmark comprising support and query sets for few-shot finetuning and testing, involving 5 classic graph-based COPs with diverse distributions and scales. The hope is to promote rigorous assessment of the model adaptability to OOD instances with real-world relevance, a critical aspect insufficiently heeded by existing test datasets. We outline the key comparative points of closely related works in Table 1.

**Main highlights of this paper include: 1)** We propose the M$^2$GenCO framework, first conceptualizing the *task* in meta-learning as the type (and distribution) of distinct

*Table 1.* Comparing M$^2$GenCO and existing NCO methods that conceptually involve 1) meta-learning, 2) "unified"/"general"/"multi-task", or 3) generative modeling for CO. $^*$ denotes partial conformity.

| Method | Meta-learning | Diffusion training | Multi-task | Bench-mark | Evaluated Problems | Method Features |
|---|---|---|---|---|---|---|
| DiffUCO (Sanokowski et al., 2024) | ✗ | ✓ | ✗ | ✗ | MIS, MVC, MCl, MCut | Diffusion+UL |
| (Fast-)T2T (Li et al., 2024; 2023) | ✗ | ✓ | ✗ | ✗ | TSP, MIS | Diffusion+SL+GS |
| DIMES (Qiu et al., 2022) | ✓ | ✗ | ✗ | ✗ | TSP, MIS | Meta+RL+AS |
| Meta-EGN (Wang & Li, 2023) | ✓ | ✗ | ✗ | ✗ | MIS, MVC, MCl | Meta+UL |
| MOCO (Dernedde et al., 2024) | ✓ | ✗ | ✗ | ✗ | TSP, MIS | Meta+RL, Dual GNN |
| EMNH (Chen et al., 2023) | ✓ | ✗ | ✓$^*$ | ✗ | Bi-/Tri-TSP/CVRP/KP | Meta+RL,$^*$MOCOP |
| MAB-MTL (Wang & Yu, 2023) | ✗ | ✗ | ✓ | ✗ | TSP, CVRP, OP, KP | RL, i.i.d evaluation |
| MVMoE (Zhou et al., 2024) | ✗ | ✗ | ✓$^*$ | ✗ | 10+ VRP variants | RL+MoE, $^*$VRPs only |
| GOAL (Drakulic et al., 2025) | ✗ | ✗ | ✓ | ✗ | ATSP, CVRP, OP, JSSP | Sequential-SL+finetune |
| UniCO (Pan et al., 2025b) | ✗ | ✓$^*$ | ✓$^*$ | ✗ | TSP, ATSP, 3SAT, HCP | RL/GenSL, $^*$reduced to TSP |
| **M$^2$GenCO** (Ours) | ✓ | ✓ | ✓ | ✓ | TSP, ATSP, MIS, MCut, MCl, over 9+ distributions | Meta-Diffusion+SL pretrain, o.o.d finetune & benchmark |

COPs for cross-problem pretraining, rather than instance-wise optimization. **2)** This framework bridges the generative paradigm (with strong task-specific fitting capability) and the multi-task meta-learning mechanism (with effective OOD adaptation), offering synergy towards unified distribution learning and efficient generalization for graph-based COPs. **3)** We construct systematic multi-distribution benchmarks to enhance the evaluation protocol for NCO methods (usually tested on uniform instances only) with richer real-world implications. **4)** Extensive experiments on 5 graph-based COPs across data distributions and scales show that M$^2$GenCO not only achieves SOTA performance but also realizes significant inference speedup and consumes only approximately one-tenth of the computational resources required by mainstream generative methods.

**Conflict of Interest Disclosure.** We declare that we have no financial or other substantive conflicts of interest that could reasonably be perceived to influence the work presented in this paper. In particular, this work does not evaluate or promote any product, system, model, or technology developed by a company that employs any of the authors.

## 2. Related Works

**Problem-specific Methods for CO. 1) Exact/Heuristic Solvers.** Representatively, CPLEX (Studio, 2020) and Gurobi (Gurobi Optimization, 2023) are globally leading exact and generic optimizers. For certain tasks, Concorde (Applegate et al., 2006), LKH (Helsgaun, 2017), GA-EAX (Nagata & Kobayashi, 2013) for TSP, and KaMIS (Lamm et al., 2016) for MIS, etc., are designed with reputed solving quality based on expert knowledge. **2) Neural Local-Constructive Solvers.** Recent methods of this type, e.g., AM (Kool et al., 2018), SYM-NCO (Kim et al., 2022), MatNet (Kwon et al., 2021), POMO (Kwon et al., 2020), BQ-NCO (Drakulic et al., 2023), etc., cast COPs as Markov

Decision Processes (MDPs) and the neural networks are trained to predict the best next-step action given the current state. **3) Neural Global-Predictive Solvers.** Approaches of this category, e.g., GCN4CO (Joshi et al., 2019), DIFUSCO (Sun & Yang, 2023), UTSP (Min et al., 2023), T2TCO (Li et al., 2023; 2024), DIMES (Qiu et al., 2022), etc., generally use neural networks to globally predict the likelihood of the variables being selected, and decode the heatmaps for solutions of certain tasks. **4) Besides the majority**, there are also branches researching neural-enhanced heuristics (Hudson et al., 2022; Xin et al., 2021; Ye et al., 2024a; Zheng et al., 2021; Sui et al., 2023; da Costa et al., 2020; Sui et al., 2021; Ma et al., 2023; Kool et al., 2022; Kim et al., 2025), solution optimization approaches (Ma et al., 2023; Chen & Tian, 2019; Fu et al., 2021), neural divide-and-conquer frameworks (Fu et al., 2021; Ye et al., 2024b; Zheng et al., 2024; Luo et al., 2024; Kim et al., 2021), and adaptive solution expansion methods (Ahn et al., 2020; Ma et al., 2025a; Zhou et al., 2025), etc.

**Unified Neural Frameworks for Multi-task CO.** Earlier works provide theoretical analysis for inter-instance generalization (Li & Liu, 2023; Wang et al., 2024a) without proposing specific solvers. MVMoE (Zhou et al., 2024) and MT-NCO (Liu et al., 2024) develop generic solvers for a range of VRP variants. Lately, MAB-MTL (Wang & Yu, 2023), GCNCO (Li et al., 2025a), and GOAL (Drakulic et al., 2025) resort to similar header-encoder-decoder structure to learn shared embeddings for different COPs, but they are generally RL-mannered, hindering the efficiency and scalability (e.g., scarcely evaluated on graph larger than 100 nodes). Other endeavors are towards generic representations, e.g., matrix-encoded TSP in UniCO (Pan et al., 2025b), LLM-driven text-attributed formulation in UNCO (Jiang et al., 2024), and abstract syntax trees in (Boisvert et al., 2024), etc., for consistent learning of different COPs, yet costing extra overhead of problem transformation or re-formulation.

**Meta-learning for CO.** The model-agnostic meta-learning (Finn et al., 2017; Antoniou et al., 2019; Yu et al., 2020) propose to train a model on a variety of tasks and then quickly adapts to new tasks with few data. (Wang et al., 2021) theoretically bridges the view of multi-task and meta-learning. In the context of CO solving, (Chen et al., 2023) targets meta-learning for CO, yet drawing from a different perspective from ours, focusing on the realm of multi-objective COP (MOCOP). Recently, DIMES (Qiu et al., 2022), Meta-EGN (Wang & Li, 2023) and (Dernedde et al., 2024), combine meta-learning (RL/UL) with in-distribution CO solving and per-instance finetuning for single task.

**Note.** We provide in Appendix **C** a retrospect of more related works that are not discussed in detail above.

## 3. Preliminary: Problem Formulation

In this paper, our research focus is on the broad family of **graph-based** COPs, evaluating (and benchmarking) 5 tasks: node-oriented *Maximum Independent Set* (MIS), *Maximum Clique* (MCl), *Maximum Cut* (MCut), and edge-oriented *Traveling Salesman Problem* (TSP) and *Asymmetric TSP* (ATSP). Definitions of the covered COPs and a look-up table of mathematical notations used hereinafter are presented in **Appendix A**. Adopting established conventions (Li et al., 2025b; Qiu et al., 2022; Li et al., 2023; Fu et al., 2021), these COPs are consistently characterized by graphs $G = (\mathcal{V}, \mathcal{E})$, where $\mathcal{V}$ and $\mathcal{E}$ denote the set for nodes and edges respectively. Let $\mathbf{x} = \{0, 1\}^N$ denote the decision variables (solution space) of a given COP, i.e., For MCl and MIS, $N = |\mathcal{V}|$ and $\mathbf{x}_i$ indicates whether the node $i$ is selected. For MCut, $N = |\mathcal{V}|$ and $\mathbf{x}_i$ indicates which subset node $i$ belongs to. For TSP and ATSP, $N = |\mathcal{V}|^2(= |\mathcal{E}|)$ and $\mathbf{x}_{i \cdot |\mathcal{V}| + j}$ indicates whether the edge $(i, j)$ is included in the solution tour. Mathematically, the optimization objective is to find optimal solution $\mathbf{x}^* = \underset{\mathbf{x} \in \Omega}{\operatorname{argmin}} c(\mathbf{x}, G)$, where $\Omega \subseteq \{0, 1\}^N$ denotes the feasible set of $\mathbf{x}$ satisfying the constraints, and $c(\cdot, G) : \{0, 1\}^N \to \mathbb{R}_{\geq 0}$ is the objective function of corresponding problems.

## 4. Methodology

**Overview.** Meta-learning (Finn et al., 2017) is designed to learn generalizable priors across a family of tasks so that the model can rapidly adapt to a new task with only few-shot supervision. Existing meta-NCO works (Qiu et al., 2022; Wang & Li, 2023) typically define the distribution of the meta-task at instance-level, focusing on instance-wise adaptation within a single problem class. In contrast, we propose the ***generalized, multi-task, and cross-distribution formulation***, where a meta-task is defined at the level of a (COP type $i$, graph distribution $j$, scale) tuple. The model is, therefore, jointly meta-pretrained across similarly structured

yet distinct graph-based COPs, explicitly optimizing for a cross-task, adaptation-friendly initialization. Algorithm **1** in Appendix **B** presents the complete streamlined pipeline.

Importantly, although our setting involves distribution shifts across COP or graph types, it remains fundamentally distinct from standard *transfer learning* as we employ a bi-level inner–outer optimization objective that explicitly optimizes the initialization for adaptability, rather than optimizing for source-task performance alone. It can be naturally understood as ***an extension of conventional MAML-style meta-learning***, where the only difference lies in how tasks are sampled: rather than drawing from a single-task family i.e., $G \sim p(\mathcal{T})$, we sample instances from the joint distribution over an expanded task family, i.e., $G \sim p(\{\mathcal{T}_{i,j}\})$. To avoid overloading terminology, we describe our approach as a multi-task, cross-distribution meta-learning–inspired framework, and leave a detailed discussion of the conceptual nuances from MAML to NCO in Appendix **H.2**.

### 4.1. Generalized Meta Pretraining: In search of Cross-problem Initializer

In practice, we follow Ma et al. (2025a) to divide the covered graph-based COPs into node-oriented and edge-oriented tasks and train respective meta models for them. To maintain consistency with mainstream studies, for node-oriented-tasks, the problem pool includes $m = 3$ tasks: MIS (RB), MCl (RB), and MCut (BA), whereas for edge-oriented tasks, the problem pool consists of $m = 2$ tasks, i.e., uniformly distributed Euclidean 2D-TSP and ATSP instances satisfying triangle inequality (Pan et al., 2025b; Kwon et al., 2021). In the pretraining stage, first, a sequence of task types $\mathcal{T}_1, \cdots \mathcal{T}_k$ is randomly determined (e.g., [MIS, MCut, MIS, MCl, ...]) with a hyper-parameter $k$ controlling its length. Then, the task sequence is traversed and corresponding instance batches are sampled via $G_i \sim p(\mathcal{T}_i), i = 1, \cdots k$, alongside their optimal solutions $\mathbf{x}^*$ as ground truth for supervision. During the $n$-th *outer* iteration, the trainable parameters of a neural network, i.e., $\theta^{(n)}$, is repeatedly loaded to try adapting to the $k$ distinctive task batches with only one-step *inner* gradient descent (Wang & Li, 2023):

$$\theta_i'^{(n)} = \theta^{(n)} - \alpha \nabla_{\theta^{(n)}} \mathcal{L}_{\mathcal{T}_i}(f_{\theta^{(n)}}) \text{ for } i = 1, \cdots, k, \quad (1)$$

where $\alpha$ is the inner learning rate, $\mathcal{L}(\cdot)$ is any loss function (specified in Sec. **4.2**), and $f_\theta(\cdot)$ is any neural backbone for heatmap prediction (specified in Sec. **4.4**). Subsequently, the neural network is optimized via AdamW (Loshchilov & Hutter, 2018) using the temporarily stored adapted parameters $\theta'^{(n)}$ and the meta-objective $\mathcal{L}_{\text{meta}}(f_{\theta^{(n)}}) = \mathbb{E}_{\mathcal{T}_i \sim p(\mathcal{T})} \mathcal{L}_{\mathcal{T}_i}(f_{\theta_i'^{(n)}})$. Mathematically, the outer update:

$$\theta^{(n+1)} = \theta^{(n)} - \beta \nabla_{\theta^{(n)}} \sum_{\mathcal{T}_i \sim p(\mathcal{T})} \mathcal{L}_{\mathcal{T}_i}(f_{\theta_i'^{(n)}}), \quad (2)$$

where $\beta$ is the outer learning rate. To ensure a consistent scale of gradients across tasks, the gradients are normalized over the $k$ task batches, i.e., the adapted gradients are mended prior to the Eq. **2** via

$$\nabla_{\theta^{(n)}} \mathcal{L}_{\mathcal{T}_i}\big(f_{\theta_i'^{(n)}}\big) \leftarrow \frac{\nabla_{\theta^{(n)}} \mathcal{L}_{\mathcal{T}_i}\big(f_{\theta_i'^{(n)}}\big)}{\left[\sum_i^k \big(\nabla_{\theta^{(n)}} \mathcal{L}_{\mathcal{T}_i}(f_{\theta_i'^{(n)}})\big)^2\right]^{\frac{1}{2}}}. \quad (3)$$

After iterations, we obtain a initializer model $\theta_{\text{meta}}$ which contains sufficient knowledge from various COPs for fast adaptive tuning on specific tasks or distributions later. This pipeline evolves the research attempt for meta solvers of certain single COP (Qiu et al., 2022; Wang & Li, 2023; Dernedde et al., 2024), and better aligns with the essence of meta-learning in the broader machine learning context.

## 4.2. Generative Modeling: On Effective Distribution Learning

Previous meta solvers for CO generally employed reinforcement or energy-based unsupervised learning, struggling to scale to large instances without the aid of post-processing techniques, and failing to keep pace with the latest advancements in model backbones and training schemes. Recognizing the potential of generative models for single COP solving, we adopt a graph-based diffusion approach to mitigate the limited expressivity of existing meta CO solvers.

As stated in Sec. **3**, the solution space of a COP graph instance with $N$ decision variables can be characterized by an $N$-dimensional Bernoulli distribution $p(\mathbf{x}) \in [0,1]^{N \times 2}$. Inspired by (Sun & Yang, 2023; Li et al., 2023; Sanokowski et al., 2024), the conditional generation of optimal solution is adopted to instantiate the meta-training paradigm. Generally, the neural network is expected to predict $p_\theta(\mathbf{x}|G)$ through a manual noising procedure and a learned denoising process. At training, given the ground truth solution as the starting point $\mathbf{x}_0 = \mathbf{x}^*$, the categorical noise is added to $\mathbf{x}_0$ sequentially as a trajectory $\mathbf{x}_{0:T} = \mathbf{x}_0, \mathbf{x}_1, \mathbf{x}_2, \cdots, \mathbf{x}_T$, with each $\mathbf{x}_i$ sampled from the distribution

$$q(\mathbf{x}_t|\mathbf{x}_0) = p(\mathbf{x}_0)\overline{\mathbf{Q}}_t, \ \mathbf{Q}_t = \begin{bmatrix} r_t & 1-r_t \\ 1-r_t & r_t \end{bmatrix}, \quad (4)$$

where $\overline{\mathbf{Q}}_t = \mathbf{Q}_1 \mathbf{Q}_2 \cdots \mathbf{Q}_t$ is the cumulative transition probability matrix with $r_t \in [0,1]$ to ensure double-stochasticity. While vanilla diffusion requires multiple inference steps to model $p_\theta(\mathbf{x}_{t-1}|\mathbf{x}_t, G)$ which is time-consuming in practice, consistency models (CM) (Song et al., 2023) have introduced the self-consistency scheme to map any point at any time step to the starting point of the trajectory. We follow Fast-T2T (Li et al., 2024) to instantiate the neural network $f_\theta$ conceptualized in Sec. **4.1** with the learnable consistency function $f_\theta(\mathbf{x}_t, t, G)$. The self-consistency mechanism is enforced by two noised points $\mathbf{x}_{t_1}, \mathbf{x}_{t_2}$ sampled on the same

trajectory that minimizes the distances between their respective mapping to $\mathbf{x}_0$. In practice, the triangle inequality is employed to bound and reformulate the training objective, with $d(\cdot, \cdot)$ being a distance metric and practically instantiated by the binary cross entropy loss ($\mathcal{L}_{\text{BCE}}$) following Joshi et al. (2019); Li et al. (2024); Sun & Yang (2023) to leverage the labeled data for supervised learning.

$$\mathcal{L}_{\text{CM}}(\theta) = \mathbb{E}\Big[d\big(f_\theta(\mathbf{x}_{t_1}, t_1, G), f_\theta(\mathbf{x}_{t_2}, t_2, G)\big)\Big] \leq$$
$$\mathbb{E}\Big[d\big(f_\theta(\mathbf{x}_{t_1}, t_1, G), \mathbf{x}^*\big) + d\big(f_\theta(\mathbf{x}_{t_2}, t_2, G), \mathbf{x}^*\big)\Big]. \quad (5)$$

This way, the meta-objective calculated in Eq. **2** w.r.t. the $n$-th outer iteration is embodied as:

$$\mathcal{L}_{\text{meta}}(f_{\theta^{(n)}}) = \mathbb{E}_{\mathcal{T}_i \sim p(\mathcal{T})}\Big[\mathcal{L}_{\text{BCE}}\big(f_{\theta_i'^{(n)}}(\mathbf{x}_{t_1}, t_1, G_i), \mathbf{x}^*\big)$$
$$+ \mathcal{L}_{\text{BCE}}\big(f_{\theta_i'^{(n)}}(\mathbf{x}_{t_2}, t_2, G_i), \mathbf{x}^*\big)\Big], \quad (6)$$

where $G_i$ is the training instance batch sampled from problem type $i$, and $\mathbf{x}_{t_1}, \mathbf{x}_{t_2}, \mathbf{x}^*$ are the solution points corresponded with $G_i$ without abusing the notation of $i$ or $\mathcal{T}_i$. Thus far, the discrete diffusion with optimization consistency has been readily incorporated for modeling the various COPs and serves to capture instance-wise characteristics orthogonally along the multi-task meta-learning process.

## 4.3. Few-shot Finetuning: Towards Fast Adaptation and Robust Extrapolation

In former works, the initializer model $\theta_{\text{meta}}$ undergoes "finetuning" towards incoming in-distribution testing instances of the same problem type by replicating $T$ inner meta-updates (Qiu et al., 2022; Wang & Li, 2023). Noting the time-consuming gradient descents at testing phase which is far from practical COP solving, we conduct the adaptation process in a more mainstream way, i.e., tuning $\theta_{\text{meta}}$ offline on few-shot support data of a given task type and distribution. Specifically, we constructed a new systematic benchmark dataset including 5 COPs with multiple distributions in the support-query (with a ratio of 5:1) manner, following the vanilla setting of MAML in the image fields (Finn et al., 2017). Upon these datasets, we equip the model with the shared starting point of pretrained $\theta_{\text{meta}}$, and conduct in-distribution (optional) as well as OOD finetuning on specific problem type and distribution. To gain the best efficiency, a straightforward supervised learning of binary classification (following Kipf & Welling (2016)) is adopted for MIS, MCl and edge-oriented tasks with $\mathcal{L}_{\text{sup-tune}} = \text{BCE}(\widetilde{\mathbf{x}}_0, \mathbf{x}^*)$. For MCut, we follow Ma et al. (2025a) to employ the unsupervised energy loss $\mathcal{L}_{\text{unsup-tune(MCut)}} = \sum_{(i,j) \in \mathcal{E}} \lambda(2\mathbf{x}_i - 1)(2\mathbf{x}_j - 1)$ due to its unconstrained nature (Sanokowski et al., 2024). After finetuning, a bunch of specialized models, i.e., $\theta_{\mathcal{T}_i,j}$, are obtained for problem type $i$ with distribution $j$, respectively.

*Table 2.* Main results (full in Table **18**) for edge-oriented problems. ‡ denotes applying Monte Carlo tree search (following Fu et al. (2021); Qiu et al. (2022), etc) for post inference improvement. Competitive learning-based results with ours are **bolded**.

| Method | TSP-Gaussian-50 | | | TSP-Gaussian-100 | | | TSP-Gaussian-200 | | | TSP-Gaussian-500 | | |
|---|---|---|---|---|---|---|---|---|---|---|---|---|
| | Obj.↓ | Gap↓ | Time↓ | Obj.↓ | Gap↓ | Time↓ | Obj.↓ | Gap↓ | Time↓ | Obj.↓ | Gap↓ | Time↓ |
| Concorde (Applegate et al., 2006) | 23.84* | 0.00% | 0.17s | 34.03* | 0.00% | 0.44s | 48.13* | 0.00% | 1.75s | 77.52* | 0.00% | 19.95s |
| LKH3 (Helsgaun, 2017) | 23.84 | 0.00% | 0.06s | 34.03 | 0.00% | 0.14s | 48.13 | 0.00% | 0.37s | 77.52 | 0.00% | 1.22s |
| GNNGLS (Hudson et al., 2022) | 25.18 | 5.61% | 0.01s | 37.29 | 9.55% | 0.01s | 55.63 | 15.57% | 0.09s | – | – | – |
| DIMES (Qiu et al., 2022) | 26.37 | 10.63% | **0.00s** | 38.16 | 12.13% | **0.01s** | 54.98 | 14.21% | 0.09s | 89.14 | 14.98% | 0.37s |
| UTSP (Min et al., 2023) | 29.77 | 24.89% | 0.01s | 44.71 | 31.40% | 0.01s | – | – | – | 97.08 | 25.24% | **0.05s** |
| SymNCO (Kim et al., 2022) | **23.96** | **0.49%** | 0.07s | 34.81 | 2.26% | 0.32s | 50.48 | 4.88% | 0.47s | – | – | – |
| GOAL (Drakulic et al., 2025) | 24.09 | 1.06% | 0.48s | **34.52** | **1.43%** | 0.98s | **49.17** | **2.16%** | 2.02s | 80.26 | 3.54% | 5.21s |
| DIFUSCO (Sun & Yang, 2023) | 24.56 | 2.97% | 0.28s | 35.97 | 5.66% | 0.38s | 52.45 | 8.96% | 1.23s | 105.91 | 36.61% | 8.99s |
| M²GenCO (ours) | **24.07** | **0.95%** | **0.01s** | **34.57** | **1.58%** | **0.02s** | **49.24** | **2.31%** | **0.06s** | **79.80** | **2.95%** | **0.35s** |
| M²GenCO (ours)‡ | **23.90** | **0.25%** | **0.01s** | **34.26** | **0.68%** | **0.02s** | **48.69** | **1.18%** | **0.09s** | **78.90** | **1.78%** | **0.72s** |

| Method | TSP-Cluster-50 | | | TSP-Cluster-100 | | | TSP-Cluster-200 | | | TSP-Cluster-500 | | |
|---|---|---|---|---|---|---|---|---|---|---|---|---|
| | Obj.↓ | Gap↓ | Time↓ | Obj.↓ | Gap↓ | Time↓ | Obj.↓ | Gap↓ | Time↓ | Obj.↓ | Gap↓ | Time↓ |
| Concorde (Applegate et al., 2006) | 3.73* | 0.00% | 0.14s | 5.53* | 0.00% | 0.29s | 6.91* | 0.00% | 0.97s | 10.72* | 0.00% | 5.07s |
| LKH3 (Helsgaun, 2017) | 3.73 | 0.00% | 0.04s | 5.53 | 0.01% | 0.11s | 6.91 | 0.01% | 0.54s | 10.73 | 0.02% | 2.06s |
| GNNGLS (Hudson et al., 2022) | 4.46 | 19.51% | 0.01s | 6.89 | 24.79% | 0.01s | 9.10 | 31.66% | **0.09s** | – | – | – |
| DIMES (Qiu et al., 2022) | 4.14 | 10.94% | **0.00s** | 6.38 | 15.43% | **0.01s** | 8.28 | 19.86% | **0.09s** | 13.00 | 21.26% | 1.81s |
| UTSP (Min et al., 2023) | 4.60 | 23.50% | 0.01s | 6.90 | 25.06% | 0.01s | – | – | – | 14.28 | 33.20% | **0.06s** |
| SymNCO (Kim et al., 2022) | 3.74 | 0.34% | 0.07s | 5.59 | **1.14%** | 0.12s | 7.13 | 3.16% | 0.61s | – | – | – |
| GOAL (Drakulic et al., 2025) | 3.76 | 0.69% | 0.50s | 5.62 | 1.65% | 1.11s | **7.09** | **2.57%** | 2.24s | 11.24 | 4.79% | 5.25s |
| DIFUSCO (Sun & Yang, 2023) | 3.92 | 5.13% | 0.28s | 5.76 | 4.14% | 0.45s | 7.94 | 14.88% | 1.23s | 15.21 | 41.84% | 7.23s |
| M²GenCO (ours) | **3.74** | **0.29%** | **0.01s** | **5.59** | **1.18%** | **0.02s** | **7.14** | **3.21%** | **0.12s** | **11.17** | **4.18%** | **0.60s** |
| M²GenCO (ours)‡ | **3.73** | **0.06%** | **0.01s** | **5.55** | **0.43%** | **0.02s** | **7.05** | **1.94%** | **0.16s** | **11.06** | **3.10%** | **0.98s** |

| Method | ATSP-HCP-50 | | | ATSP-HCP-100 | | | ATSP-HCP-200 | | | ATSP-HCP-500 | | |
|---|---|---|---|---|---|---|---|---|---|---|---|---|
| | Obj.↓ | Gap↓ | Time↓ | Obj.↓ | Gap↓ | Time↓ | Obj.↓ | Gap↓ | Time↓ | Obj.↓ | Gap↓ | Time↓ |
| LKH3 (Helsgaun, 2017) | 0.00 | N/A | 0.11s | 0.00 | N/A | 0.21s | 0.00 | N/A | 0.36s | 0.00 | N/A | 1.41s |
| MatNet (Kwon et al., 2021) | **1.33** | N/A | 0.05s | 17.54 | N/A | 0.09s | 97.36 | N/A | 0.45s | – | – | – |
| GOAL (Drakulic et al., 2025) | 3.61 | N/A | 0.50s | 1.75 | N/A | 0.98s | 4.38 | N/A | 2.21s | 3.38 | N/A | 7.25s |
| UniCO (Pan et al., 2025b) | 6.11 | N/A | 0.06s | 17.63 | N/A | 0.10s | 85.25 | N/A | 0.48s | – | – | – |
| DIMES (Qiu et al., 2022) | 4.89 | N/A | 0.03s | 5.45 | N/A | 0.06s | 5.47 | N/A | 0.22s | – | – | – |
| M²GenCO (ours) | **1.36** | N/A | **0.01s** | **1.28** | N/A | **0.01s** | **0.93** | N/A | **0.03s** | **0.68** | N/A | **0.17s** |

| Method | ATSP-SAT-54 | | | ATSP-SAT-102 | | | ATSP-SAT-200 | | | ATSP-SAT-507 | | |
|---|---|---|---|---|---|---|---|---|---|---|---|---|
| | Obj.↓ | Gap↓ | Time↓ | Obj.↓ | Gap↓ | Time↓ | Obj.↓ | Gap↓ | Time↓ | Obj.↓ | Gap↓ | Time↓ |
| LKH3 (Helsgaun, 2017) | 0.15 | N/A | 0.08s | 0.08 | N/A | 0.13s | 0.13 | N/A | 0.19s | 0.43 | N/A | 0.78s |
| MatNet (Kwon et al., 2021) | 32.37 | N/A | 0.05s | 38.71 | N/A | 0.09s | 162.80 | N/A | 0.44s | – | – | – |
| GOAL (Drakulic et al., 2025) | 12.67 | N/A | 0.54s | 14.67 | N/A | 1.03s | 18.78 | N/A | 2.17s | 60.19 | N/A | 7.52s |
| UniCO (Pan et al., 2025b) | 30.15 | N/A | 0.06s | 34.80 | N/A | 0.11s | 97.64 | N/A | 0.50s | – | – | – |
| DIMES (Qiu et al., 2022) | 7.83 | N/A | 0.03s | 13.17 | N/A | 0.07s | 21.81 | N/A | 0.21s | – | – | – |
| M²GenCO (ours) | **2.31** | N/A | **0.01s** | **4.06** | N/A | **0.01s** | **5.10** | N/A | **0.03s** | **2.31** | N/A | **0.14s** |

## 4.4. Model Architecture

**Overview.** Drawing inspiration from Drakulic et al. (2025); Li et al. (2025a), M²GenCO extends the vanilla GCN as backbone (Joshi et al., 2019). At high-level, the graph encoder consists of three sections, i.e., the embedders (**Embed**$_{\theta_0}$), graph convolutional blocks (**Conv**$_{\theta_1}$), and the output layers (**Out**$_{\theta_2}$), where the learnables $\Theta = \{\theta_0, \theta_1, \theta_2\}$ of the model are categorized into "shared" or "separate" classes. In practice, the model is equipped with individual embedders for each task type, followed by the main of encoding blocks that are shared across tasks. Then, the features with generalized knowledge are again processed by task-specific encoding blocks and output layers. We leave the detailed (e.g., per-layer message-passing manners) mathematics of the architecture in Appendix **E**.

## 5. Experiments

**Metrics. 1) Obj.**: average objective value on certain dataset. **2) Gap**: optimality gap w.r.t. the objective of the reference solutions. **3) Time**: per-instance average solving time.

**Datasets.** Following conventions, we train meta model on RB (Xu et al., 2005) graphs for MIS and MCl, BA (Barabási & Albert, 1999) graphs for MCut, and uniform TSP and ATSP. For model evaluation, beyond the commonly tested ID instances, we test the methods on our proposed multi-distribution benchmarks: ER (Erdős & Rényi, 1960), HK (Holme & Kim, 2002) and WS (Watts & Strogatz, 1998) graphs for MIS, MCl and MCut; gaussian, cluster distributions for TSP; and HCP, SAT distributions for ATSP. Details of the collated benchmark are introduced in Appendix **D**.

**Baselines. Learning-free solvers:** Learning-free solvers

*Table 3.* Main results (full in **Table 17** with larger instances) for node-oriented problems. ‡ denotes applying the energy-based sampling (following Feng & Yang (2025); Ma et al. (2025a), etc.) for refinement. Learning-based results that are *competitive* with ours are **bolded**.

| Method | MIS-BA-200-300 | | | MIS-HK-200-300 | | | MIS-WS-200-300 | | | MIS-SATLIB | | |
|---|---|---|---|---|---|---|---|---|---|---|---|---|
| | Obj.↑ | Gap↓ | Time↓ | Obj.↑ | Gap↓ | Time↓ | Obj.↑ | Gap↓ | Time↓ | Obj.↑ | Gap↓ | Time↓ |
| KaMIS (Lamm et al., 2016) | 72.77* | 0.00% | 52.94s | 79.37* | 0.00% | 54.17s | 76.90* | 0.00% | 51.49s | 425.95* | 0.00% | 24.37s |
| GCN4CO (Joshi et al., 2019) | 66.96 | 8.00% | 0.02s | 74.86 | 5.68% | 0.02s | 72.12 | 6.24% | 0.02s | 408.33 | 4.41% | 0.03s |
| VAG-CO (Sanokowski et al., 2023) | 66.34 | 8.85% | **0.01s** | 73.69 | 7.16% | **0.01s** | 71.20 | 7.43% | **0.01s** | 339.20 | 20.36% | **0.02s** |
| Meta-EGN (Wang & Li, 2023) | 58.90 | 18.98% | 0.01s | 64.54 | 18.62% | 0.01s | 62.53 | 18.67% | 0.01s | 399.83 | 6.14% | 0.02s |
| Fast-T2T (Li et al., 2024) | 69.56 | 4.43% | 0.26s | 76.97 | 3.03% | 0.35s | 74.25 | 3.47% | 0.37s | 410.26 | 3.69% | 0.38s |
| DiffUCO (Sanokowski et al., 2024) | 68.80 | 5.45% | 0.26s | 74.05 | 6.64% | 0.27s | 71.73 | 6.72% | 0.27s | 400.50 | 5.98% | 0.65s |
| GOAL (Drakulic et al., 2025) | 62.77 | 13.79% | 0.71s | 69.73 | 12.24% | 0.80s | 67.52 | 12.28% | 0.80s | 418.64 | **1.72%** | 24.80s |
| DIFUSCO (Sun & Yang, 2023) | 66.03 | 9.29% | 0.59s | 73.80 | 7.01% | 0.63s | 70.97 | 7.74% | 0.61s | 408.63 | 4.07% | 0.83s |
| M³GenCO (ours) | **70.01** | **3.80%** | **0.01s** | **77.40** | **2.49%** | **0.01s** | **74.80** | **2.74%** | **0.01s** | **414.63** | **2.66%** | **0.02s** |
| M²GenCO (ours)‡ | **72.66** | **0.14%** | **0.03s** | **79.31** | **0.08%** | **0.03s** | **76.81** | **0.11%** | **0.03s** | **420.30** | **1.32%** | **1.09s** |

| Method | MCl-BA-200-300 | | | MCl-HK-200-300 | | | MCl-WS-200-300 | | | MCl-TWITTER | | |
|---|---|---|---|---|---|---|---|---|---|---|---|---|
| | Obj.↑ | Gap↓ | Time↓ | Obj.↑ | Gap↓ | Time↓ | Obj.↑ | Gap↓ | Time↓ | Obj.↑ | Gap↓ | Time↓ |
| Gurobi (Gurobi Optimization, 2023) | 7.48* | 0.00% | 1.43s | 6.79* | 0.00% | 1.84s | 7.16* | 0.00% | 1.59s | 14.21* | 0.00% | 0.28s |
| GCN4CO (Joshi et al., 2019) | 7.14 | 4.41% | 0.03s | 6.14 | 9.43% | 0.02s | 6.51 | 8.98% | 0.02s | 12.90 | 12.00% | 0.02s |
| VAG-CO (Sanokowski et al., 2023) | 7.05 | 5.62% | 0.01s | 6.05 | 10.76% | **0.01s** | 6.47 | 9.54% | **0.01s** | 11.96 | 16.74% | 0.01s |
| Meta-EGN (Wang & Li, 2023) | 7.06 | 5.58% | **0.01s** | 6.12 | 9.73% | 0.01s | 6.52 | 8.82% | 0.01s | 12.83 | 10.33% | **0.01s** |
| Fast-T2T (Li et al., 2024) | 7.05 | 5.61% | 0.02s | 6.07 | 10.48% | 0.02s | 6.42 | 10.18% | 0.02s | 13.05 | 9.54% | 0.02s |
| GOAL (Drakulic et al., 2025) | 7.18 | 3.88% | 0.10s | 6.40 | 5.63% | 0.10s | 6.88 | 3.84% | 0.14s | **13.26** | **7.12%** | 0.21s |
| DIFUSCO (Sun & Yang, 2023) | 7.11 | 4.86% | 0.56s | 6.18 | 8.84% | 0.56s | 6.66 | 6.88% | 0.55s | 12.79 | 10.72% | 0.52s |
| M³GenCO (ours) | **7.30** | **2.30%** | **0.01s** | **6.42** | **5.41%** | **0.01s** | **6.95** | **2.92%** | **0.01s** | **13.11** | **7.84%** | **0.01s** |
| M²GenCO (ours)‡ | **7.48** | **0.00%** | **0.04s** | **6.79** | **0.00%** | **0.04s** | **7.16** | **0.00%** | **0.04s** | **14.21** | **0.00%** | **0.03s** |

| Method | MCut-RB-200-300 | | | MCut-HK-200-300 | | | MCut-WS-200-300 | | | MCut-ER-700-800 | | |
|---|---|---|---|---|---|---|---|---|---|---|---|---|
| | Obj.↑ | Gap↓ | Time↓ | Obj.↑ | Gap↓ | Time↓ | Obj.↑ | Gap↓ | Time↓ | Obj.↑ | Gap↓ | Time↓ |
| Gurobi (Gurobi Optimization, 2023) | 2526.13* | 0.00% | 60.23s | 1540.61* | 0.00% | 60.09s | 872.12* | 0.00% | 60.36s | 23597.93* | 0.00% | 360.13s |
| GCN4CO (Joshi et al., 2019) | 1604.24 | 41.50% | 0.02s | 1161.48 | 24.65% | 0.02s | 421.75 | 51.64% | 0.02s | 20294.07 | 13.98% | **0.01s** |
| VAG-CO (Sanokowski et al., 2023) | 203.32 | 89.57% | **0.01s** | 631.09 | 58.85% | **0.01s** | 471.72 | 45.90% | **0.01s** | 0.00 | N/A | 0.05s |
| Meta-EGN (Wang & Li, 2023) | 1723.53 | 37.55% | 0.01s | 1420.52 | 7.79% | 0.01s | 724.07 | 16.92% | 0.01s | 20061.20 | 14.96% | 0.07s |
| Fast-T2T (Li et al., 2024) | 2323.11 | 7.30% | 0.35s | 1519.22 | 1.39% | 0.38s | **862.69** | **1.09%** | 0.38s | 21685.75 | 8.12% | 0.17s |
| DiffUCO (Sanokowski et al., 2024) | 2229.66 | 11.79% | 0.48s | **1530.31** | **0.67%** | 0.29s | 842.10 | 3.44% | 0.20s | **23054.31** | **2.31%** | 2.00s |
| DIFUSCO (Sun & Yang, 2023) | 2192.14 | 12.92% | 0.66s | 1432.27 | 7.08% | 0.60s | 788.37 | 9.61% | 0.55s | 21199.20 | 10.15% | 3.54s |
| M²GenCO (ours) | **2451.10** | **2.95%** | **0.01s** | **1510.60** | **1.95%** | **0.01s** | **839.47** | **3.76%** | **0.01s** | **22556.57** | **4.43%** | **0.04s** |
| M²GenCO (ours)‡ | **2526.23** | **-0.01%** | **0.05s** | **1540.43** | **0.01%** | **0.05s** | **871.34** | **0.08%** | **0.05s** | **23504.01** | **0.40%** | **0.07s** |

serve to produce reference solutions to compute the optimality gap. 1) Gurobi (Gurobi Optimization, 2023) for MCl/MCut/MIS, 2) KaMIS (Lamm et al., 2016) for MIS, 3) LKH (Helsgaun, 2017) for ATSP/TSP, and 4) Concorde (Applegate et al., 2006) for TSP. **Learning-based solvers:** 1) GCN4CO (Joshi et al., 2019), 2) GNNGLS (Hudson et al., 2022); 3) UTSP (Min et al., 2023) 4) VAG-CO (Sanokowski et al., 2023) 5) Sym-NCO (Kim et al., 2022; Berto et al., 2023), 6) MatNet (Kwon et al., 2021); 7) DIMES (Qiu et al., 2022), 8) Meta-EGN (Wang & Li, 2023); 9) DIFUSCO (Sun & Yang, 2023), 10) DiffUCO (Sanokowski et al., 2024), 11) GOAL (Drakulic et al., 2025), 12) UniCO (Pan et al., 2025b). **Note.** We are committed to encompassing a wide spectrum of learning-based baselines: heatmap-based methods trained either supervised (1-2) or unsupervised (3-4), sequential methods trained by RL (5-6), meta learning-based methods (7-8), diffusion-based methods (9-10), and multi-task frameworks (11-12). Full model settings are in Appendix **F**.

### 5.1. Main Results and Analyses

**Overview: Performance and Efficiency.** Main results for edge- and node-oriented tasks are exhibited in Table **2** and Table **3**. As demonstrated, M²GenCO with sim-

ple greedy decoders outperforms comparable neural methods with an average **9.16%** improvement compared to per-dataset SOTA, meanwhile reducing the per-instance solving time (averaged over all benchmarks and compared with the benchmark-wise best solver) from ∼ 4.5s to ∼ 0.05s (a **95.6x** speedup, detailed in Appendix **G.1**). Results reporting *performance on larger instances*, with *more decimal places* and *standard deviations* are in Appendix **G.2** and **G.3**.

**RQ1: Compared with Meta-learners.** Empirical results show that M²GenCO outperforms Meta-EGN on 22 node-oriented datasets with an average gain of **71.9%**. Advantage is also observed compared with DIMES with an average improvement of **23.9%** on 14 applicable edge-based datasets.

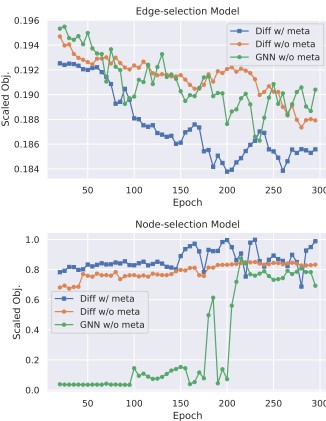

*Figure 2.* Training curve w/ and w/o diffusion model and meta-learning.

*Table 4.* Representative results of the objective with (w/) and without (w/o) finetuning on certain OOD distributions.

| Mode | Node-200-300 (Obj.↑) | | | Edge-50 (Obj.↓) | |
|------|------|------|------|------|------|
| | MIS-BA | MCl-HK | MCut-WS | TSP-Gauss. | ATSP-SAT |
| w/o f-t | 65.45 | 6.07 | 795.63 | 24.31 | 5.30 |
| w/ f-t | **70.01** | **6.42** | **839.47** | **24.07** | **2.31** |
| Mode | Node-800-1200 (Obj.↑) | | | Edge-200 (Obj.↓) | |
| | MIS-BA | MCl-HK | MCut-WS | TSP-Gauss. | ATSP-SAT |
| w/o f-t | 273.18 | 5.82 | 2711.10 | 51.98 | 25.02 |
| w/ f-t | **291.07** | **6.38** | **3162.11** | **49.24** | **5.10** |

*Table 5.* Objective(time, ms) on in-distribution, comparing M²GenCO with task-specific meta-learners. uni: uniform.

| Method | MIS-RB | MCl-RB | MCut-BA | TSP-uni. | ATSP-uni. |
|--------|--------|--------|---------|----------|-----------|
| DIMES | – | – | – | 6.32(15) | 2.57(33) |
| DIMES ($T = 5$) | – | – | – | 6.25(>1k) | 2.14(>1k) |
| Meta-EGN | 16.40(10) | 13.80(8) | 673.97(9) | – | – |
| Meta-EGN (4×) | 17.33(40) | 16.95(36) | 684.29(38) | – | – |
| M²GenCO | **17.54**(8) | **17.04**(8) | **703.30**(8) | **5.73**(7) | **1.64**(6) |

*Table 6.* Effect of gradient normalization (Eq. 3) on meta-pretrain. RB-200-300 for MIS/MCl and BA-200-300 for MCut.

| Setting | MIS (Obj.↑) | MCl (Obj.↑) | MCut (Obj.↑) |
|---------|-------------|-------------|--------------|
| w/o grad norm | 16.678 | 13.322 | 635.296 |
| **w/ grad norm** | **16.732** | **13.480** | **657.190** |

*Table 7.* Cross-task generalization (w/o f-t) and few-shot adaptation (w/ f-t) results on *Minimum Vertex Cover* (unseen task).

| Method | MVC-RB-SMALL | | | MVC-RB-LARGE | | |
|--------|------|------|------|------|------|------|
| | Obj.↓ | Gap↓ | Time↓ | Obj.↓ | Gap↓ | Time↓ |
| Gurobi | 205.76* | 0.00% | 3.34s | 968.23* | 0.00% | 290.23s |
| Meta-EGN (4×) | 218.57 | 6.25% | 0.04s | 1000.40 | 3.33% | 0.20s |
| M²GenCO (w/o f-t) | 209.45 | 1.80% | 0.01s | 979.39 | 1.16% | 0.03s |
| M²GenCO (w/ f-t) | **209.20** | **1.71%** | **0.01s** | **977.30** | **0.94%** | **0.03s** |

*Table 8.* Cross-scale generalization: representative and results (across combinations of train/test scales) on ATSP-HCP (Obj.↓).

| $N_{test}$ \ $N_{train}$ | 50 | 100 | 200 | 500 |
|------|------|------|------|------|
| 50 | 1.36 | 1.81 | 2.41 | **1.34** |
| 100 | **1.25** | 1.28 | 2.23 | 1.44 |
| 200 | 1.30 | 0.98 | **0.93** | 1.31 |
| 500 | 0.78 | 0.69 | 0.68 | **0.68** |

*Table 9.* Results for tuning with sub-optimal supervision on BA-200-300 datasets.

| Tune data | MIS (Obj.↑, Gap↓) | MCl (Obj.↑, Gap↓) |
|-----------|-------------------|-------------------|
| Best-quality | 69.76, 4.15% | 7.24, 3.12% |
| 10% flipped | 69.53, 4.47% | 7.18, 3.97% |

**RQ2: Compared with Multi-task Methods.** M²GenCO outperforms recent unified frameworks GOAL (Drakulic et al., 2025) and UniCO (Pan et al., 2025b) with a mean enhancement of **19.2%** and **94.5%** (over applicable benchmarks respectively) along a speedup of **104.3x** and **12.8x**. Note that GOAL is still appreciable as an effective generalizer for CO, whereas substantial time cost is traded off.

**RQ3: Compared with Diffusion-based Solvers.** Powerful diffusion models like DIFUSCO (Sun & Yang, 2023) and Fast-T2T (Li et al., 2024) fall short generalizing to our OOD benchmarks, with a mean performance drop of **34.4%** and **50.5%** from ours. A mean speedup of **29.6x** and **14.3x** is also reached by our alleviated backbone against theirs, e.g., 6 (v.s. 12) GCN layers and 64 (v.s. 256) hidden dimensions.

**RQ4: Post-inference Improving Techniques.** The analyses above are based on a consistent *greedy* strategy for all neural methods to ensure fair comparison. However, the high efficiency of M²GenCO prompts us to reasonably apply sampling or MCTS techniques to further improve the solution quality. Results show that our method achieves large performance superiority with the post-processors enabled, ***without*** sacrificing the dominant advantage in efficiency.

## 5.2. Supplementary Studies

**1) Ablation Study** (Details in Appendix G.4). **1) On Core Design Components.** Fig. 2 demonstrates the training curves ablating the meta-learning[1] and the diffusion mechanism, and Table 4 gives overview of M²GenCO with and without finetuning. We leave the complete numerical results in Table 19, with 45 experiments confirming the synergy of M²GenCO's multi-task meta-learning, generative modeling, and adaptive finetuning. **2) On the Gradient Normalization.** Results in Table 6 validates the gradient normalization across different tasks alleviates training instability.

**2) Generalization Study** (Details in Appendix G.5). **1) Cross-task scenario.** Results on (new task) MVC in Table 7 strengthen the effectiveness of both zero-shot generalization and few-shot adaptation (only **1.16%/0.94%** gap on large instances) towards new graph tasks. **2) Cross-scale scenario.** Results on ATSP-HCP in Table 8 exemplify its strong cross-scale generalizability. Complete results are provided in Table 23 (TSP 50-500), Table 24 (TSP-1000), and Table 25 (MIS, MCl and MCut) in the Appendix.

**3) Stability Study** (Details in Appendix G.6). **1) Anti-noise robustness.** We simulate real-world flawed data by randomly flipping variables in the reference solutions. Table 9 along the tables in Appendix G.6.2 show only minor performance drop (e.g., TSP-Gauss-100: 34.56 v.s. 34.94), indicating robustness with only sub-optimal supervision. **2) Solving stability.** Results in Table 26 show small execution-wise objective variance (e.g., MIS-BA: 70.04±0.02) over 10 random runs, confirming statistical stability of our solver.

---

[1]Ablation experiments marked as "w/o meta" throughout this paper refer to the setting of standard multi-task learning.

*Table 10.* Effect of inner-loop steps ($t_{in}$). Time: training time per outer loop. RB-200-300 for MIS/MCl and BA-200-300 for MCut.

| Setting | MIS (Obj.↑) | MCl (Obj.↑) | MCut (Obj.↑) | Time |
|---|---|---|---|---|
| $t_{in} = 1$ | **16.732** | 13.480 | **657.190** | **1.29s** |
| $t_{in} = 5$ | 16.248 | 13.364 | 632.530 | 1.97s |
| $t_{in} = 10$ | 16.480 | **13.528** | 572.754 | 2.94s |

*Table 11.* Results on different task-pool compositions. RB-200-300 for MIS/MCl and BA-200-300 for MCut.

| Task Pool | MIS (Obj.↑) | MCl (Obj.↑) | MCut (Obj.↑) |
|---|---|---|---|
| MIS, MCl, MCut | 16.33 | 11.37 | 621.82 |
| MIS, MCl | 16.30 | 12.04 | *412.44* |
| MIS, MCut | 16.24 | *10.30* | 631.13 |
| MCl, MCut | *14.84* | 11.67 | 620.94 |

**4) In-distribution (ID) Results** (Details in Appendix **G.7**). Apart from the efficient cross-distribution adaptation ability, Table **5** compares M$^2$GenCO with former meta-learners on ID scenarios. Full results are presented in Table **33**.

**5) Hyper-parameter Study. i) On Inner/Outer Learning Rate.** We leave the entire searching results in Appendix **G.8**, which demonstrate stable training within our specified ranges and supports our particular choices. **ii) On Inner Meta-Steps.** Table **10** indicates that doing one step inner meta-update suffices for our meta-pretraining, aligning with the observation in Wang & Li (2023). **iii) On Sensitivity of the Task Pool.** We conduct experiments removing each of the task in the pool for node-oriented scenario. Table **11** shows retained tasks remain stable and that adding or removing structurally similar graph-based COPs does not significantly undermine the meta-pretraining process.

**Training Cost** (Details in Appendix **G.9**). M$^2$GenCO requires less than **15 GB** for large node tasks and **8 GB** for dense (A)TSP graphs up to 200 nodes of GPU memory, saving up to **82%** computational resources compared to Li et al. (2023; 2024). Also, a total **48h (mostly <24h) on a single GPU** suffices for both pretrain and finetune for most tasks, reducing a roughly **91%** of training time.

**6) Extended Discussions.** We further discuss the synergy between multi-task joint training and meta-learning on graphs, subtle differences among similar concepts, and the design choice of the single-step inner meta-update in Appendix **H**.

## 6. Conclusion

We propose M$^2$GenCO, first formulating diverse problem types (and distributions) as the tasks for meta-learning to solve COPs on graphs. We instantiate the framework with a tailored diffusion-based training scheme and backbone so that multi-task NCO could also enjoy the expressivity of generative modeling. Further, our proposed multi-distribution

benchmark facilitates comprehensive evaluation of NCO methods. Empirical results show M$^2$GenCO's superior performance and efficiency against existing learning-based SOTA. We detail the discussion of broader impacts and the limitations (along with future work) in Appendix **I** and **J**.

## Acknowledgements

This work was in part supported by NSFC (92370201, 625B2119), and the Fundamental and Interdisciplinary Disciplines Breakthrough Plan of the Ministry of Education of China, JYB2025XDXM411.

## Impact Statement

This paper presents work whose goal is to advance the field of Neural Combinatorial Optimization (NCO). Specifically, it introduces a multi-task meta-learning framework for generative combinatorial optimization that considerably improves cross-problem joint pre-training and cross-distribution generalization and adaptation, while significantly reducing training and inference costs. There are many potential societal consequences of our work, none which we feel must be specifically highlighted here. Nevertheless, a detailed discussion of the broader impact of this work is included in Appendix **I**.

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

# Appendix

# A. Notations and Problem Definitions

## A.1. Main Notations

*Table 12.* Main notations and descriptions throughout this paper.

| Scope | Notation | Description |
|---|---|---|
| **Graph Repr.** | $G$ | The graph instance of a CO problem |
| | $\mathcal{E}$ | The edge set of a graph |
| | $\mathcal{V}$ | The node set of a graph |
| | $\mathcal{A}$ | The adjacency matrix of a graph |
| | $\mathcal{D}$ | The distance (cost) matrix of a graph (for TSP) |
| | $N$ | The number of decision variables for a CO problem |
| **Learning Basics** | $\mathbf{x}^* \in \{0,1\}^N$ | The ground truth solution of a CO problem as supervision |
| | $\mathcal{H} \in [0,1]^N$ | The probability heatmap output by neural model |
| | $\widetilde{\mathbf{x}}_0 \in \{0,1\}^N$ | The solution solved by a neural model (i.e. decoded from $\mathcal{H}$) |
| | $\mathcal{L}(\cdot)$ | A loss function |
| | $f_\theta(\cdot)$ | A neural function parameterized by $\theta$ |
| **Diffusion** | $t$ | The time step for diffusion modeling |
| | $\mathbf{Q}_t$ | The matrix of transition probability for the noising process at step $t$ |
| | $\mathbf{x}_0$ | The initial solution point of of a CO instance, assigned $\mathbf{x}^*$ |
| | $\mathbf{x}_t \in \mathbb{R}^N, t = 0:T$ | The noised solution trajectory with a $T$-step noising process |
| **Meta-learning** | $m$ | The number of task types in the multi-task meta-learning data pool |
| | $k$ | The number of batches sampled for meta updates |
| | $\mathcal{T}_i$ | A task for meta-(pre)training, i.e., a type of CO problem |
| | $\mathcal{T}_{i,j}$ | Task $i$ with specific distribution $j$ in the context of OOD-finetuning |
| | $G_i$ | A batch of graph instances sampled from a certain task distribution $p(\mathcal{T}_i)$ |
| | $\theta^{(n)}$ | The model parameter after $n$-th outer iteration |
| | $\theta_i'^{(n)}$ | The (temporary) adapted parameter from $\theta^{(n)}$ with respect to task $\mathcal{T}_i$ |
| | $\theta_{\text{meta}}$ | The initializer model parameter after multi-task meta-pretraining |
| | $\theta_{\mathcal{T}_i,j}$ | The finetuned model parameter w.r.t. task $\mathcal{T}_i$ with particular distribution $j$ |
| | $\alpha$ | The step-size (learning rate) for inner meta-updates |
| | $\beta$ | The step-size (learning rate) for outer meta-updates |
| | $\gamma$ | The learning rate for finetuning |
| **Backbone** | $L$ | The number of layers for neural networks |
| | $d$ | The hidden dimension of the neural layers |
| | $\mathbf{h}^l_{\{e,v,t\},\{\mathcal{T}_i,\emptyset\}}$ | The {edge, node, time} embedding w.r.t. {task $\mathcal{T}_i$, shared} after the $l$-th layer |
| | $\Theta = \{\theta_0, \theta_1, \theta_2\}$ | The learnable parameters of {embedders, graph convolution, output} layers |
| | $\theta_{i,\{\mathcal{T}_j,\emptyset\}}, i \in \{0,1,2\}$ | The task-specific ($\mathcal{T}_j$) or shared ($\emptyset$) parameters |

## A.2. Problem Definitions

**Maximum Independent Set (MIS).** Given $G$, an *independent set* $I \subseteq \mathcal{V}$ is a subset of nodes such that no two nodes in $I$ are adjacent, i.e., it aims to find $I$ that maximizes $|I|$ s.t., $\forall i, j \in I, (i,j) \notin \mathcal{E}$.

**Maximum Clique (MCl).** Given $G$, a *clique* $K \subseteq \mathcal{V}$ is a subset of nodes such that every pair of nodes in $K$ is adjacent. Mathematically, it aims to find $K$ that maximizes $|K|$ s.t., $\forall i, j \in K, (i,j) \in \mathcal{E}$.

**Maximum Cut (MCut).** Given $G$, a *cut* $C = (S, \overline{S})$ is a partition of $\mathcal{V}$ into two disjoint node sets $S$ and $\overline{S}$, i.e., it aims to find $C$ to maximize $\sum_{i \in S, j \in \overline{S}} \mathcal{A}_{ij}$, where $\mathcal{A}$ is the adjacency matrix of $G$.

**Traveling Salesman Problem (TSP).** Given $G$ and a distance matrix $\mathcal{D}$ where $\mathcal{D}_{ij}$ denotes the cost for edge $(i,j) \in \mathcal{E}$, it aims to find the tour $\tau = (i_1, \cdots, i_N)$ to minimize the cost $\sum_{k=1}^{N-1} \mathcal{D}_{i_k i_{k+1}} + \mathcal{D}_{i_N i_1}$.

**Asymmetric Traveling Salesman Problem (ATSP).** ATSP is a special case of TSP where the distance matrix is not necessarily symmetric, i.e., $\mathcal{D}_{ij} = \mathcal{D}_{ji}$ does not have to hold. We follow Drakulic et al. (2023); Pan et al. (2025b); Kwon et al. (2021); Ye et al. (2024b) to study the *metric* (a.k.a. the *tmat* class) ATSP where the triangle inequality for distances

holds universally, i.e., $\mathcal{D}_{ij} + \mathcal{D}_{jk} \geq \mathcal{D}_{ik}, \forall i, j, k \in \mathcal{V}$.

**Minimum Vertex Cover (MVC).** Given $G$, a *vertex cover* $C \subseteq \mathcal{V}$ is a subset of nodes such that for every edge $(i, j) \in \mathcal{E}$, at least one of $i$ or $j$ is in $C$. Mathematically, it aims to find $C$ that minimizes $|C|$ s.t., $\forall (i, j) \in \mathcal{E}, i \in C$ or $j \in C$.

# B. Algorithm 1: Overall Pipeline of M$^2$GenCO

We have streamlined the complete process of the meta-pretraining, OOD finetuning, and problem solving of our proposed M$^2$GenCO framework in Algorithm 1 below, incorporating both high-level training paradigm and specific model updates, data manipulations as well as the diffusion mechanism as a whole, thereby forming a general picture of the comprehensive method innovated in this paper.

---

**Algorithm 1** Pretraining, finetuning, and problem-solving pipeline of M$^2$GenCO through the meta-diffusion learning paradigm, where various COPs with distinctive distributions are viewed as $tasks$.

---

**Require:** A pool of mixed datasets containing $m$ different problem types $\{\mathcal{T}_1, \mathcal{T}_2, \cdots, \mathcal{T}_m\}$ for pretraining, few instances of different target distributions of each task $\{\mathcal{T}_{1,1}, \cdots \mathcal{T}_{1,d_1}, \cdots, \mathcal{T}_{m,d_m}\}$ for finetuning, any heatmap predictor neural network $f_\theta(\cdot, \cdot, \cdot)$ with trainable parameters $\theta$, step size hyper-parameters $\alpha, \beta, \gamma$.

Randomly initialize $\theta^{(0)}$ // **_Meta-pretraining starts_**

**for** $n = 1$ to $N_{\text{outer}} - 1$ **do**

    // **_Outer loop for meta-updates_**

    Sample a $k$-length sequence of task types $\mathcal{T}_1, \cdots \mathcal{T}_k$, each from Uniform$\{\mathcal{T}_1, \cdots, \mathcal{T}_m\}$

    **for** $i = 1$ to $k$ **do**

        // **_Inner loop adaptation with diffusion modeling_**

        Randomly sample a mini-batch of graph instances $G_i \sim p(\mathcal{T}_i)$ with optimal solutions $\mathbf{x}^*$

        Sample a trajectory of noised solution $\mathbf{x}_{0:T} = \mathbf{x}_0, \cdots, \mathbf{x}_T$ with $\mathbf{x}_0 = \mathbf{x}^*$ via $q(\mathbf{x}_t|\mathbf{x}_0)$

        Compute loss $\mathcal{L}_{\mathcal{T}_i}(f_{\theta^{(n)}}) = \mathcal{L}_{\text{BCE}}(f_{\theta^{(n)}}(\mathbf{x}_{t_1}, t_1, G_i), \mathbf{x}^*) + \mathcal{L}_{\text{BCE}}(f_{\theta^{(n)}}(\mathbf{x}_{t_2}, t_2, G_i), \mathbf{x}^*)$

        Evaluate $\nabla_{\theta^{(n)}} \mathcal{L}_{\mathcal{T}_i}(f_{\theta^{(n)}})$ and obtain $\theta_i'^{(n)} \leftarrow \theta^{(n)} - \alpha \nabla_{\theta^{(n)}} \mathcal{L}_{\mathcal{T}_i}(f_{\theta^{(n)}})$

    **end for**

    Update $\theta$ using each $\theta_i'^{(n)}$ with normalized gradients: $\theta^{(n+1)} \leftarrow \theta^{(n)} - \beta \nabla_{\theta^{(n)}} \left( \sum_{\mathcal{T}_i \sim p(\mathcal{T})} \mathcal{L}_{\mathcal{T}_i}(f_{\theta_i'^{(n)}}) \right)$

**end for** // **_Pretraining ends with_** $\theta_{\text{meta}} = \theta^{(N_{\text{outer}})}$

- - - - - - - - - - - - - - - - - - - - - - - - - - - - - - - - - - - - - - - - - - -

Initialize $\theta \leftarrow \theta_{\text{meta}}$ // **_Finetuning starts_**

**while** not done **do**

    Sample a batch of graphs from certain distribution $G \sim p(\mathcal{T}_{i,j})$ {task type $i$ with distribution $j$}

    Finetune $\theta \leftarrow \theta - \gamma \nabla \mathcal{L}_{\mathcal{T}_{i,j}}(f_\theta)$

**end while** // **_Finetuning ends with_** $\theta_{\mathcal{T}_{i,j}}$

- - - - - - - - - - - - - - - - - - - - - - - - - - - - - - - - - - - - - - - - - - -

Initialize $\theta \leftarrow \theta_{\mathcal{T}_{i,j}}$ // **_Solving starts_**

For each test instance $G \sim p(\mathcal{T}_{i,j})$, uniformly sample a noised vector $\mathbf{x}_T$

Model inference: $p_\theta(\mathbf{x}_0|G) \leftarrow f_\theta(\mathbf{x}_T, T, G)$

**for** $n = 1$ to $N_\tau - 1$ **do**

    Sample $\mathbf{x}_0 \sim p_\theta(\mathbf{x}_0|G)$

    $\mathbf{x}_{\tau_n} \leftarrow p(\mathbf{x}_0)\overline{\mathbf{Q}}_{\tau_n}$ // **_Add noise to time-step_** $\tau_n$

    $p_\theta(\mathbf{x}_0|G) \leftarrow f_\theta(\mathbf{x}_{\tau_n}, \tau_n, G)$ // **_Repeat inference with_** $\tau_n$

**end for**

Acquire probability heatmap: $\mathcal{H} \in [0,1]^N \leftarrow \text{Softmax}(\widetilde{\mathbf{x}}_0), \widetilde{\mathbf{x}}_0 \sim p_\theta(\mathbf{x}_0|G)$

**Output:** $\mathbf{x} \in \{0,1\}^N \leftarrow \text{Decoder}(\mathcal{H})$ // **_Solving ends with the final solution_**

---

# C. Extended Discussion of Related Works

Extending upon the summary presented in Sec. 2, we now offer a more in-depth exploration of the related works for (neural) solving of (COPs), following the well-structured taxonomy adopted in Ma et al. (2025a) with minor modifications. The aim is to present a more comprehensive literature review and provide deeper insights into the various method categories within

neural combinatorial optimization field.

**Traditional Learning-free Solvers.** These methods rely on linear (integer) programming algorithms to find exact or near-optimal solutions for COPs. They commonly employ techniques such as branch and bound, cutting planes, and meta-heuristics. Notable examples include globally leading large-scale optimizers like CPLEX (Studio, 2020) and Gurobi (Gurobi Optimization, 2023). For specific tasks, algorithms such as Concorde (Applegate et al., 2006), LKH (Helsgaun, 2017), and GA-EAX (Nagata & Kobayashi, 2013) for the TSP, and KaMIS (Lamm et al., 2016) for the MIS problem, are renowned for their efficiency and solution quality. While these traditional solvers have demonstrated strong performance on their respective problems, they come at the expense of substantial expert knowledge and a large amount of solving time.

**Neural Local Constructive Methods.** Representative methods of this kind, such as AM (Kool et al., 2018), Sym-NCO (Kim et al., 2022), MatNet (Kwon et al., 2021), POMO (Kwon et al., 2020) and BQ-NCO (Drakulic et al., 2023), etc., are typically trained to predict the next node to select or visit based on the current state, rather than determining whether the next nodes belong to the solution set. This process continues until a complete solution is constructed, thus especially suitable for small-to-medium-scale routing problems like TSP and CVRP. However, their autoregressive-style (AR) decision-making process is typically computationally expensive (e.g., with costly RL rollouts), hindering their scalability for larger cases. As a result, hardly can they be directly applied for the node-oriented problems.

**Neural Global Predictive Methods.** Approaches like GCN (Joshi et al., 2019), DIFUSCO (Sun & Yang, 2023), UTSP (Min et al., 2023), T2TCO (Li et al., 2023; 2024), DIMES (Qiu et al., 2022), MaskCO (Chen et al., 2026), and DiffUCO (Sanokowski et al., 2024) generally use neural networks to globally predict the likelihood of variables being selected and decode heatmaps to obtain solutions for certain tasks. After numerous epochs of gradient descent to approximate solution distributions, they often achieve advanced solving quality on specific COPs. It is observed that nearly all these methods perform convolutions on nodes, which requires initial node features (e.g., node coordinates) in the Euclidean 2D-TSP. Consequently, these GNNs have difficulty applying to ATSP described by arbitrary (abstract) distance matrices yet without node coordinates.

**Machine Learning Enhanced Heuristics.** These works concentrate on devising or enhancing heuristics by incorporating machine learning components. They either adjust some parameters of traditional solvers or design heuristics under neural guidance. For example, VSRLKH (Zheng et al., 2021) and NeuroLKH (Xin et al., 2021) combine LKH with reinforcement learning (RL) and supervised learning (SL) respectively. GNNGLS (Hudson et al., 2022) and NeuralGLS (Sui et al., 2023) integrate GNNs and Guided Local Search (GLS) for TSP solving. Similar studies (da Costa et al., 2020; Sui et al., 2021; Ma et al., 2023) solve COPs by neurally defining 2-opt, 3-opt, and k-opt heuristics in a data-driven manner. Works like Kool et al. (2022); Ye et al. (2024a); Kim et al. (2025) also explore the combination of ML techniques with heuristics based on dynamic programming, ant colony, probabilistic search, graph embedding, etc. A common shortcoming of these methods is their heavy reliance on established solvers (e.g., LKH (Helsgaun, 2017)) and lack of universality, necessitating specialized neural techniques (e.g., RL) to achieve satisfactory synergy with the tailored learning-free heuristics. Yan et al. (2026) designs a plug-in differentiable neural layer that enforces general linear constraints via implicit convex optimization, where a GNN is adopted for initial solution prediction, followed by iterative refinement algorithms.

**Solution Optimization Approaches.** These works focus on optimizing an initial solution, usually through a simple greedy heuristic. NeuRewriter (Chen & Tian, 2019) uses neural networks to learn to select heuristics (e.g., $k$-opt) and iteratively rewrite local parts of the current solution for optimization. NeuOpt (Ma et al., 2023) learns to perform flexible $k$-opt exchanges based on a tailored action factorization method and a customized recurrent dual-stream decoder, and proposes the Guided Infeasible Region Exploration (GIRE) scheme, applying the concept of taboo search to combinatorial optimization. Att-GCRN (Fu et al., 2021) uses the heatmap from the global predictor to guide the Monte Carlo Tree Search (MCTS). More recently, GenSCO (Li et al., 2026) leverages the generative modeling as test-time search operator and achieves significant post-inference performance gains. A major issue with the strong learning-free optimizers like MCTS is that they might conceal the real capability of the neural parts, resulting in leveled-off overall solving performance which is misleading for the progress of NCO.

**Adaptive Expansion Paradigms.** First proposed in COExpander (Ma et al., 2025a), the adaptive expansion paradigm orients at utilizing the intermediate state information and maintaining an adaptively controllable granularity of the decisive predictions produced by the neural networks. Similarly, LwD (Ahn et al., 2020) introduces a deferred Markov Decision Process (MDP) that dynamically controls the agent to either determine or defer vertex inclusion at each step. Extended upon LCP (Kim et al., 2021) and GLOP (Ye et al., 2024b), DualOPT (Zhou et al., 2025) integrates a grid-based partitioning phase and a path-based optimization phase. The nodes and edges within each round of conquering process is determined

by the rough solution from LKH, and only interior determinations are fixed towards subsequent rounds. It is expected to incorporate this paradigm upon our framework to further enhance performance, as it leverages multiple inferences upon intermediate decision states.

**Divide-and-Conquer (D&C) Frameworks.** When it comes to solving large-scale CO problems, the divide-and-conquer paradigm is widely used and has proven to be both feasible and effective. For instance, Fu et al. (2021) trains a lightweight model to predict sub-heatmaps that are later combined for large TSP instances. Luo et al. (2024) proposes a self-improved learning method for better scalability. More recently, GLOP (Ye et al., 2024b) and UDC (Zheng et al., 2024) learn to partition large routing problems into sub-TSPs or sub-VRPs and solve them using local revisers. Kim et al. (2021) introduces a collaborative policy framework that explicitly separates exploration and exploitation. In this paper, our primary focus is on multi-task meta-learning and cross-distribution generalization capabilities. Note that D&C methods can be orthogonally applied on top of specific solvers (e.g., GLOP (Ye et al., 2024b) utilizes MatNet (Kwon et al., 2021) as the "local-reviser" for addressing the ATSP instances, and DualOPT (Zhou et al., 2025) adopts LKH (Helsgaun, 2017) as the conquerer), including our M$^2$GenCO, to tackle larger instances as future work.

**Multi-task NCO Solvers.** In earlier research efforts, works such as Li & Liu (2023); Wang et al. (2024a) focus on providing theoretical analyses aimed at enhancing the solving quality across different instances. However, these studies do not put forward specific solvers for COPs. In the work Wang & Yu (2023), the authors introduce a multi-armed bandit framework for training a neural solver to address various CO problems. MVMoE (Zhou et al., 2024) develops a multi-task Vehicle Routing Problem (VRP) solver that employs a mixture-of-experts approach along with a hierarchical gating mechanism. This solver demonstrate good zero-shot generalization performance across multiple variants of the VRP. The study in Boisvert et al. (2024) proposes a generic representation method by encoding problem constraints into a graph structure and decomposing each constraint into an abstract syntax tree where related variables and constraints are linked through edges. However, the authors themselves recognize limitations in terms of the training time required and the size of the generated graphs. MTNCO (Liu et al., 2024), similar to MVMoE, addresses the issue of cross-problem generalization among different variants of VRPs. It leverages shared underlying attributes and solves these problems simultaneously using a single model through the process of attribute composition. More recently, UNCO (Jiang et al., 2024) has resorted to large language models (LLMs). These LLMs take natural language descriptions to formulate text-attributed instances for different COPs and encode them within the same embedding space. However, both the solving quality and scalability of this method still have significant room for improvement. GOAL (Drakulic et al., 2025) proposes a solution that consists of a single backbone combined with lightweight problem-specific adapters. This setup is capable of solving a variety of COPs. Most recently, GCNCO (Li et al., 2025a) builds on a similar header-encoder-decoder structure. In addition, it enforces the consistency of the optimization trajectories across different problems, promoting the learning of generalizable strategies that correspond to the shared structural elements among different CO problems. UniCO (Pan et al., 2025b) proposes to reduce different COPs to the unified representations of a general matrix-encoded TSP. Yet, it incurs additional computational overhead and poses challenges with tasks for which there lacks well-established transformation algorithms (e.g., MIS to TSP).

**Remarks.** From the retrospect of the works from the NCO community, abundant research directions promise with their respective innovations. As reiterated in the main context, this work is primarily motivated by the seek for a balanced objective among 1) state-of-the-art backbone architecture for in-distribution CO solving, 2) the generic representation and jointly multi-task training, and 3) the important yet unheeded ability of effective adaption towards unseen data distributions within few-shot finetuning. The belief is held firmly that M$^2$GenCO shall cooperate with concurrent NCO approaches in harmony towards better neural backbones, training paradigms, effective and efficient multi-task learning of more categories of COPs with further enhanced solving quality and time efficiency.

## D. Details of the Proposed Multi-distribution Benchmark Datasets

Several studies have contributed to benchmark construction in the discrete optimization field. For instance, Pygm-Tools (Wang et al., 2024b) provide portable access to graph matching benchmarks via Python interfaces, MIPLearn (Xavier et al., 2024) and MIPLib (Huang et al., 2024) focus primarily on mixed-integer programming (MIP) benchmarks, while ML4TSPBench (Li et al., 2025b) is dedicated to benchmarks for the TSP task; Böther et al. (2022) proposes to benchmark the maximum independent set problem, and ML4CO-Bench-101 (Ma et al., 2025b) establishes systematic benchmarking datasets for in-distribution graph-based CO tasks, etc. Our work complements these existing efforts by constructing benchmarks that emphasize diverse graph-based COPs across multiple distributions. Importantly, this benchmark serves as auxiliary infrastructure to rigorously evaluate our core contribution: a multi-task meta-diffusion learner designed for

cross-task and cross-distribution generalization on graph-based COPs.

In this section, we provide a comprehensive introduction to the background, synthetic algorithm with specific settings, and broader real-world implications of the multi-distribution data encompassed in our constructed benchmark datasets. Table 13 lists the meta information of the entire benchmark.

*Table 13.* Our collated 38 datasets (line 1-16 for edge-selection tasks and line 17-38 for node-selection tasks), each consists of a support set (optionally used for few-shot finetuning) and a query set (for evaluation). "SMALL" indicates the number of nodes ranges 200-300, "LARGE" denotes 800-1200.

| ID | Problem | Dataset | Generation Settings | | Support and Query | |
| | | | Parameters | Solver | Support Size | Query Size |
|---|---|---|---|---|---|---|
| 1 | TSP | Gaussian-50 | $\mu_x = 0, \mu_y = 0, \sigma = 1$ | Concorde (Applegate et al., 2006) | 6400 | 1280 |
| 2 | TSP | Gaussian-100 | $\mu_x = 0, \mu_y = 0, \sigma = 1$ | Concorde (Applegate et al., 2006) | 6400 | 1280 |
| 3 | TSP | Gaussian-200 | $\mu_x = 0, \mu_y = 0, \sigma = 1$ | Concorde (Applegate et al., 2006) | 640 | 128 |
| 4 | TSP | Gaussian-500 | $\mu_x = 0, \mu_y = 0, \sigma = 1$ | Concorde (Applegate et al., 2006) | 640 | 128 |
| 5 | TSP | Cluster-50 | $C = 10, \sigma_c = 0.03$ | Concorde (Applegate et al., 2006) | 6400 | 1280 |
| 6 | TSP | Cluster-100 | $C = 20, \sigma_c = 0.03$ | Concorde (Applegate et al., 2006) | 6400 | 1280 |
| 7 | TSP | Cluster-200 | $C = 20, \sigma_c = 0.03$ | Concorde (Applegate et al., 2006) | 640 | 128 |
| 8 | TSP | Cluster-500 | $C = 25, \sigma_c = 0.03$ | Concorde (Applegate et al., 2006) | 640 | 128 |
| 9 | ATSP | HCP-50 | – | – | 12500 | 2500 |
| 10 | ATSP | HCP-100 | – | – | 12500 | 2500 |
| 11 | ATSP | HCP-200 | – | – | 500 | 100 |
| 12 | ATSP | HCP-500 | – | – | 500 | 100 |
| 13 | ATSP | SAT-54 | $N_v = 4, N_c = 6$ | – | 12500 | 2500 |
| 14 | ATSP | SAT-102 | $N_v = 8, N_c = 6$ | – | 12500 | 2500 |
| 15 | ATSP | SAT-200 | $N_v = 12, N_c = 8$ | – | 500 | 100 |
| 16 | ATSP | SAT-507 | $N_v = 19, N_c = 13$ | – | 500 | 100 |
| 17 | MIS | BA-SMALL | $N_d = 10$ | KaMIS (60s) (Lamm et al., 2016) | 2500 | 500 |
| 18 | MIS | HK-SMALL | $N_d = 10, p = 0.3$ | KaMIS (60s) (Lamm et al., 2016) | 2500 | 500 |
| 19 | MIS | WS-SMALL | $N_k = 10, p = 0.3$ | KaMIS (60s) (Lamm et al., 2016) | 2500 | 500 |
| 20 | MIS | BA-LARGE | $N_d = 10$ | KaMIS (360s) (Lamm et al., 2016) | 2500 | 500 |
| 21 | MIS | HK-LARGE | $N_d = 10, p = 0.3$ | KaMIS (360s) (Lamm et al., 2016) | 2500 | 500 |
| 22 | MIS | WS-LARGE | $N_k = 10, p = 0.3$ | KaMIS (360s) (Lamm et al., 2016) | 2500 | 500 |
| 23 | MIS | SATLIB | – | KaMIS (60s) (Lamm et al., 2016) | 39500 | 500 |
| 24 | MCl | BA-SMALL | $N_d = 10$ | Gurobi (60s) (Gurobi Optimization, 2023) | 2500 | 500 |
| 25 | MCl | HK-SMALL | $N_d = 10, p = 0.3$ | Gurobi (60s) (Gurobi Optimization, 2023) | 2500 | 500 |
| 26 | MCl | WS-SMALL | $N_k = 10, p = 0.3$ | Gurobi (60s) (Gurobi Optimization, 2023) | 2500 | 500 |
| 27 | MCl | TWITTER | – | Gurobi (60s) (Gurobi Optimization, 2023) | 778 | 195 |
| 28 | MCl | ER-700-800 | $p = 0.15$ | Gurobi (360s) (Gurobi Optimization, 2023) | 640 | 128 |
| 29 | MCl | BA-LARGE | $N_d = 10$ | Gurobi (360s) (Gurobi Optimization, 2023) | 2500 | 500 |
| 30 | MCl | HK-LARGE | $N_d = 10, p = 0.3$ | Gurobi (360s) (Gurobi Optimization, 2023) | 2500 | 500 |
| 31 | MCl | WS-LARGE | $N_k = 10, p = 0.3$ | Gurobi (360s) (Gurobi Optimization, 2023) | 2500 | 500 |
| 32 | MCut | RB-SMALL | $n \in (20, 25), k \in (5, 12)$ | Gurobi (60s) (Gurobi Optimization, 2023) | 2500 | 500 |
| 33 | MCut | HK-SMALL | $N_d = 10, p = 0.3$ | Gurobi (60s) (Gurobi Optimization, 2023) | 2500 | 500 |
| 34 | MCut | WS-SMALL | $N_k = 10, p = 0.3$ | Gurobi (60s) (Gurobi Optimization, 2023) | 2500 | 500 |
| 35 | MCut | ER-700-800 | $p = 0.15$ | Gurobi (360s) (Gurobi Optimization, 2023) | 640 | 128 |
| 36 | MCut | RB-LARGE | $n \in (40, 55), k \in (20, 25)$ | Gurobi (360s) (Gurobi Optimization, 2023) | 2500 | 500 |
| 37 | MCut | HK-LARGE | $N_d = 10, p = 0.3$ | Gurobi (360s) (Gurobi Optimization, 2023) | 2500 | 500 |
| 38 | MCut | WS-LARGE | $N_k = 10, p = 0.3$ | Gurobi (360s) (Gurobi Optimization, 2023) | 2500 | 500 |

### D.1. TSP

**Uniform Distribution.** The uniform TSP enjoys the widest application for assessing the TSP solvers in the NCO community. Each instance consists of $|\mathcal{V}|$ node coordinates sampled from the unit square $[0, 1]$, and the distance matrix $\mathcal{D}$ can be computed via the Euclidean distance formula in the 2D space.

**Gaussian Distribution.** The Gaussian TSP reflects the real-world scenarios where locations are concentrated around a central hub, such as delivery points around a city warehouse or demand points in densely populated metropolitan areas. Each set of node coordinates is sampled from a two-dimensional Gaussian distribution $\mathcal{N}\big((\mu_x, \mu_y), \sigma^2 I\big)$, where $(\mu_x, \mu_y)$ is the center (average) of the distribution and $\sigma$ controls the spatial spread (standard deviation), and $I$ is the identity matrix.

**Cluster Distribution.** The Cluster TSP simulates scenarios where nodes are organized into distinct spatial regions. Specifically, multiple ($C$) cluster centers are first sampled uniformly within the unit square, i.e., $[0, 1]^2$, and then the nodes

are drawn from Gaussian distributions centering these locations with a predefined cluster-level standard deviation $\sigma_c$ as a hyper-parameter.

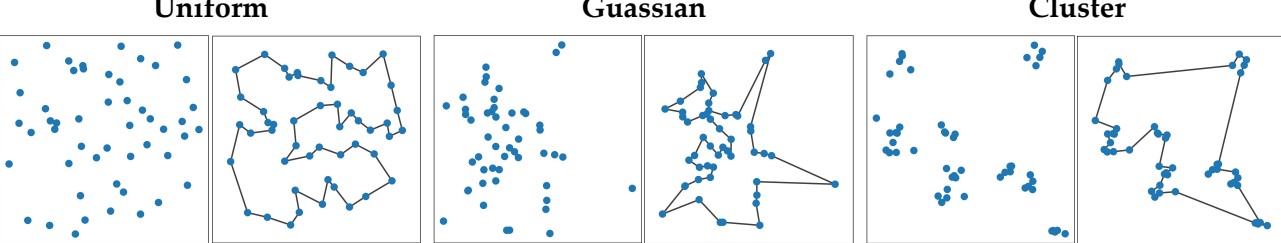

*Figure 3.* Visualizing the differently distributed TSP in the Euclidean plane. Scale: $|\mathcal{V}| = 50$ nodes.

Fig. 3 illustrates the differences among the 3 (symmetric, coordinate-based) TSP instances. Note that the coordinates of the Gaussian- and cluster-distributed TSP instances are normalized within [0,1] before training and testing, to guarantee a consistent evaluation with the uniform datasets.

### D.2. ATSP

**Uniform Distribution.** To our knowledge, MatNet (Kwon et al., 2021) is one of the earliest attempts to study neural solving of ATSP. Subsequent works (Drakulic et al., 2025; Pan et al., 2025b; Lischka et al., 2024; Ma et al., 2025a) (including this paper) follow the protocol to generate uniform ATSP (a.k.a. the "tmat" class ATSP) data with the triangle inequality globally holding, mathematically,

$$\mathcal{D}_{ij} + \mathcal{D}_{jk} \geq \mathcal{D}_{ik}, \forall i, j, k \in \mathcal{V}. \tag{7}$$

The generation is performed by first populating the distance matrix $\mathcal{D}$ with independently uniform values from $[1, M]$, and then repeatedly replace $\mathcal{D}_{ij}$ with $\mathcal{D}_{ik} + \mathcal{D}_{kj}$ wherever $\mathcal{D}_{ij} > \mathcal{D}_{ik} + \mathcal{D}_{kj}$ is detected until no more changes are made. In the end, the diagonal entries of $\mathcal{D}$ are set to zero, and scale $\mathcal{D}$ back to $[0, 1]$ with the same large scaling factor of $M$ (usually set as $10^6$).

Beyond the uniform (tmat) ATSP, we follow UniCO (Pan et al., 2025b), which proposes the new perspective of generating multi-distribution ATSP (generalized TSP) instances through problem reduction among the NP-hard COPs, to include two more ATSP variants, transformed from the Hamiltonian Cycle Problem (HCP) and the 3-Satisfiability (3-SAT) problem. The discrete nature (i.e., with merely binary values in the distance matrix) poses significant challenge for current neural solvers to conquer the HCP- and SAT-distributed ATSP instances with satisfactory quality, therefore appropriately serving as the benchmark datasets to assess the adaptability and generalizability of neural CO solvers.

**HCP Distribution.** Given a graph $G = (\mathcal{V}, \mathcal{E})$, an HCP instance aims to determine whether $G$ contains a Hamiltonian cycle that visits every node $v \in \mathcal{V}$ once and only once. By simply setting

$$\mathcal{D}_{ij} = \begin{cases} 0, & (i,j) \in \mathcal{E} \\ 1, & \text{otherwise} \end{cases}, \tag{8}$$

the problem of finding an Hamiltonian cycle becomes finding a tour of length 0 (also the minimum length) for the TSP instance with distance matrix $\mathcal{D}$. We randomly generate such distance matrices of $|\mathcal{V}| = 50, 100, 200, 500$ nodes for OOD finetune and evaluation of the ATSP models.

**3-SAT Distribution.** Given a 3-SAT instance of $N_v$ variables and $N_c$ clauses, reducing it to an HCP instance which is then regarded as a TSP graph generates a binary distance matrix with $(2N_vN_c + N_c)$ nodes. Mathematically, the transformation procedure can be done through the steps following Pan et al. (2025b)[2]:

*Step 1 (Variable-Clause Graph Construction).* We initialize a graph with $2N_vN_c$ variable-clause nodes, where each variable $x_i$ ($0 \leq i < N_v$) is associated with $2N_c$ nodes indexed from $2N_ci$ to $2N_c(i+1) - 1$. For each variable $x_i$, we establish bidirectional edges[3] between consecutive nodes $(m, m+1)$ and $(m+1, m)$ for $m \in [2N_ci, 2N_c(i+1) - 2]$. The boundary connections differ based on variable position:

---

[2]The transformation algorithm from 3-SAT to HCP is implemented upon the guidance from https://opendsa-server.cs.vt.edu/ODSA/Books/Everything/html/threeSAT_to_hamiltonianCycle.html (MIT license).

[3]The addition of edges described in this section is practically implemented through assigning $\mathcal{D}_{ij} = 1$ in the distance matrix (originally filled with 0) with corresponding node indices.

- For $i < N_v - 1$: Add 4 inter-variable edges $(2N_c i, 2N_c(i+1))$, $(2N_c i, 2N_c(i+2)-1)$, $(2N_c(i+1)-1, 2N_c(i+1))$, and $(2N_c(i+1)-1, 2N_c(i+2)-1)$;

- For $i = N_v - 1$: Add 4 cyclic edges $(2N_c i, 0)$, $(2N_c i, 2N_c - 1)$, $(2N_c(i+1)-1, 0)$, and $(2N_c(i+1)-1, 2N_c - 1)$.

*Step 2 (Clause Node Integration).* We introduce $N_c$ clause nodes indexed from $2N_v N_c$ to $2N_v N_c + N_c - 1$, corresponding to clauses $\{C_j\}_{j=0}^{N_c-1}$. Edge connections are polarity-dependent:

- For $x_i \in C_j$: Add directed edges $(2N_c i + 2j, 2N_v N_c + j)$ and $(2N_v N_c + j, 2N_c i + 2j + 1)$;

- For $\neg x_i \in C_j$: Add reverse edges $(2N_v N_c + j, 2N_c i + 2j)$ and $(2N_c i + 2j + 1, 2N_v N_c + j)$.

This polynomial-time reduction yields a graph $G$ with $2N_v N_c + N_c$ nodes, where the existence of a Hamiltonian path in $G$ is equivalent to the satisfiability of the original 3SAT instance. Denoting the distance matrix (i.e., the adjacency matrix with all connections weighted 1 and otherwise 0) as $\mathcal{D}$, we generate such distance matrices with $N_v$ and $N_c$ specified in Table **13** to form OOD ATSP datasets with $|\mathcal{V}| = 54, 102, 200, 507$, respectively. Note that the computational time overhead of this transformation procedure can be neglected upon our empirical observation.

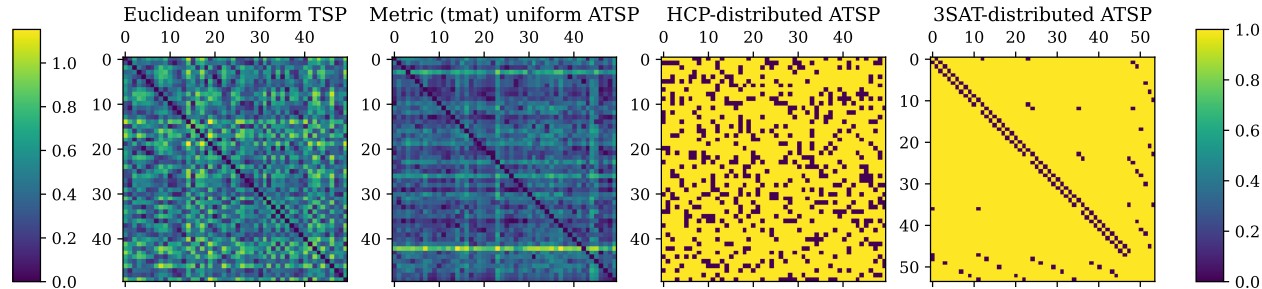

*Figure 4.* Visualizing the distance matrices for ATSP instances with different distributions and comparing with the conventionally studied 2-D uniform TSP instance. Scale: $|\mathcal{V}| = 50$ nodes.

**Remarks.** The incorporation of the binary ATSPs in our benchmark enriches the datasets for generalization evaluation of ATSP, and moreover, demonstrate the potential of transforming discrete decision problems into an optimization task like TSP. Note that for both HCP and SAT distributions of ATSP, our generation algorithms guarantee the existence of a 0-length tour as the reference solution. Thus, the metric of optimality gap does not pertain here, and we use only the objective value (tour length, the lower the better) to indicate the solving performance on these instances.

### D.3. MIS, MCl and MCut

**RB Graph Distribution.** RB Graph is generated by RB-Model (Xu et al., 2005), commonly used to simulate the interactions between multiple groups in social networks. Three main generation parameters: 1) $n$: the number of cliques, which are groups of nodes that are fully interconnected; 2) $k$: the number of nodes within the clique are specified. 3) $p$: the parameter that controls the level of interconnectivity between different cliques. Following previous works (Sanokowski et al., 2024; Li et al., 2024), the range of values for $p$ is $p \in (0.3.1.0)$.

**BA Graph Distribution.** BA Graph (Barabási & Albert, 1999) is a scale-free graph where nodes are added incrementally, linking preferentially to highly connected nodes. At each step, a new node is added to the graph, which connects to at most $N_d$ existing nodes. In practice, we set $N_d$ as 10 in our benchmark.

**HK Graph Distribution.** Holme-Kim (HK) Graph (Holme & Kim, 2002), proposed by Holme and Kim in 2002, is an extension of the BA graph, aiming at generating scale-free networks with high clustering coefficients. It begins with a small fully connected network and grows by adding new nodes that preferentially attach to $N_d$ existing nodes with higher degrees. Additionally, with a probability $p$, the new node also connects to a neighbor of the target node, promoting the formation of triangles. This mechanism results in a network with a power-law degree distribution and a high clustering coefficient, making it suitable for modeling real-world scenarios such as the social networks as well as the biological systems, where both scale-free and high clustering properties are possessed.

**WS Graph Distribution.** Watts-Strogatz (WS) Graph (Watts & Strogatz, 1998), introduced by Watts and Strogatz in 1998, is a small-world network model designed to capture the balance between high clustering and short path lengths observed in

many real-world networks. It starts with a regular ring lattice where each node is connected to its top-$N_k$ nearest neighbors. Then, with a probability $p$, each edge is rewired to a random node, maintaining network connectivity. This process creates a network with high clustering due to the initial regular structure and short average path lengths due to the random rewiring, making it ideal for modeling social and biological networks that exhibit small-world properties.

**ER Graph Distribution.** ER Graph (Erdős & Rényi, 1960) is randomly generated with each edge maintaining a fixed probability of being present or absent, independently of the other edges. We follow DIFUSCO (Sun & Yang, 2023), Fast-T2T (Li et al., 2024) to set the probability $p$ as 0.15, with the number of nodes ranging from 700 to 800.

**TWITTER.** Twitter Graph dataset (Jure, 2014) is part of the Stanford Network Analysis Project collection, which represents a snapshot of the social network formed by Twitter users and their interactions. We download the original datasets from TUDataset (Morris et al., 2020)[4], as it is the widely used dataset for generalization evaluation in previous literature (Wang & Li, 2023; Sanokowski et al., 2023; 2024) that mainly study the node-based tasks.

**SATLIB.** SATLIB[5] is a classic dataset for SAT problems, and we follow previous works (Qiu et al., 2022; Sun & Yang, 2023; Li et al., 2024) to transform SAT into MIS, which demonstrates considerable density regarding the MIS graphs.

Fig. 5 visualizes the different graph structures for node-oriented tasks, providing an intuitive sense of diversity of our generated and organized benchmark regarding the node-oriented COPs.

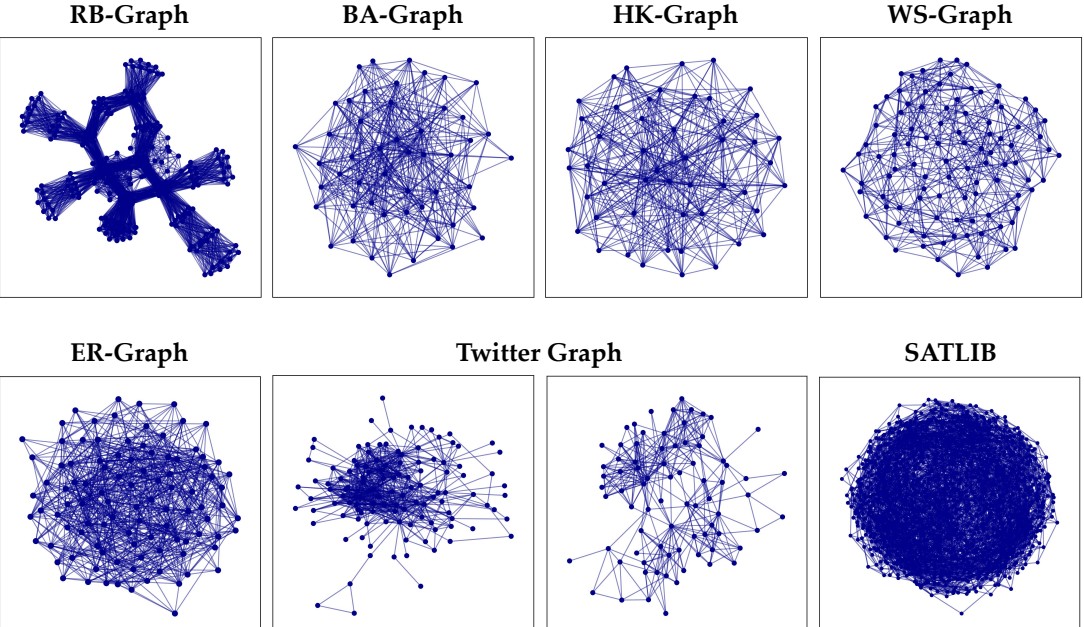

*Figure 5.* Visualizing the structure of different graph distributions. The two sub-images in the middle of the second line are both instances from the TWITTER dataset, indicating its widely varying nature.

# E. Details of Model Architecture

The high-level architecture of the graph neural network employed in M²GenCO has been delineated in Sec. **4.4**.

Below, we provide the mathematical details of the backbone, especially the improved message-passing scheme for the graph convolutions.

---

[4] https://chrsmrrs.github.io/datasets/docs/datasets/
[5] https://www.cs.ubc.ca/~hoos/SATLIB/Benchmarks/SAT/CBS/descr_CBS.html

### E.1. Embedding Layers

Recall that the inputs (noised solution vector $\mathbf{x}_t$, time step $t$, and graph $G$) are first projected to hidden embeddings by the task-specifically parameterized $\mathbf{Embed}_{\theta_0, \mathcal{T}_i}$ with respect to task $\mathcal{T}_i$:

$$\{\mathbf{h}_{v,\mathcal{T}_i}^0, \mathbf{h}_{e,\mathcal{T}_i}^0, \mathbf{h}_{t,\mathcal{T}_i}^0\} = \mathbf{Embed}_{\theta_0, \mathcal{T}_i}(\mathbf{x}_t, t, G_i), \tag{9}$$

where $\mathbf{h}_v$, $\mathbf{h}_e$ and $\mathbf{h}_t$ are the embeddings for the node, edge and the time step, respectively. Specifically, the graph-related features $G_i$ consists of both node features ($x \in \mathbb{R}^{|\mathcal{V}| \times 2}$) and edge features ($e \in \mathbb{R}^{|\mathcal{E}|}$). For TSP with Euclidean distances, $x$ is naturally given by the 2-D node coordinates, while for ATSP and the node-oriented tasks where no inherent node features are applicable, $x$ is randomly initialized from $[0, 1]$ for either dimension. Similarly, the input edge feature $e$ for TSP and ATSP is the pair-wise distance value and for the node-oriented problems it encodes the adjacency of graphs with binary values. Subsequently, we follow Joshi et al. (2019); Sun & Yang (2023); Li et al. (2023); etc., to compute the sinusoidal features of each input element respectively, which serves as a positional encoding for node ($\widetilde{x}$), edge ($\widetilde{e}$) and timestep ($\widetilde{t}$):

$$\widetilde{x}_i = \mathrm{concat}(x_{i,0}, x_{i,1}), \tag{10}$$

$$\widetilde{x}_{i,j} = \mathrm{concat}\left(\sin\frac{x_{i,j}}{T^{\frac{0}{d}}}, \cos\frac{x_{i,j}}{T^{\frac{0}{d}}}, \sin\frac{x_{i,j}}{T^{\frac{2}{d}}}, \cos\frac{x_{i,j}}{T^{\frac{2}{d}}}, \ldots, \sin\frac{x_{i,j}}{T^{\frac{d}{d}}}, \cos\frac{x_{i,j}}{T^{\frac{d}{d}}}\right), \tag{11}$$

$$\widetilde{e}_i = \mathrm{concat}\left(\sin\frac{e_i}{T^{\frac{0}{d}}}, \cos\frac{e_i}{T^{\frac{0}{d}}}, \sin\frac{e_i}{T^{\frac{2}{d}}}, \cos\frac{e_i}{T^{\frac{2}{d}}}, \ldots, \sin\frac{e_i}{T^{\frac{d}{d}}}, \cos\frac{e_i}{T^{\frac{d}{d}}}\right), \tag{12}$$

$$\widetilde{t} = \mathrm{concat}\left(\sin\frac{t}{T^{\frac{0}{d}}}, \cos\frac{t}{T^{\frac{0}{d}}}, \sin\frac{t}{T^{\frac{2}{d}}}, \cos\frac{t}{T^{\frac{2}{d}}}, \ldots, \sin\frac{t}{T^{\frac{d}{d}}}, \cos\frac{t}{T^{\frac{d}{d}}}\right), \tag{13}$$

where $d$ is the embedding dimension, $T$ is a large number (10000 by default), $\mathrm{concat}(\cdot)$ denotes concatenation. The denoising timestep $t$ is required by the diffusion model with $t \in \{\tau_1, \ldots, \tau_M\}$. Note that $\widetilde{x}_{i,j}$ and $\widetilde{e}_i$ are the respective entries of the embedded vectors $\widetilde{x}$ and $\widetilde{e}$. Then, the input features of the graph convolution layer is projected by the linear transformations:

$$\mathbf{h}_v^0 = W_1^0 \widetilde{x}, \tag{14}$$

$$\mathbf{h}_e^0 = W_2^0 \widetilde{e}, \tag{15}$$

$$\mathbf{h}_t^0 = W_4^0 (\mathrm{ReLU}(W_3^0 \widetilde{t})), \tag{16}$$

where $W_1^0, W_2^0, W_3^0, W_4^0$ are the learnable parameters, constituting $\theta_0$ for the $\mathbf{Embed}(\cdot, \cdot, \cdot)$ for each individual task $\mathcal{T}_i$ which gets optimized separately during the multi-task training.

### E.2. Graph Convolutional Layers with Intermediate Node Embedding

Once embedded, the features are passed through $L_{\mathrm{shared}}$ shared and $L_{\mathcal{T}_i}$ separate GCN layers for each task $i$, formally,

$$\{\mathbf{h}_v^{l+1}, \mathbf{h}_e^{l+1}, \mathbf{h}_t^{l+1}\} = \mathbf{Conv}_{\theta_{1,\mathrm{shared}}}^l(\mathbf{h}_{v,\mathcal{T}_i}^l, \mathbf{h}_{e,\mathcal{T}_i}^l, \mathbf{h}_{t,\mathcal{T}_i}^l) \text{ for } l = 0 \text{ to } L_{\mathrm{shared}} - 1, \forall i, \tag{17}$$

$$\{\mathbf{h}_{v,\mathcal{T}_i}^{l+1}, \mathbf{h}_{e,\mathcal{T}_i}^{l+1}, \mathbf{h}_{t,\mathcal{T}_i}^{l+1}\} = \mathbf{Conv}_{\theta_{1,\mathcal{T}_i}}^l(\mathbf{h}_v^l, \mathbf{h}_e^l, \mathbf{h}_t^l) \text{ for } l = L_{\mathrm{shared}} \text{ to } L_{\mathrm{shared}} + L_{\mathcal{T}_i} - 1. \tag{18}$$

Here, we expand the convolution layers, i.e., $\{\mathbf{h}_v^{l+1}, \mathbf{h}_e^{l+1}, \mathbf{h}_t^{l+1}\} = \mathbf{Conv}_{\theta_1}^l(\mathbf{h}_v^l, \mathbf{h}_e^l, \mathbf{h}_t^l)$. The cross-layer convolution operation is originally formulated in Joshi et al. (2019) as:

$$\mathbf{h}_{v_i}^{l+1} = \mathbf{h}_{v_i}^l + \mathrm{ReLU}\left(\mathrm{BN}\left(W_5^l \mathbf{h}_{v_i}^l + \sum_{j \sim i} \eta_{ij}^l \odot W_6^l \mathbf{h}_{v_j}^l\right)\right), \tag{19}$$

$$\mathbf{h}_{e_{ij}}^{l+1} = \mathbf{h}_{e_{ij}}^l + \mathrm{ReLU}\left(\mathrm{BN}\left(W_7^l \mathbf{h}_{e_{ij}}^l + W_8^l \mathbf{h}_{v_i}^l + W_9^l \mathbf{h}_{v_j}^l\right)\right), \tag{20}$$

$$\eta_{ij}^l = \frac{\sigma(\mathbf{h}_{e_{ij}}^l)}{\sum_{j' \sim i} \sigma(e_{ij'}^l) + \epsilon}, \tag{21}$$

where $\mathbf{h}_{v_i}^l$ and $\mathbf{h}_{e_{ij}}^l$ denote the node feature vector and edge feature vector at layer $l$, $W_i \in \mathbb{R}^{d \times d}$ denote the model weights, $\eta_{ij}^l$ denotes the dense attention map. The convolution operation integrates the edge feature to accommodate the significance

of edges in routing problems. However, with the research scope for TSP expanding from uniform Euclidean 2-D to arbitrary ATSP instances, a significant performance drop has been observed as demonstrated in Pan et al. (2025b). The authors there resort to a bipartite scheme to represent the asymmetric graphs and maintain dual node features throughout the convolutions, which is effective though less efficient and is hardly generalized to problems other than (A)TSP. To this end, we propose a novel message-passing scheme to address the cases where initial node embeddings are lacking for the edge-model of our multi-task meta-learner. In each convolution layer, the informative edge features are first projected bi-directionally and combined with the initial (random) node feature, forming a new node embedding:

$$\widetilde{\mathbf{h}}_{v_i}^l = W_8^l \Big( \text{concat} \left[ W_5(\mathbf{h}_{v_i}^l), W_6(\mathbf{h}_e^l)_{ij}, W_7(\mathbf{h}_e^l)_{ji} \right] \Big). \tag{22}$$

Next, the informed node feature $\widetilde{\mathbf{h}}_v$ is used to update the edge features as with Eq. **20**. Note that the intermediate edge embedding $\widetilde{\mathbf{h}}_{e_{ij}}^l$ will later directly serves as the output edge feature of the $l$-th layer, as in Eq. **25**.

$$\widetilde{\mathbf{h}}_{e_{ij}}^l = \mathbf{h}_{e_{ij}}^l + \text{ReLU}\left( \text{BN}\left( W_9^l \mathbf{h}_{e_{ij}}^l + W_{10}^l \widetilde{\mathbf{h}}_{v_i}^l + W_{11}^l \widetilde{\mathbf{h}}_{v_j}^l \right) \right). \tag{23}$$

Finally, the node features are updated with the new node and edge embeddings with the same gating mechanism as in Eq. **21**:

$$\mathbf{h}_{v_i}^{l+1} = \mathbf{h}_{v_i}^l + \text{ReLU}\left( \text{BN}\left( W_{12}^l \widetilde{\mathbf{h}}_{v_i}^l + \sum_{j \sim i} \eta_{ij}^l \odot W_{13}^l \widetilde{\mathbf{h}}_{v_j}^l \right) \right) ; \ \eta_{ij}^l = \frac{\sigma(\widetilde{\mathbf{h}}_{e_{ij}}^l)}{\sum_{j' \sim i} \widetilde{\mathbf{h}}_{e_{ij'}}^l + \epsilon}, \tag{24}$$

$$\mathbf{h}_{e_{ij}}^{l+1} \leftarrow \widetilde{\mathbf{h}}_{e_{ij}}^l. \tag{25}$$

To adapt for the diffusion time-steps, we follow Sun & Yang (2023); Li et al. (2023) to add the layer-wise time embedding $W_{\text{time}}^l(\text{ReLU}(t^0))$ to $\mathbf{h}_e^l$ or $\mathbf{h}_v^l$ for edge- or node- selection tasks, respectively. Thus far, the per-layer parameters for $\textbf{Conv}_{\theta_1}(\cdot, \cdot, \cdot)$ have been introduced, i.e., $W_i^l \in \theta_1$ for both shared and task-specific graph convolutional layers.

### E.3. Output Layers

Eventually, a 2-D convolution layer is employed to transform the output embeddings from the $L$-th (final, $L = L_{\text{shared}} + L_{\mathcal{T}_i}$) graph convolution (preveniently normalized via $\text{Norm}(\cdot)$) into binary classification logits, and the $\text{Softmax}(\cdot)$ is utilized to produce probabilistic heatmaps:

$$\mathcal{H}_{node} = \textbf{Out}_{\theta_2, \mathcal{T}_i}(\mathbf{h}_{v, \mathcal{T}_i}^L) = \text{Softmax}(\text{Conv2d}_{\theta_2, \mathcal{T}_i}(\text{Norm}(\mathbf{h}_{v, \mathcal{T}_i}^L)), \tag{26}$$

$$\mathcal{H}_{edge} = \textbf{Out}_{\theta_2, \mathcal{T}_i}(\mathbf{h}_{e, \mathcal{T}_i}^L) = \text{Softmax}(\text{Conv2d}_{\theta_2, \mathcal{T}_i}(\text{Norm}(\mathbf{h}_{e, \mathcal{T}_i}^L)). \tag{27}$$

## F. Details of Experimental Settings

### F.1. Hardware

The training for large scale instances are conducted on a single NVIDIA H100 80GB GPU with an Intel(R) Xeon(R) Platinum 8558 96-Core Processor CPU. The training for small-to-medium scale instances and all the evaluations (including the large parts) are conducted on a single NVIDIA RTX3090 24GB GPU with AMD 3970X 32-Core CPU to ensure comparative consistency.

### F.2. Hyper-parameters

**Backbone.** For all models trained under our M$^2$GenCO framework in this paper, they share the same neural backbone as introduced in Appendix **E**. Particularly, $L_{\text{shered}} = 3$ shared followed by $L_{\mathcal{T}_i} = 3$ task-specific GCN layers are employed to constitute the main body of the encoder. The hidden dimension are set to $d = 64$ wherever applicable in the model. Unless otherwise stated, the internal design choices and hyper-parameters for the diffusion mechanism are kept consistent with Li et al. (2024).

**Training Details.** In general, we pretrain two models for both the node and edge selection tasks, where the multi-task meta-learning mechanism is enabled within the respective task pool. Subsequent finetuning is conducted from the corresponding pretrained weights towards new data distributions. The AdamW (Loshchilov & Hutter, 2018) optimizer is

*Table 14.* Amount of pre-training data used for each task, following the standardized and publicly released datasets from Ma et al. (2025a).

| Task | Benchmark | Data Size |
|------|-----------|-----------|
| MIS | RB-SMALL | 64k |
| MIS | RB-LARGE | 6.4k |
| MCl | RB-SMALL | 64k |
| MCl | RB-LARGE | 6.4k |
| MCut | BA-SMALL | 128k |
| MCut | BA-LARGE | 128k |
| TSP | uniform-50 | 1.28M |
| TSP | uniform-100 | 1.28M |
| TSP | uniform-500 | 64k |
| ATSP | uniform-50 | 640k |
| ATSP | uniform-100 | 128k |
| ATSP | uniform-200 | 32k |
| ATSP | uniform-500 | 6.4k |

globally adopted with a cosine-decayed LR scheduler and a weight decay of 0.0001 (for the outer loop) for all the model training. Following Sun & Yang (2023); Li et al. (2023); Ma et al. (2025a), the models are trained in a manner of curriculum learning, i.e., models trained for large-scale tasks can be initialized from the weights pretrained on smaller data. The settings that vary with the task scales are listed in Table **15**. Note that for meta-learning, the training duration is usually specified by $N_{\text{outer}}$ which is independent of a fixed number of rounds traversing a certain training set. To keep a notational consistency, we use $N_{\text{f-t}}$ to refer to the global optimizing steps (i.e., the total number of batches seen by the model), so the number of *epochs* can be calculated by $N_{\text{epoch}} = N_{\text{f-t}} \cdot B/S$, where $B$ is the batch size for tuning and $S$ is the data size specified in Table 13 from the column of "SUPPORT SIZE".

*Table 15.* Training settings of models for different task scales. SMALL: 200-300 nodes; LARGE: 800-1200 nodes.

| **Meta-Pretrain Settings** | TSP/ATSP | | | | MIS/MCl/MCut | | | |
|---|---|---|---|---|---|---|---|---|
| | 50 | 100 | 200 | 500 | SMALL | LARGE | | |
| Max outer iterations $N_{\text{outer}}$ | 128k | 64k | – | 32k | 128k | 12.8k | | |
| Max inner iterations $N_{\text{inner}}$ | 1 | 1 | – | 1 | 1 | 1 | | |
| Inner learning rate $\alpha$ | 5e-5 | 5e-5 | – | 5e-5 | 5e-5 | 5e-5 | | |
| Outer learning rate $\beta$ | 2e-4 | 2e-4 | – | 1e-4 | 5e-4 | 5e-4 | | |
| Length of task sequence $k$ | 4 | 4 | – | 2 | 4 | 2 | | |
| Batch size | 4 | 4 | – | 2 | 4 | 2 | | |
| **Finetune Settings** | TSP/ATSP | | | | MIS/MCl | | MCut | |
| | 50 | 100 | 200 | 500 | SMALL | LARGE | SMALL | LARGE |
| Max iterations $N_{\text{f-t}}$ | 64k | 64k | 32k | 6.4k | 64k | 12.8k | 12.8k | 12.8k |
| Finetuning learning rate $\gamma$ | 5e-5 | 5e-5 | 2e-5 | 1e-5 | 5e-5 | 2e-5 | 5e-5 | 2e-5 |
| Batch size | 32 | 12 | 8 | 8 | 32 | 16 | 32 | 16 |

**Graph Sparsification.** In this work, all the models for node-oriented tasks are learned with the graph instances represented in sparse format (i.e., $O(|\mathcal{E}|)$ edge features indicating the connectivity rather than $O(|\mathcal{V}^2|)$ adjacency matrix). For the edge-oriented tasks, despite the prevalence of the k-nearest neighbor (KNN) technique to reduce computational overhead for large instances, we refrain from adopting KNN even for TSP/ATSP-500. On the one hand, we note an insightful experiment conducted by the authors of Ma et al. (2025a), revealing that a sparsification factor as large as $K = 250$ fails to maintain the optimality of the solution for ATSP500, whereas $K = 50$ has been more than sufficient to cover all optimal edges for TSP500 (which also indicates the difficulty level from the widely studied uniform TSP to the under-explored ATSP cases). Note that M$^2$GenCO needs to conduct simultaneous pretraining on both TSP and ATSP instances, we thus choose to remain the dense matrix input for the pair-wise distance information, enforcing representational consistency between ATSP and TSP instances meanwhile not compromising the quality of supervision for ATSP. On the other hand, we empirically validated this design choice would not induce unaffordable memory consumption on account of our light-weight architecture, as further demonstrated in Appendix **G.9**.

## F.3. Evaluation Settings

**Overview.** The governing principle of all evaluations in this paper is to enforce a serialized solving pipeline across various solving methods. Specifically, the batch size is strictly set as 1 for neural solvers at the testing phase, and all reference solutions are obtained in the single thread model (e.g., for Gurobi and LKH, etc.), which guarantees a fair and transparent comparison of the solving efficiency. In terms of the solving effectiveness, we disabled any means of multiple sampling tricks that may involves additional randomness or solution diversity, e.g., Meta-EGN (Wang & Li, 2023) computes 8 solutions sequentially for one instance and reports the best, RL4CO (Berto et al., 2023) methods resort to multiple starting point to enhance performance, MatNet (Kwon et al., 2021) augments the solving quality via parallelized random inferences, etc. We ensure the reported results in our experiments reflect a fair comparison under the simplest greedy policy. Post-processing techniques such as 2-OPT could be optionally performed with a strict precondition of not demising the advantage on our solving efficiency.

**Specific Settings for Baseline Methods.** For exact or heuristic solvers, the solving time limits are set as follows: Gurobi (Gurobi Optimization, 2023) or KaMIS (Lamm et al., 2016) is allocated 60 seconds for small-scale node-oriented tasks and 360 seconds for large-scale instances. For TSP/ATSP solving with LKH (Helsgaun, 2017), each instance is executed with 1 run and 500/1000 trials respectively.

For the compared neural solvers, all experiments are conducted using the official pretrained weights (where available) or retrained by us according to the original papers. Other settings are specified as follows:

- DIFUSCO (Sun & Yang, 2023)[6]. We use its default test-time inference steps ($I_s$).

- Fast-T2T (Li et al., 2024)[7]. We set $I_s = 20$ and others by default.

- DiffUCO (Sanokowski et al., 2024)[8]. We follow (Ma et al., 2025a) to set $F = 1$ (diffusion steps equal to training) and $S = 1$ sampling for fair comparison. Other settings remain default.

- Meta-EGN (Wang & Li, 2023)[9]. We employ the "fast" mode from its original paper, using 1 random seed per instance without fine-tuning. Other settings remain default.

- DIMES (Qiu et al., 2022)[10]. We use the dense version without active search (AS) in main experiments for TSP; comparative experiments with $T_{AS} = 5$ are conducted in ID evaluations (Appendix **G.7**). Other settings remain default. For ATSP, we adopt an extension of DIMES in (Pan et al., 2025b) that applies DIMES to ATSP instances beyond 2D coordinate-based TSP.

- UniCO (Pan et al., 2025b)[11]. We use the MatPOENet with single-task weights trained on uniform ATSP data to assess generalization, and no parallel augmentation is enabled for fair comparison. Other settings remain default.

- GCN4CO (Joshi et al., 2019) and GNNGLS (Hudson et al., 2022)[12]. They generate neural heatmaps through supervised learning and decode the heatmaps via greedy algorithms. We adopt the re-implementation of the two methods proposed in (Li et al., 2025b).

- VAG-CO (Sanokowski et al., 2023) and UTSP (Min et al., 2023). They generate neural heatmaps via unsupervised learning and decode the heatmaps via greedy algorithms. We adopt a re-implementation of the two methods[13] for convenient and consistent evaluation.

**Note.** All evaluations are done with batch size set to 1 in single-thread mode without any multi-sampling tricks on our consistent hardware.

---

[6] https://github.com/Edward-Sun/DIFUSCO (MIT license)
[7] https://github.com/Thinklab-SJTU/Fast-T2T
[8] https://github.com/ml-jku/DIffUCO
[9] https://github.com/Graph-COM/Meta_CO (LGPL-2.1 license)
[10] https://github.com/DIMESTeam/DIMES (MIT license)
[11] https://github.com/Thinklab-SJTU/UniCO
[12] https://github.com/proroklab/gnngls
[13] https://github.com/Thinklab-SJTU/ML4CO-Bench-101 (CC BY 4.0 license)

*Table 16.* A leaderboard on our constructed benchmark datasets, providing an overview of the comparative results regarding the dataset-level and average performance (**Gap** for node tasks and **Obj.** for edge tasks) and the efficiency (**Time**) of problem solving for the neural solvers compared. *The overall index is calculated on average of the "Average (Node)" and "Average (Edge)", weighted over the number of datasets respectively to represent our overall performance on the whole benchmark. Note that the data presented in this table are derived from experiments using consistent **greedy** decoders, with **no** post-processing techniques applied.

| Benchmark | Previous Best (Metric: Gap or Obj.) | | | M$^2$GenCO (ours) | |
|---|---|---|---|---|---|
| | Method | Metric | Time | Metric | Time |
| MIS-BA-SMALL | Fast-T2T (Li et al., 2024) | 4.425% | 0.264s | 3.804% | 0.009s |
| MIS-HK-SMALL | Fast-T2T (Li et al., 2024) | 3.034% | 0.349s | 2.490% | 0.008s |
| MIS-WS-SMALL | Fast-T2T (Li et al., 2024) | 3.470% | 0.372s | 2.741% | 0.008s |
| MIS-BA-LARGE | GOAL (Drakulic et al., 2025) | 4.684% | 36.289s | 4.137% | 0.029s |
| MIS-HK-LARGE | GOAL (Drakulic et al., 2025) | 3.396% | 37.375s | 2.869% | 0.026s |
| MIS-WS-LARGE | GOAL (Drakulic et al., 2025) | 8.360% | 33.232s | 8.701% | 0.026s |
| MIS-SATLIB | GOAL (Drakulic et al., 2025) | 1.719% | 24.799s | 2.662% | 0.020s |
| MCl-BA-SMALL | GOAL (Drakulic et al., 2025) | 3.883% | 0.102s | 2.296% | 0.009s |
| MCl-HK-SMALL | GOAL (Drakulic et al., 2025) | 5.628% | 0.097s | 5.413% | 0.008s |
| MCl-WS-SMALL | GOAL (Drakulic et al., 2025) | 3.839% | 0.142s | 2.917% | 0.010s |
| MCl-TWITTER | GOAL (Drakulic et al., 2025) | 7.124% | 0.208s | 7.836% | 0.013s |
| MCl-BA-LARGE | GOAL (Drakulic et al., 2025) | 5.340% | 1.248s | 2.779% | 0.019s |
| MCl-HK-LARGE | GOAL (Drakulic et al., 2025) | 5.651% | 1.213s | 5.771% | 0.022s |
| MCl-WS-LARGE | Meta-EGN (Wang & Li, 2023) | 28.000% | 0.016s | 17.293% | 0.020s |
| MCl-ER-700-800 | GOAL (Drakulic et al., 2025) | 15.774% | 0.111s | 25.372% | 0.046s |
| MCut-RB-SMALL | Fast-T2T (Li et al., 2024) | 7.304% | 0.349s | 2.951% | 0.009s |
| MCut-HK-SMALL | DiffUCO (Sanokowski et al., 2024) | 0.670% | 0.293s | 1.954% | 0.013s |
| MCut-WS-SMALL | Fast-T2T (Li et al., 2024) | 1.087% | 0.357s | 3.756% | 0.007s |
| MCut-RB-LARGE | Fast-T2T (Li et al., 2024) | 7.304% | 0.349s | 2.951% | 0.009s |
| MCut-HK-LARGE | Fast-T2T (Li et al., 2024) | 0.077% | 0.343s | 1.086% | 0.022s |
| MCut-WS-LARGE | Fast-T2T (Li et al., 2024) | 0.405% | 0.381s | 8.470% | 0.019s |
| MCut-ER-700-800 | DiffUCO (Sanokowski et al., 2024) | 2.309% | 2.000s | 4.425% | 0.043s |
| **Average (Node)** | – | 5.613% | 6.359s | **5.576%** | **0.018s** |
| TSP-Gaussian-50 | Sym-NCO (Kim et al., 2022) | 23.959 | 0.072s | 24.006 | 0.014s |
| TSP-Gaussian-100 | GOAL (Drakulic et al., 2025) | 34.517 | 0.977s | 34.568 | 0.016s |
| TSP-Gaussian-200 | GOAL (Drakulic et al., 2025) | 49.165 | 2.017s | 49.237 | 0.059s |
| TSP-Gaussian-500 | GOAL (Drakulic et al., 2025) | 80.262 | 5.208s | 79.803 | 0.348s |
| TSP-Cluster-50 | Sym-NCO (Kim et al., 2022) | 3.742 | 0.071s | 3.741 | 0.009s |
| TSP-Cluster-100 | Sym-NCO (Kim et al., 2022) | 5.589 | 0.122s | 5.592 | 0.017s |
| TSP-Cluster-200 | GOAL (Drakulic et al., 2025) | 7.091 | 2.243s | 7.135 | 0.121s |
| TSP-Cluster-500 | GOAL (Drakulic et al., 2025) | 11.236 | 5.249s | 11.170 | 0.602s |
| ATSP-HCP-50 | GOAL (Drakulic et al., 2025) | 3.605 | 0.499s | 1.362 | 0.008s |
| ATSP-HCP-100 | GOAL (Drakulic et al., 2025) | 1.748 | 0.983s | 1.284 | 0.010s |
| ATSP-HCP-200 | GOAL (Drakulic et al., 2025) | 4.380 | 2.214s | 0.930 | 0.031s |
| ATSP-SAT-54 | UniCO-DIMES (Pan et al., 2025b) | 7.825 | 0.032s | 2.311 | 0.009s |
| ATSP-SAT-102 | UniCO-DIMES (Pan et al., 2025b) | 13.172 | 0.066s | 4.065 | 0.009s |
| ATSP-SAT-200 | GOAL (Drakulic et al., 2025) | 18.780 | 2.166s | 5.100 | 0.033s |
| **Average (Edge)** | – | 18.934 | 1.566s | **16.451** | **0.092s** |
| **Overall Index*** | – | 10.794 | 4.495s | **9.805** | **0.047s** |
| **Overall Improvement** | – | – | – | **9.162%** | **95.638×** |

# G. Supplementary Experimental Results

## G.1. A General Demonstration of Performance and Efficiency

The relative improvements in solving quality and time efficiency presented in the main text are derived from the averaged results in Table **16**. It should be emphasized that datasets for which a neural method is inapplicable, along with the ATSP-500 results (owing to the scarcity of comparable methods), are excluded from the calculation of the reported average values. With the transparent comparison, it is undoubted that M$^2$GenCO achieve the Pareto optimality over the entire benchmark, since notably previous competitors hardly maintain a balanced optimality of both the solving quality and time efficiency.

*Table 17.* Full results for node-selection problems. Neural models for MIS and MCl are trained on *RB-SMALL/LARGE* dataset, and the models for MCut are trained on *BA-SMALL/LARGE* dataset. For clarity, we keep the comparison with the official Meta-EGN implementation (denoted by [†]), since 8x random solutions are implicitly calculated for best-picking by default. [‡] denotes applying the energy-based sampling (following Feng & Yang (2025); Ma et al. (2025a), etc) for post inference improvement. Competitive learning-based results with ours are **bolded** (both performance and solving time). SMALL: 200-300 nodes; LARGE: 800-1200 nodes.

| MIS | BA-SMALL | | | HK-SMALL | | | WS-SMALL | | |
| --- | --- | --- | --- | --- | --- | --- | --- | --- | --- |
| | Obj.↑ | Gap↓ | Time↓ | Obj.↑ | Gap↓ | Time↓ | Obj.↑ | Gap↓ | Time↓ |
| KaMIS | 72.772* | 0.000±0.000% | 52.940s | 79.372* | 0.000±0.000% | 54.174s | 76.904* | 0.000±0.000% | 51.490s |
| GCN4CO | 66.958 | 7.999±2.431% | 0.021s | 74.860 | 5.678±2.120% | 0.021s | 72.124 | 6.244±2.305% | 0.021s |
| VAG-CO | 66.338 | 8.850±2.629% | **0.009s** | 73.688 | 7.157±2.318% | **0.008s** | 71.198 | 7.433±2.439% | **0.008s** |
| Meta-EGN | 58.900 | 18.983±4.672% | 0.013s | 64.538 | 18.621±4.936% | 0.012s | 62.528 | 18.669±4.814% | 0.011s |
| Meta-EGN[†] | 62.666 | 13.826±2.661% | 0.091s | 69.550 | 12.326±2.366% | 0.093s | 67.046 | 12.739±2.580% | 0.091s |
| DIFUSCO | 66.026 | 9.289±2.763% | 0.587s | 73.802 | 7.008±2.429% | 0.628s | 70.968 | 7.738±2.456% | 0.614s |
| Fast-T2T | 69.556 | 4.425±2.032% | 0.264s | 76.970 | 3.034±1.632% | 0.349s | 74.248 | 3.470±1.794% | 0.372s |
| DiffUCO | 68.802 | 5.453±2.298% | 0.256s | 74.050 | 6.640±2.584% | 0.274s | 71.726 | 6.716±2.511% | 0.265s |
| GOAL | 62.766 | 13.792±3.303% | 0.708s | 69.730 | 12.239±3.471% | 0.804s | 67.520 | 12.282±3.312% | 0.803s |
| M²GenCO | **70.014** | **3.804±1.974%** | **0.009s** | **77.396** | **2.490±1.493%** | **0.008s** | **74.802** | **2.741±1.651%** | **0.008s** |
| M²GenCO‡ | **72.662** | **0.142±0.415%** | **0.034s** | **79.306** | **0.077±0.290%** | **0.033s** | **76.814** | **0.110±0.367%** | **0.034s** |

| MIS | BA-LARGE | | | HK-LARGE | | | WS-LARGE | | | SATLIB | | |
| --- | --- | --- | --- | --- | --- | --- | --- | --- | --- | --- | --- | --- |
| | Obj.↑ | Gap↓ | Time↓ | Obj.↑ | Gap↓ | Time↓ | Obj.↑ | Gap↓ | Time↓ | Obj.↑ | Gap↓ | Time↓ |
| KaMIS | 303.610* | 0.000±0.000% | 59.146s | 330.946* | 0.000±0.000% | 67.272s | 262.570* | 0.000±0.000% | 37.792s | 425.954 | 0.000±0.000% | 24.368s |
| GCN4CO | 269.108 | 11.376±1.566% | 0.025s | 302.946 | 8.452±1.298% | 0.027s | 225.316 | 14.188±1.483% | 0.023s | 408.332 | 4.413±0.848% | 0.028s |
| VAG-CO | 203.552 | 32.952±2.260% | 0.019s | 221.152 | 33.191±2.331% | 0.021s | 169.598 | 35.403±1.832% | **0.015s** | 339.204 | 20.358±1.620% | **0.016s** |
| Meta-EGN | 227.184 | 25.188±2.592% | **0.018s** | 234.544 | 29.128±3.887% | **0.018s** | 213.248 | 18.788±1.639% | 0.018s | 399.828 | 6.138±1.138% | 0.022s |
| Meta-EGN[†] | 237.388 | 21.767±1.654% | 0.142s | 250.206 | 24.349±2.115% | 0.144s | 219.156 | 16.524±1.164% | 0.139s | 406.064 | 4.672±0.687% | 0.170s |
| DIFUSCO | 281.464 | 7.306±1.190% | 1.097s | 312.696 | 5.505±0.948% | 1.067s | 227.464 | 13.375±1.379% | 0.735s | 408.448 | 4.074±0.841% | 0.833s |
| Fast-T2T | 282.168 | 7.059±1.222% | 0.366s | 313.568 | 5.241±1.049% | 0.425s | 232.108 | 11.594±1.476% | 0.337s | 410.260 | 3.688±0.802% | 0.378s |
| DiffUCO | 282.006 | 7.129±1.397% | 0.909s | 303.040 | 8.428±1.363% | 0.595s | 234.854 | 10.545±1.814% | 0.376s | 400.498 | 5.976±1.339% | 0.654s |
| GOAL | 289.406 | 4.684±1.081% | 36.289s | 319.720 | 3.396±0.862% | 37.375s | **240.626** | **8.360±1.285%** | 33.232s | **418.640** | **1.719±0.554%** | 24.799s |
| M²GenCO | **291.290** | **4.056±0.980%** | **0.029s** | **321.426** | **2.869±0.726%** | **0.026s** | 239.738 | 8.701±1.257% | **0.026s** | 414.632 | 2.662±0.691% | **0.020s** |
| M²GenCO‡ | **302.062** | **0.499±0.317%** | **0.444s** | **329.760** | **0.346±0.317%** | **0.446s** | 258.356 | 1.589±0.491% | 0.465s | 420.320 | 1.317±0.417% | 1.091s |

| MCl | BA-SMALL | | | HK-SMALL | | | WS-SMALL | | | TWITTER | | |
| --- | --- | --- | --- | --- | --- | --- | --- | --- | --- | --- | --- | --- |
| | Obj.↑ | Gap↓ | Time↓ | Obj.↑ | Gap↓ | Time↓ | Obj.↑ | Gap↓ | Time↓ | Obj.↑ | Gap↓ | Time↓ |
| Gurobi | 7.478* | 0.000±0.000% | 1.426s | 6.792* | 0.000±0.000% | 1.838s | 7.164* | 0.000±0.000% | 1.589s | 14.210* | 0.000±0.000% | 0.276s |
| GCN4CO | 7.142 | 4.414±6.703% | 0.026s | 6.136 | 9.433±8.864% | 0.024s | 6.512 | 8.975±8.693% | 0.022s | 12.897 | 12.002±16.094% | 0.016s |
| VAG-CO | 7.054 | 5.616±7.732% | 0.008s | 6.050 | 10.756±9.760% | **0.008s** | 6.472 | 9.543±9.843% | **0.007s** | 11.959 | 16.736±17.034% | 0.011s |
| Meta-EGN | 7.056 | 5.575±8.281% | **0.008s** | 6.118 | 9.730±9.325% | 0.010s | 6.522 | 8.821±9.388% | **0.008s** | 12.826 | 10.327±12.429% | **0.008s** |
| Meta-EGN[†] | 7.294 | 2.407±5.376% | 0.112s | 6.396 | 5.669±7.634% | 0.114s | 6.850 | 4.285±7.023% | 0.117s | 13.764 | 3.184±6.561% | 0.115s |
| DIFUSCO | 7.110 | 4.864±7.001% | 0.559s | 6.180 | 8.836±8.733% | 0.561s | 6.664 | 6.883±8.120% | 0.550s | 12.790 | 10.717±12.873% | 0.519s |
| Fast-T2T | 7.052 | 5.609±7.517% | 0.021s | 6.068 | 10.483±9.188% | 0.021s | 6.424 | 10.181±9.451% | 0.021s | 13.051 | 9.535±13.448% | 0.020s |
| GOAL | 7.182 | 3.883±6.442% | 0.102s | 6.398 | 5.628±7.532% | 0.097s | 6.882 | 3.839±6.359% | 0.142s | **13.262** | **7.124±9.726%** | 0.208s |
| M²GenCO | **7.304** | **2.296±5.079%** | **0.009s** | **6.418** | **5.413±7.340%** | **0.008s** | **6.952** | **2.917±5.694%** | 0.010s | 13.108 | 7.836±10.902% | **0.013s** |
| M²GenCO‡ | **7.478** | **0.000±0.000%** | **0.036s** | **6.792** | **0.000±0.000%** | **0.037s** | **7.164** | **0.000±0.000%** | 0.035s | **14.210** | **0.000±0.000%** | 0.032s |

| MCl | BA-LARGE | | | HK-LARGE | | | WS-LARGE | | | ER-700-800 | | |
| --- | --- | --- | --- | --- | --- | --- | --- | --- | --- | --- | --- | --- |
| | Obj.↑ | Gap↓ | Time↓ | Obj.↑ | Gap↓ | Time↓ | Obj.↑ | Gap↓ | Time↓ | Obj.↑ | Gap↓ | Time↓ |
| Gurobi | 7.528* | 0.000±0.000% | 24.524s | 6.774* | 0.000±0.000% | 46.502s | 5.978* | 0.000±0.000% | 27.051s | 6.023* | 0.000±0.000% | 362.808s |
| GCN4CO | 7.080 | 5.866±7.794% | 0.024s | 6.056 | 10.298±9.052% | 0.023s | 3.098 | 48.120±19.082% | 0.022s | 4.227 | 29.781±9.336% | 0.040s |
| VAG-CO | 6.180 | 17.690±12.681% | 0.018s | 5.412 | 19.721±12.779% | 0.018s | 3.404 | 43.000±19.171% | **0.015s** | 4.438 | 26.302±9.283% | 0.047s |
| Meta-EGN | 5.146 | 31.459±19.613% | **0.017s** | 4.754 | 29.399±16.825% | **0.017s** | 4.302 | 28.000±10.832% | 0.016s | 4.523 | 24.851±12.429% | **0.018s** |
| Meta-EGN[†] | 7.056 | 6.188±8.271% | 0.130s | 6.188 | 8.429±9.174% | 0.136s | 5.168 | 13.500±7.636% | 0.124s | 5.188 | 13.821±7.482% | 0.146s |
| DIFUSCO | 7.006 | 6.935±8.568% | 1.426s | 6.064 | 10.267±9.679% | 1.473s | 3.078 | 48.473±19.996% | 1.120s | 4.352 | 27.734±8.721% | 3.888s |
| Fast-T2T | 6.968 | 7.388±8.889% | 0.024s | 6.040 | 10.604±9.276% | 0.024s | 3.578 | 40.127±19.193% | 0.023s | 4.078 | 32.254±9.368% | 0.044s |
| GOAL | 7.118 | 5.340±7.502% | 1.248s | **6.378** | **5.651±7.357%** | 1.213s | 4.176 | 30.113±12.105% | 0.525s | 5.070 | **15.774±7.535%** | 0.111s |
| M²GenCO | **7.318** | **2.779±5.581%** | **0.019s** | 6.376 | 5.771±7.396% | 0.022s | **4.942** | **17.293±12.984%** | 0.020s | 4.492 | 25.372±10.156% | 0.046s |
| M²GenCO‡ | **7.528** | **0.000±0.000%** | **0.070s** | **6.774** | **0.000±0.000%** | 0.071s | **5.978** | **0.000±0.000%** | 0.070s | 5.805 | 3.627±6.858% | 0.087s |

| MCut | RB-SMALL | | | HK-SMALL | | | WS-SMALL | | |
| --- | --- | --- | --- | --- | --- | --- | --- | --- | --- |
| | Obj.↑ | Gap↓ | Time↓ | Obj.↑ | Gap↓ | Time↓ | Obj.↑ | Gap↓ | Time↓ |
| Gurobi | 2526.128* | 0.000±0.000% | 60.228s | 1540.608* | 0.000±0.000% | 60.089s | 872.116* | 0.000±0.000% | 60.357s |
| GCN4CO | 1604.242 | 41.502±27.511% | 0.020s | 1161.482 | 24.650±2.651% | 0.021s | 421.746 | 51.644±4.785% | 0.020s |
| VAG-CO | 203.322 | 89.571±20.701% | **0.009s** | 631.090 | 58.854±8.381% | **0.008s** | 471.720 | 45.900±4.908% | **0.007s** |
| Meta-EGN | 1723.534 | 37.553±35.192% | 0.010s | 1420.520 | 7.792±1.514% | 0.010s | 724.066 | 16.920±3.231% | 0.010s |
| Meta-EGN[†] | 2103.032 | 21.498±28.616% | 0.085s | 1449.694 | 5.888±0.884% | 0.092s | 759.634 | 12.820±1.785% | 0.090s |
| DIFUSCO | 2192.140 | 12.916±5.196% | 0.664s | 1432.272 | 7.080±1.173% | 0.595s | 788.366 | 9.611±1.167% | 0.550s |
| Fast-T2T | 2323.106 | 7.304±5.634% | 0.349s | 1519.216 | 1.393±0.870% | 0.377s | **862.688** | **1.087±0.684%** | 0.357s |
| DiffUCO | 2229.664 | 11.788±5.121% | 0.478s | **1530.308** | **0.670±0.531%** | 0.293s | 842.104 | 3.441±1.389% | 0.203s |
| M²GenCO | **2451.100** | **2.951±2.436%** | **0.009s** | 1510.604 | 1.954±0.821% | 0.013s | 839.466 | 3.756±0.935% | **0.007s** |
| M²GenCO‡ | **2526.232** | **-0.005±0.045%** | **0.053s** | **1540.434** | **0.006±0.186%** | 0.054s | **871.336** | **0.083±0.273%** | 0.051s |

| MCut | RB-LARGE | | | HK-LARGE | | | WS-LARGE | | | ER-700-800 | | |
| --- | --- | --- | --- | --- | --- | --- | --- | --- | --- | --- | --- | --- |
| | Obj.↑ | Gap↓ | Time↓ | Obj.↑ | Gap↓ | Time↓ | Obj.↑ | Gap↓ | Time↓ | Obj.↑ | Gap↓ | Time↓ |
| Gurobi | 31034.390* | 0.000±0.000% | 360.258s | 6401.320* | 0.000±0.000% | 360.357s | 3454.176* | 0.000±0.000% | 360.247s | 23597.930* | 0.000±0.000% | 360.125s |
| GCN4CO | 18617.338 | 41.114±16.153% | 0.046s | 5498.898 | 14.120±1.487% | 0.024s | 2681.698 | 22.374±2.043% | 0.028s | 20294.070 | 13.977±1.369% | **0.010s** |
| VAG-CO | 2365.562 | 90.943±11.100% | 0.050s | 5369.424 | 16.164±2.595% | **0.018s** | 464.110 | 86.548±1.734% | **0.013s** | 0.000 | – | 0.047s |
| Meta-EGN | 22740.768 | 30.254±22.085% | 0.052s | 5880.034 | 8.189±1.600% | 0.069s | 2641.004 | 23.499±2.356% | 0.073s | 20061.195 | 14.964±1.486% | 0.073s |
| Meta-EGN[†] | 25903.352 | 18.767±14.634% | 0.694s | 5994.390 | 6.380±0.703% | 0.641s | 2738.006 | 20.732±1.190% | 0.633s | 20574.094 | 12.807±0.794% | 0.617s |
| DIFUSCO | 26924.378 | 13.079±3.823% | 3.854s | 5502.092 | 14.062±2.834% | 1.259s | 2859.182 | 17.232±1.052% | 0.910s | 21199.203 | 10.149±1.010% | 3.543s |
| Fast-T2T | 27366.140 | **9.926±18.524%** | 0.770s | **6397.248** | **0.077±0.358%** | 0.343s | **3440.488** | **0.405±0.408%** | 0.381s | 21685.750 | 8.115±0.563% | 0.165s |
| DiffUCO | 26394.726 | 15.183±3.154% | 3.710s | 6311.334 | 1.410±0.496% | 0.935s | 3101.098 | 10.231±0.935% | 0.699s | **23054.313** | **2.309±0.727%** | 2.000s |
| M²GenCO | **28186.258** | **8.628±4.294%** | **0.043s** | 6332.672 | 1.086±0.323% | **0.022s** | 3345.523 | 3.155±0.762% | **0.019s** | 22556.570 | 4.425±0.556% | **0.043s** |
| M²GenCO‡ | **31045.142** | **-0.002±0.098%** | **0.071s** | **6415.636** | **-0.217±0.274%** | 0.054s | 3443.290 | 0.314±0.315% | 0.053s | 23504.010 | 0.396±0.085% | 0.072s |

*Table 18.* Full results for edge-selection problems. Neural models are trained on *uniform* 2D-TSP and ATSP dataset and adapted to solving various distinctively distributed instances at different scales. Results on (A)TSP-200 are generalized from the models trained on instances with 100 nodes. [‡] denotes applying Monte Carlo tree search (following Fu et al. (2021), Qiu et al. (2022), etc) for post inference improvement. Competitive learning-based results with ours are **bolded** in terms of both performance and solving time. "Exact" optimal lengths for ATSP instance strictly equal zero as we guarantee the existence of Hamiltonian paths for HCP and satisfiability of SAT cases.

| TSP | Gaussian-50 | | | Gaussian-100 | | | Gaussian-200 | | | Gaussian-500 | | |
|---|---|---|---|---|---|---|---|---|---|---|---|---|
| | Obj.↓ | Gap↓ | Time↓ | Obj.↓ | Gap↓ | Time↓ | Obj.↓ | Gap↓ | Time↓ | Obj.↓ | Gap↓ | Time↓ |
| Concorde | 23.840* | 0.000±0.000% | 0.166s | 34.031* | 0.000±0.000% | 0.438s | 48.127* | 0.000±0.000% | 1.753s | 77.521* | 0.000±0.000% | 19.952s |
| LKH3 | 23.840 | 0.000±0.000% | 0.057s | 34.031 | 0.000±0.000% | s | 48.127 | 0.000±0.000% | 0.373s | 77.521 | 0.000±0.000% | 1.218s |
| GNNGLS | 25.184 | 5.612±5.322% | 0.008s | 37.286 | 9.548±5.140% | 0.012s | 55.625 | 15.570±3.850% | 0.092s | – | – | – |
| UTSP | 29.766 | 24.890±8.160% | 0.006s | 44.714 | 31.400±7.336% | **0.007s** | – | – | – | 97.080 | 25.235±2.770% | **0.054s** |
| DIMES | 26.370 | 10.625±4.297% | **0.004s** | 38.158 | 12.128±3.480% | **0.007s** | 54.977 | 14.210±2.967% | 0.091s | 89.136 | 14.983±1.914% | 0.366s |
| SymNCO | **23.959** | **0.493±0.562%** | 0.072s | 34.806 | 2.264±1.105% | 0.315s | 50.481 | 4.884±1.130% | 0.474s | – | – | – |
| GOAL | 24.094 | 1.064±1.072% | 0.478s | **34.517** | **1.429±0.965%** | 0.977s | **49.165** | **2.161±0.875%** | 2.017s | 80.262 | 3.535±0.928% | 5.208s |
| DIFUSCO | 24.561 | 2.974±5.419% | 0.277s | 35.971 | 5.662±6.484% | 0.377s | 52.447 | 8.965±3.444% | 1.226s | 105.911 | 36.610±5.930% | 8.988s |
| M$^2$GenCO | 24.066 | 0.951±1.059% | **0.014s** | 34.568 | 1.577±1.029% | 0.016s | 49.237 | 2.309±0.951% | 0.059s | 79.803 | 2.945±0.715% | 0.348s |
| M$^2$GenCO[‡] | **23.901** | **0.248±0.231%** | 0.014s | 34.258 | 0.681±0.437% | 0.018s | 48.690 | 1.177±0.525% | 0.088s | 78.899 | 1.783±0.587% | 0.716s |

| TSP | Cluster-50 | | | Cluster-100 | | | Cluster-200 | | | Cluster-500 | | |
|---|---|---|---|---|---|---|---|---|---|---|---|---|
| | Obj.↓ | Gap↓ | Time↓ | Obj.↓ | Gap↓ | Time↓ | Obj.↓ | Gap↓ | Time↓ | Obj.↓ | Gap↓ | Time↓ |
| Concorde | 3.730* | 0.000±0.000% | 0.140s | 5.526* | 0.000±0.000% | 0.290s | 6.912* | 0.000±0.000% | 0.965s | 10.723* | 0.000±0.000% | 5.073s |
| LKH3 | 3.730 | 0.000±0.000% | 0.044s | 5.530 | 0.005±0.003% | 0.114s | 6.913 | 0.011±0.005% | 0.542s | 10.725 | 0.017±0.007% | 2.063s |
| GNNGLS | 4.455 | 19.511±15.824% | 0.008s | 6.890 | 24.790±13.818% | 0.011s | 9.096 | 31.659±7.173% | **0.090s** | – | – | – |
| UTSP | 4.600 | 23.504±12.629% | 0.006s | 6.903 | 25.063±8.997% | **0.007s** | – | – | – | 14.275 | 33.199±4.143% | **0.060s** |
| DIMES | 4.135 | 10.937±5.465% | **0.004s** | 6.378 | 15.429±5.225% | **0.007s** | 8.284 | 19.857±4.294% | 0.090s | 13.003 | 21.262±2.814% | 1.810s |
| SymNCO | 3.742 | 0.340±0.514% | 0.071s | **5.589** | **1.143±0.935%** | 0.122s | 7.130 | 3.158±1.174% | 0.608s | – | – | – |
| GOAL | 3.756 | 0.687±1.126% | 0.504s | 5.618 | 1.651±1.811% | 1.108s | **7.091** | **2.567±1.599%** | 2.243s | 11.236 | 4.786±1.691% | 5.249s |
| DIFUSCO | 3.922 | 5.127±8.310% | 0.276s | 5.756 | 4.142±6.368% | 0.454s | 7.944 | 14.883±9.073% | 1.227s | 15.205 | 41.835±4.866% | 7.268s |
| M$^2$GenCO | 3.741 | 0.291±0.694% | **0.009s** | 5.592 | 1.179±1.405% | 0.017s | 7.135 | 3.207±1.553% | 0.121s | 11.170 | 4.175±1.279% | 0.602s |
| M$^2$GenCO[‡] | **3.732** | **0.061±0.131%** | 0.010s | 5.549 | 0.426±0.566% | 0.019s | 7.050 | 1.939±0.986% | 0.163s | 11.063 | 3.103±0.846% | 0.977s |

| ATSP | ATSP-HCP-50 | | | ATSP-HCP-100 | | | ATSP-HCP-200 | | | ATSP-HCP-500 | | |
|---|---|---|---|---|---|---|---|---|---|---|---|---|
| | Obj.↓ | Gap↓ | Time↓ | Obj.↓ | Gap↓ | Time↓ | Obj.↓ | Gap↓ | Time↓ | Obj.↓ | Gap↓ | Time↓ |
| Exact | 0.000 | – | – | 0.000 | – | – | 0.000 | – | – | 0.000 | – | – |
| LKH3 | 0.000 | N/A | 0.107s | 0.000 | N/A | 0.211s | 0.000 | N/A | 0.355s | 0.000 | N/A | 1.410s |
| MatNet | **1.331** | N/A | 0.050s | 17.538 | N/A | 0.090s | 97.360 | N/A | 0.450s | – | – | – |
| UniCO | 6.109 | N/A | 0.055s | 17.628 | N/A | 0.104s | 85.245 | N/A | 0.477s | – | – | – |
| GOAL | 3.605 | N/A | 0.499s | 1.748 | N/A | 0.983s | 4.380 | N/A | 2.214s | 3.380 | N/A | 7.247s |
| DIMES | 4.885 | N/A | 0.030s | 5.447 | N/A | 0.064s | 5.470 | N/A | 0.220s | – | – | – |
| M$^2$GenCO | 1.362 | N/A | **0.008s** | 1.284 | N/A | **0.010s** | 0.930 | N/A | **0.031s** | 0.680 | N/A | **0.167s** |

| ATSP | ATSP-SAT-54 | | | ATSP-SAT-102 | | | ATSP-SAT-200 | | | ATSP-SAT-507 | | |
|---|---|---|---|---|---|---|---|---|---|---|---|---|
| | Obj.↓ | Gap↓ | Time↓ | Obj.↓ | Gap↓ | Time↓ | Obj.↓ | Gap↓ | Time↓ | Obj.↓ | Gap↓ | Time↓ |
| Exact | 0.000 | – | – | 0.000 | – | – | 0.000 | – | – | 0.000 | – | – |
| LKH3 | 0.151 | N/A | 0.079s | 0.079 | N/A | 0.128s | 0.130 | N/A | 0.192s | 0.430 | N/A | 0.781s |
| MatNet | 32.372 | N/A | 0.053s | 38.713 | N/A | 0.094s | 162.080 | N/A | 0.443s | – | – | – |
| UniCO | 30.152 | N/A | 0.058s | 34.795 | N/A | 0.112s | 97.640 | N/A | 0.503s | – | – | – |
| GOAL | 12.668 | N/A | 0.538s | 14.671 | N/A | 1.029s | 18.780 | N/A | 2.166s | 60.190 | N/A | 7.516s |
| DIMES | 7.825 | N/A | 0.032s | 13.172 | N/A | 0.066s | 21.810 | N/A | 0.212s | – | – | – |
| M$^2$GenCO | 2.311 | N/A | **0.009s** | 4.065 | N/A | **0.009s** | 5.100 | N/A | **0.033s** | 2.310 | N/A | **0.142s** |

## G.2. Complete Results of node-oriented Tasks

The complete generalization results (OOD) of node-oriented tasks (MIS, MCl, and MCut) on our proposed multi-distribution benchmark datasets are presented in Table **17**, as supplementary to Table **3**.

## G.3. Complete Results of edge-oriented Tasks

The complete generalization results (OOD) of edge-oriented tasks (TSP and ATSP) on our proposed multi-distribution benchmark datasets are presented in Table **18**, as supplementary to Table **2**.

## G.4. Supplementary Results of Ablation Study

### G.4.1. On Generative Modeling and Multi-task Meta-pretraining

In addition to Fig **2** in the main content, results in Table **19** compare the full model (w/ multi-task meta-learning and generative backbone) against variants lacking these components (vanilla SL and GNN). Evaluations on both seen and OOD data for node and edge tasks confirm the synergy of these design choices in practice.

*Table 19.* Ablation results on key components (multi-task meta-pretraining and diffusion-based generative modeling).

| Task Type | Seen Data | OOD Data 1 | OOD Data 2 |
|---|---|---|---|
| **MIS** (Obj.↑, Gap↓) | RB-[200-300] | HK-[200-300] | WS-[200-300] |
| | **17.55, 12.49%** (ours) | **73.20, 7.77%** (ours) | **70.44, 8.43%** (ours) |
| | 16.08, 19.77% (w/o meta) | 65.19, 17.86% (w/o meta) | 62.96, 18.12% (w/o meta) |
| | 15.65, 21.95% (w/o model) | 72.58, 8.57% (w/o model) | 69.82, 9.24% (w/o model) |
| **MCl** (Obj.↑, Gap↓) | RB-[200-300] | HK-[200-300] | WS-[200-300] |
| | **16.90, 11.83%** (ours) | **6.08, 10.32%** (ours) | **6.47, 9.56%** (ours) |
| | 11.99, 36.68% (w/o meta) | 5.48, 19.09% (w/o meta) | 5.69, 20.33% (w/o meta) |
| | 13.57, 28.68% (w/o model) | 3.60, 46.67% (w/o model) | 3.68, 48.24% (w/o model) |
| **MCut** (Obj.↑, Gap↓) | BA-[200-300] | HK-[200-300] | WS-[200-300] |
| | **672.66, 7.58%** (ours) | 1114.55, 27.28% (ours) | **754.89, 13.46%** (ours) |
| | 623.24, 14.38% (w/o meta) | **1315.68, 14.61%** (w/o meta) | 487.93, 44.08% (w/o meta) |
| | 616.71, 15.27% (w/o model) | 0.00, 100.00% (w/o model) | 255.88, 70.66% (w/o model) |
| **TSP** (Obj.↓, Gap↓) | Uniform-50 | Gaussian-50 | Cluster-50 |
| | **5.76, 1.27%** (ours) | **24.23, 1.64%** (ours) | **3.76, 0.78%** (ours) |
| | 5.83, 2.41% (w/o meta) | 24.45, 2.56% (w/o meta) | 3.77, 1.02% (w/o meta) |
| | 5.82, 2.35% (w/o model) | 24.44, 2.51% (w/o model) | 3.77, 1.03% (w/o model) |
| **ATSP** (Obj.↓, Gap↓) | Uniform-50 | HCP-50 | SAT-54 |
| | **1.64, 5.49%** (ours) | **1.40** (ours) | 2.30 (ours) |
| | 1.84, 18.33% (w/o meta) | 1.42 (w/o meta) | 3.73 (w/o meta) |
| | 1.76, 13.14% (w/o model) | 1.74 (w/o model) | **1.64** (w/o model) |

### G.4.2. ON FEW-SHOT FINETUNING

The complete ablation results concerning the impact of few-shot finetuning on the 38 support datasets we constructed are presented in Table 20, serving as an augmentation to Table 4. The empirical findings corroborate the substantial efficacy of our finetuning procedure in enabling M$^2$GenCO to rapidly adapt to diverse shifted data distributions.

## G.5. Supplementary Results of Generalization Study

### G.5.1. CROSS-TASK GENERALIZATION

To further validate the generalization capability of the multi-task meta-pretraining, we have conducted supplementary experiments on a new and entirely unseen problem: the Minimum Vertex Cover (MVC) problem. As shown in Table 21, without exposure to any MVC instances and relying solely on model parameters trained via meta-learning on MIS, MCl, and MCut tasks, our proposed model achieves average optimality gaps of only 1.799% and 1.156% on RB-SMALL and RB-LARGE datasets, respectively. Note that these performance can be further enhanced through a minimal number of finetuning steps (empirically, we performed 100 tuning steps). We compare our proposed model with Meta-EGN (Wang & Li, 2023), which is renowned for its generalizability and features a comparable meta-learning paradigm, trained on the MCl problem[14]. Notably, M$^2$GenCO demonstrates far superior zero-shot performance on the new task type while maintaining a favorable solving efficiency.

### G.5.2. CROSS-SCALE GENERALIZATION

Table 22 presents the generalization results for ATSP under the HCP distribution, while Table 23 and Table 24 report the generalization results for TSP. Table 25 lists the generalization results for the three node-oriented tasks. For all experiments in this section, all models are evaluated on data with different scales from the training data. These results demonstrate that our framework retains reasonable performance across scale shifts, regardless of the problem type or graph distribution.

---

[14]In Meta-EGN (Wang & Li, 2023), the model architecture is identical for MCl and MVC, while the MIS model exhibits structural differences, e.g., the number of GCN layers.

*Table 20.* Full results comparing M²GenCO with (w/) and without (w/o) finetuning on the benchmarked OOD distributions. The solving times are omitted for clarity as already provided in previous tables. SMALL: 200-300 nodes; LARGE: 800-1200 nodes.

| Method | MIS-BA-SMALL | | MIS-HK-SMALL | | MIS-WS-SMALL | | | |
|---|---|---|---|---|---|---|---|---|
| | Obj. (↑) | Gap (↓) | Obj. (↑) | Gap (↓) | Obj. (↑) | Gap (↓) | | |
| KaMIS | 72.772 | 0.000% | 79.372 | 0.000% | 76.904 | 0.000% | | |
| M²GenCO w/o f-t | 65.442 | 10.091% | 73.250 | 7.709% | 70.440 | 8.436% | | |
| M²GenCO w/ f-t | 70.014 | 3.804% | 77.396 | 2.490% | 74.802 | 2.741% | | |

| Method | MIS-BA-LARGE | | MIS-HK-LARGE | | MIS-WS-LARGE | | MIS-SATLIB | |
|---|---|---|---|---|---|---|---|---|
| | Obj. (↑) | Gap (↓) | Obj. (↑) | Gap (↓) | Obj. (↑) | Gap (↓) | Obj. (↑) | Gap (↓) |
| KaMIS | 303.610 | 0.000% | 330.946 | 0.000% | 262.570 | 0.000% | 425.954 | 0.000% |
| M²GenCO w/o f-t | 273.362 | 9.961% | 304.342 | 8.028% | 221.222 | 15.755% | 406.33 | 4.614% |
| M²GenCO w/ f-t | 291.070 | 4.137% | 321.426 | 2.869% | 239.738 | 8.701% | 414.632 | 2.662% |

| Method | MCl-BA-SMALL | | MCl-HK-SMALL | | MCl-WS-SMALL | | TWITTER | |
|---|---|---|---|---|---|---|---|---|
| | Obj. (↑) | Gap (↓) | Obj. (↑) | Gap (↓) | Obj. (↑) | Gap (↓) | Obj. (↑) | Gap (↓) |
| Gurobi | 7.478 | 0.000% | 6.792 | 0.000% | 7.164 | 0.000% | 14.210 | 0.000% |
| M²GenCO w/o f-t | 6.998 | 6.316% | 6.070 | 10.455% | 6.410 | 10.373% | 14.210 | 11.084% |
| M²GenCO w/ f-t | 7.304 | 2.296% | 6.418 | 5.413% | 6.952 | 2.917% | 13.108 | 7.836% |

| Method | MCl-BA-LARGE | | MCl-HK-LARGE | | MCl-WS-LARGE | | MCl-ER-700-800 | |
|---|---|---|---|---|---|---|---|---|
| | Obj. (↑) | Gap (↓) | Obj. (↑) | Gap (↓) | Obj. (↑) | Gap (↓) | Obj. (↑) | Gap (↓) |
| Gurobi | 7.528 | 0.000% | 6.774 | 0.000% | 5.978 | 0.000% | 6.023 | 0.000% |
| M²GenCO w/o f-t | 6.422 | 14.521% | 5.818 | 13.918% | 3.360 | 43.760% | 4.054 | 32.645% |
| M²GenCO w/ f-t | 7.318 | 2.779% | 6.376 | 5.771% | 4.942 | 17.293% | 4.492 | 25.372% |

| Method | MCut-RB-SMALL | | MCut-HK-SMALL | | MCut-WS-SMALL | | | |
|---|---|---|---|---|---|---|---|---|
| | Obj. (↑) | Gap (↓) | Obj. (↑) | Gap (↓) | Obj. (↑) | Gap (↓) | | |
| Gurobi | 2526.128 | 0.000% | 1540.608 | 0.000% | 872.116 | 0.000% | | |
| M²GenCO w/o f-t | 2089.270 | 19.053% | 1451.006 | 5.828% | 795.532 | 8.763% | | |
| M²GenCO w/ f-t | 2451.100 | 2.951% | 1510.604 | 1.954% | 839.466 | 3.756% | | |

| Method | MCut-RB-LARGE | | MCut-HK-LARGE | | MCut-WS-LARGE | | MCut-ER-700-800 | |
|---|---|---|---|---|---|---|---|---|
| | Obj. (↑) | Gap (↓) | Obj. (↑) | Gap (↓) | Obj. (↑) | Gap (↓) | Obj. (↑) | Gap (↓) |
| Gurobi | 31034.390 | 0.000% | 6401.320 | 0.000% | 3454.176 | 0.000% | 23597.930 | 0.000% |
| M²GenCO w/o f-t | 11178.868 | 65.772% | 5124.662 | 19.908% | 2709.018 | 21.566% | 16317.977 | 30.513% |
| M²GenCO w/ f-t | 27611.488 | 12.481% | 6332.672 | 1.086% | 3162.112 | 8.470% | 22556.570 | 4.425% |

| Method | TSP-Gaussian-50 | | TSP-Gaussian-100 | | TSP-Gaussian-200 | | TSP-Gaussian-500 | |
|---|---|---|---|---|---|---|---|---|
| | Obj. (↓) | Gap (↓) | Obj. (↓) | Gap (↓) | Obj. (↓) | Gap (↓) | Obj. (↓) | Gap (↓) |
| Concorde | 23.840 | 0.000% | 34.031 | 0.000% | 48.127 | 0.000% | 77.521 | 0.000% |
| M²GenCO w/o f-t | 24.313 | 1.963% | 35.755 | 5.034% | 51.978 | 8.007% | 81.863 | 5.597% |
| M²GenCO w/ f-t | 24.066 | 0.951% | 34.568 | 1.577% | 49.237 | 2.309% | 79.803 | 2.945% |

| Method | TSP-Cluster-50 | | TSP-Cluster-100 | | TSP-Cluster-200 | | TSP-Cluster-500 | |
|---|---|---|---|---|---|---|---|---|
| | Obj. (↓) | Gap (↓) | Obj. (↓) | Gap (↓) | Obj. (↓) | Gap (↓) | Obj. (↓) | Gap (↓) |
| Concorde | 3.730 | 0.000% | 5.526 | 0.000% | 6.912 | 0.000% | 10.723 | 0.000% |
| M²GenCO w/o f-t | 3.752 | 0.609% | 5.614 | 1.583% | 7.315 | 6.912% | 11.201 | 4.452% |
| M²GenCO w/ f-t | 3.741 | 0.291% | 5.592 | 1.179% | 7.153 | 3.478% | 11.170 | 4.175% |

| Method | ATSP-HCP-50 | | ATSP-HCP-100 | | ATSP-HCP-200 | | ATSP-HCP-500 | |
|---|---|---|---|---|---|---|---|---|
| | Obj. (↓) | Gap (↓) | Obj. (↓) | Gap (↓) | Obj. (↓) | Gap (↓) | Obj. (↓) | Gap (↓) |
| LKH (1000) | 0.000 | N/A | 0.000 | N/A | 0.000 | N/A | 0.000 | N/A |
| M²GenCO w/o f-t | 1.421 | N/A | 1.370 | N/A | 1.740 | N/A | 0.990 | N/A |
| M²GenCO w/ f-t | 1.362 | N/A | 1.284 | N/A | 0.930 | N/A | 0.680 | N/A |

| Method | ATSP-SAT-54 | | ATSP-SAT-102 | | ATSP-SAT-200 | | ATSP-SAT-507 | |
|---|---|---|---|---|---|---|---|---|
| | Obj. (↓) | Gap (↓) | Obj. (↓) | Gap (↓) | Obj. (↓) | Gap (↓) | Obj. (↓) | Gap (↓) |
| LKH (1000) | 0.151 | N/A | 0.079 | N/A | 0.130 | N/A | 0.430 | N/A |
| M²GenCO w/o f-t | 5.363 | N/A | 7.186 | N/A | 23.900 | N/A | 2.320 | N/A |
| M²GenCO w/ f-t | 2.311 | N/A | 4.065 | N/A | 5.100 | N/A | 2.320 | N/A |

*Table 21.* Cross-task generalization results on MVC. "w/o f-t": without finetune, denoting the model equipped with $\theta_{\text{meta}}$ (trained on MIS, MCl, and MCut) for certain scales. "w/ f-t": with finetune, 100 global steps of gradient descents, learning rate: 1e-5, batch size: 32/4 for SMALL/LARGE scales. We compare M$^2$GenCO with Meta-EGN trained on the MCl-RB data and generalized to MVC.

| Method | RB-SMALL | | | RB-LARGE | | |
|---|---|---|---|---|---|---|
| | Obj.↓ | Gap↓ | Time↓ | Obj.↓ | Gap↓ | Time↓ |
| Gurobi (Gurobi Optimization, 2023) | 205.764* | 0.000±0.000% | 3.341s | 968.228* | 0.000±0.000% | 290.227s |
| Meta-EGN (4×) (Wang & Li, 2023) | 218.566 | 6.254±1.890% | 0.039s | 1000.396 | 3.332±0.745% | 0.204s |
| M$^2$GenCO (w/o f-t) | 209.450 | 1.799±0.704% | 0.007s | 979.388 | 1.156±0.246% | 0.028s |
| M$^2$GenCO (w/ f-t) | **209.202** | **1.708±0.671%** | **0.007s** | **977.300** | **0.939±0.223%** | **0.028s** |

*Table 22.* Cross-scale generalization results on ATSP (metric: **Obj.↓**). Gray box: in-distribution results. **Bold**: best results.

| Training Set — Testing Set | HCP-50 | HCP-100 | HCP-200 | HCP-500 |
|---|---|---|---|---|
| HCP-50 | 1.362 | 1.813 | 2.410 | **1.336** |
| HCP-100 | **1.254** | 1.284 | 2.234 | 1.444 |
| HCP-200 | 1.300 | 0.980 | **0.920** | 1.310 |
| HCP-500 | 0.780 | 0.690 | 0.680 | **0.680** |

*Table 23.* Cross-scale generalization results on TSP (metrics: Obj.↓, Gap↓).

| Training Set — Testing Set | Gaussian-50 | Gaussian-100 | Gaussian-200 | Gaussian-500 |
|---|---|---|---|---|
| Gaussian-50 | **24.07, 0.95%** | 24.33, 2.06% | 24.80, 4.02% | 25.32, 6.20% |
| Gaussian-100 | 36.48, 7.21% | **34.57, 1.58%** | 35.05, 3.02% | 36.61, 7.58% |
| Gaussian-200 | 52.51, 9.12% | 52.49, 9.05% | **49.24, 2.31%** | 52.49, 9.08% |
| Gaussian-500 | 84.81, 9.40% | 84.68, 9.24% | 84.89, 9.50% | **79.80, 2.95%** |

*Table 24.* Supplementary results on uniform TSP-1000.

| Method | Obj.↓ | Gap↓ |
|---|---|---|
| Concorde | 23.12 | 0.00% |
| M$^2$GenCO (w/o f-t) | 24.01 | 3.86% |
| M$^2$GenCO (w/ f-t) | 23.98 | 3.73% |

*Table 25.* Cross-scale generalization results on node-oriented tasks (metrics: Obj.↑, Gap↓).

| Training Data | Test Data | Obj.↑ | Gap↓ |
|---|---|---|---|
| MIS-BA-SMALL | MIS-BA-LARGE | 291.070 | 4.137% |
| MIS-HK-SMALL | MIS-HK-LARGE | 321.354 | 2.892% |
| MIS-WS-SMALL | MIS-WS-LARGE | 225.752 | 14.025% |
| MCl-BA-SMALL | MCl-BA-LARGE | 7.240 | 3.785% |
| MCl-HK-SMALL | MCl-HK-LARGE | 6.214 | 8.136% |
| MCl-WS-SMALL | MCl-WS-LARGE | 3.384 | 43.353% |
| MCut-RB-SMALL | MCut-RB-LARGE | 27611.488 | 12.481% |
| MCut-HK-SMALL | MCut-HK-LARGE | 6315.848 | 1.346% |
| MCut-WS-SMALL | MCut-WS-LARGE | 3162.112 | 8.470% |

*Table 26.* Run-wise variance of objective values of M$^2$GenCO. The reported values are the average objective $\pm$ standard deviation over 10 independent solving executions with random seeds 0–9. SMALL: 200-300 nodes, LARGE: 800-1200 nodes.

| MIS | | MCl | | MCut | | TSP | | ATSP | |
|---|---|---|---|---|---|---|---|---|---|
| **Dataset** | **obj.$\pm$std.** | **Dataset** | **obj.$\pm$std.** | **Dataset** | **obj.$\pm$std.** | **Dataset** | **obj.$\pm$std.** | **Dataset** | **obj.$\pm$std.** |
| BA-SMALL | 70.04$\pm$0.02 | BA-SMALL | 7.30$\pm$0.00 | RB-SMALL | 2447.66$\pm$2.19 | Gauss.-50 | 24.07$\pm$0.02 | HCP-50 | 1.37$\pm$0.01 |
| HK-SMALL | 77.38$\pm$0.01 | HK-SMALL | 6.42$\pm$0.01 | HK-SMALL | 1510.44$\pm$0.13 | Gauss.-100 | 34.56$\pm$0.02 | HCP-100 | 1.29$\pm$0.01 |
| WS-SMALL | 74.79$\pm$0.01 | WS-SMALL | 6.95$\pm$0.01 | WS-SMALL | 839.74$\pm$0.35 | Gauss.-200 | 49.24$\pm$0.03 | HCP-200 | 0.90$\pm$0.05 |
| BA-LARGE | 291.09$\pm$0.04 | TWITTER | 13.17$\pm$0.05 | RB-LARGE | 27600.42$\pm$33.93 | Gauss.-500 | 79.81$\pm$0.04 | HCP-500 | 0.65$\pm$0.08 |
| HK-LARGE | 321.42$\pm$0.02 | BA-LARGE | 7.31$\pm$0.01 | HK-LARGE | 6332.19$\pm$0.25 | Cluster-50 | 3.74$\pm$0.01 | SAT-54 | 2.31$\pm$0.01 |
| WS-LARGE | 239.72$\pm$0.08 | HK-LARGE | 6.38$\pm$0.01 | WS-LARGE | 3163.60$\pm$1.65 | Cluster-100 | 5.59$\pm$0.01 | SAT-102 | 4.05$\pm$0.01 |
| SATLIB | 414.61$\pm$0.06 | WS-LARGE | 4.88$\pm$0.03 | ER-700-800 | 22556.75$\pm$11.54 | Cluster-200 | 7.15$\pm$0.02 | SAT-200 | 5.34$\pm$0.21 |
| - | - | ER-700-800 | 4.49$\pm$0.05 | - | - | Cluster-500 | 11.17$\pm$0.03 | SAT-507 | 2.32$\pm$0.01 |

## G.6. Supplementary Results of Stability Study

### G.6.1. SOLVING STABILITY: VARIANCE OF OBJECTIVE ACROSS RUNS

To clarify, the originally reported standard deviations of gaps (as in Table **G.2** and Table **18**) reflect instance-wise heterogeneity (e.g., different problem instances have different difficulty level), rather than the instability of our solving process. 38 supplementary tests show small execution-wise std (e.g., MIS-BA: $70.04 \pm 0.02$) v.s. larger instance-wise std ($70.04 \pm 8.88$), confirming stability of our solver over 10 random seeds. Full results are provided in Table **26**.

### G.6.2. TRAINING STABILITY: ADAPTIVE TUNING WITH SUB-OPTIMAL SUPERVISION

To study the anti-noise robustness of our method, we have conducted experiments with two settings in the tuning stage, both using sub-optimal solutions for supervision.

**Setting 1: Randomly perturbed reference solutions.** For these experiments, we introduced controlled noise by randomly flipping a fraction of the binary labels in the supervision data (e.g., p=1% or 10% of edges/nodes in the solution, which is supposed to be significant). Fine-tuning was performed for 8,000 global steps (fewer than the full process in the main text due to rebuttal time constraints, but sufficient to reveal clear trends), with results compared to tuning on "best-quality" labels (originally solved by Gurobi or LKH), as shown in Table **27** and Table **28**.

*Table 27.* Tuning with suboptimal supervision (perturbed solutions): edge-oriented tasks.

| Setting | TSP (Gaussian-100) | | ATSP (HCP-100) |
|---|---|---|---|
| | **Obj.$\downarrow$** | **Gap$\downarrow$** | **Obj.$\downarrow$** |
| best-quality data (Concorde/LKH) | 34.564 | 1.568% | 1.304 |
| sub-optimal data (p=1%) | 34.608 | 1.696% | 1.200 |
| sub-optimal data (p=10%) | 34.936 | 2.660% | 1.131 |

*Table 28.* Tuning with suboptimal supervision (perturbed solutions): node-oriented tasks.

| Setting | MIS (BA-200-300) | | MCl (BA-200-300) | |
|---|---|---|---|---|
| | **Obj.$\uparrow$** | **Gap$\downarrow$** | **Obj.$\uparrow$** | **Gap$\downarrow$** |
| best-quality data (KaMIS/Gurobi) | 69.760 | 4.148% | 7.242 | 3.118% |
| sub-optimal data (p=10%) | 69.532 | 4.465% | 7.176 | 3.967% |

**Setting 2: Valid solution with subpar quality.** For TSP, supervision labels were generated using LKH with max_trials set to 1 (compared to the originally used Concorde for exact solutions). For MIS, labels were generated via a heuristic that greedily selects nodes with the lowest degrees (instead of the originally used KaMIS for solutions with near-optimal quality). The quantitative metrics for the data quality used for tuning models on respective tasks are given in Table **29** and Table **30**. The results are presented in Table **31** and Table **32**.

## G.7. Supplementary Results of In-distribution Performance

The full in-distribution (ID) results on ten datasets, including random bipartite (RB) data for the MIS and MCl problems, BA data for the MCut problem, and uniform/tmat matrices for TSP and ATSP, are reported in Table **33**, supplementing Table **5**. We benchmark M$^2$GenCO against two meta-learning frameworks: DIMES (Qiu et al., 2022) for (A)TSP and Meta-EGN (Wang & Li, 2023) for node-oriented tasks. Experimental results demonstrate that M$^2$GenCO significantly

*Table 29.* Proportion of suboptimal solutions in the tuning data used for TSP in the test with setting 2.

| Tuning Dataset | Suboptimal Rate |
|---|---|
| Gaussian-50 | 6.14% |
| Gaussian-100 | 23.8% |
| Gaussian-200 | 53.0% |
| Cluster-50 | 1.4% |
| Cluster-100 | 10.0% |
| Cluster-200 | 35.9% |

*Table 30.* Average optimality gap of the tuning data used for MIS in the robustness test with setting 2.

| Tuning Dataset | Optimality Gap |
|---|---|
| MIS-BA | 4.1% |
| MIS-HK | 4.6% |
| MIS-WS | 5.1% |

*Table 31.* Tuning with suboptimal supervision (subpar solution quality): results on TSP.

| Setting | Gaussian-50 | | Cluster-50 | | Gaussian-100 | |
|---|---|---|---|---|---|---|
| | Obj.↓ | Gap↓ | Obj.↓ | Gap↓ | Obj.↓ | Gap↓ |
| best-quality data | 24.066 | 0.951% | 3.741 | 0.291% | 34.564 | 1.568% |
| sub-optimal data | 24.088 | 1.035% | 3.745 | 0.402% | 34.566 | 1.569% |

| Setting | Cluster-100 | | Gaussian-200 | | Cluster-200 | |
|---|---|---|---|---|---|---|
| | Obj.↓ | Gap↓ | Obj.↓ | Gap↓ | Obj.↓ | Gap↓ |
| best-quality data | 5.592 | 1.179% | 49.237 | 2.309% | 7.153 | 3.478% |
| sub-optimal data | 5.605 | 1.419% | 49.590 | 3.047% | 7.174 | 3.777% |

*Table 32.* Tuning with suboptimal supervision (subpar solution quality): results on MIS.

| Setting | BA-200-300 | | HK-200-300 | | WS-200-300 | |
|---|---|---|---|---|---|---|
| | Obj.↑ | Gap↓ | Obj.↑ | Gap↓ | Obj.↑ | Gap↓ |
| best-quality data | 70.014 | 3.804% | 77.396 | 2.490% | 74.802 | 2.741% |
| sub-optimal data | 68.738 | 5.551% | 76.112 | 4.106% | 73.480 | 4.466% |

outperforms these baselines, even when DIMES employs its time-consuming active search procedure and Meta-EGN utilizes multi-sampling strategies (4x).

*Table 33.* Full results (objective, time) on the in-distribution data, comparing M²GenCO with task-specific meta-learners. † denotes results solved by UniCO (Pan et al., 2025b) which extends the DIMES to solving ATSP. "Reference" refers to the same (near-)optimal solvers aforementioned for each task.

| Method | MIS-RB-SMALL | | MCl-RB-SMALL | | MCut-BA-SMALL | | TSP-Uniform-50 | | ATSP-Uniform-50 | |
|---|---|---|---|---|---|---|---|---|---|---|
| | Obj. (↑) | Time | Obj. (↑) | Time | Obj. (↑) | Time | Obj. (↓) | Time | Obj. (↓) | Time |
| Reference | 20.09 | 45.809s | 19.08 | 0.900s | 727.84 | 60.612s | 5.69 | 0.059s | 1.55 | 0.097s |
| DIMES | – | – | – | – | – | – | 6.32 | 0.015s | 2.57† | 0.033s |
| DIMES ($T_{AS} = 5$) | – | – | – | – | – | – | 6.25 | 1.142s | 2.14† | 1.163s |
| Meta-EGN | 16.40 | 0.010s | 13.80 | 0.008s | 673.97 | 0.009s | – | – | – | – |
| Meta-EGN (4×) | 17.33 | 0.040s | 16.95 | 0.036s | 684.29 | 0.038s | – | – | – | – |
| M²GenCO (Ours) | **17.54** | **0.008s** | **17.04** | **0.008s** | **703.30** | **0.008s** | **5.73** | **0.007s** | **1.64** | **0.006s** |

| Method | MIS-RB-LARGE | | MCl-RB-LARGE | | MCut-BA-LARGE | | TSP-Uniform-100 | | ATSP-Uniform-100 | |
|---|---|---|---|---|---|---|---|---|---|---|
| | Obj. (↑) | Time | Obj. (↑) | Time | Obj. (↑) | Time | Obj. (↓) | Time | Obj. (↓) | Time |
| Reference | 43.00 | 56.974s | 40.18 | 276.657s | 2936.89 | 300.214s | 7.76 | 0.238s | 1.57 | 0.238s |
| DIMES | – | – | – | – | – | – | 8.76 | **0.018s** | 2.63† | 0.098s |
| DIMES ($T_{AS} = 5$) | – | – | – | – | – | – | 8.64 | 1.364s | 2.52† | 1.373s |
| Meta-EGN | 32.68 | 0.071s | 25.45 | 0.066s | 2704.89 | 0.104s | – | – | – | – |
| Meta-EGN (4×) | 34.18 | 0.272s | 31.70 | 0.247s | **2754.80** | 0.395s | – | – | – | – |
| M²GenCO (Ours) | **34.59** | **0.066s** | **32.20** | **0.066s** | 2735.88 | **0.035s** | 7.88 | 0.026s | 1.66 | 0.020s |

### G.8. Hypar-parameter Study on Learning Rates

We first note that prior meta-NCO works, including Meta-EGN (Wang & Li, 2023) and DIMES (Qiu et al., 2022), did not provide explicit analyses on the choices of inner/outer learning rates, nor did they present experiments examining their effects. Nevertheless, we agree that such validation is valuable. To this end, we conduct a representative and systematic (yet simple and clear) study with 1000 epochs of meta-pretraining that varies both the relative ratio and the absolute magnitude of $\alpha$ and $\beta$. Table 34, Table 35, and Table 36 gives the results on **1) fixing** $\beta/\alpha = 10$, **2) varying** $\beta$ **with fixed** $\alpha$ ($\beta/\alpha = 1, 5, 10, 20$), and **3) varying** $\alpha$ **with fixed** $\beta$ ($\beta/\alpha = 1, 5, 10, 20$).

*Table 34.* Effect of varying absolute LR magnitude while keeping $\beta/\alpha = 10$.

| Outer LR ($\beta$) | Inner LR ($\alpha$) | MIS↑ | MCl↑ | MCut↑ |
|---|---|---|---|---|
| 1e−4 | 1e−5 | 16.128 | 13.184 | 629.134 |
| 2e−4 | 2e−5 | 16.156 | 12.008 | 645.008 |
| 5e−4 | 5e−5 | **16.732** | **13.480** | **657.190** |
| 7e−4 | 7e−5 | 16.120 | 11.998 | 632.868 |
| 1e−3 | 1e−4 | 16.098 | 13.436 | 649.862 |

*Table 35.* Effect of varying $\beta$ while fixing $\alpha = 5e-5$.

| Outer LR ($\beta$) | Inner LR ($\alpha$) | MIS↑ | MCl↑ | MCut↑ |
|---|---|---|---|---|
| 5e−5 | 5e−5 | 16.390 | 12.280 | 626.846 |
| 2.5e−4 | 5e−5 | 16.186 | 12.576 | 633.964 |
| 5e−4 | 5e−5 | **16.732** | **13.480** | **657.190** |
| 1e−3 | 5e−5 | 16.110 | 12.226 | 649.698 |

*Table 36.* Effect of varying $\alpha$ while fixing $\beta = 5e-4$.

| Outer LR ($\beta$) | Inner LR ($\alpha$) | MIS↑ | MCl↑ | MCut↑ |
|---|---|---|---|---|
| 5e−4 | 5e−4 | 16.200 | 12.298 | 550.984 |
| 5e−4 | 1e−4 | 16.218 | 13.468 | 633.722 |
| 5e−4 | 5e−5 | **16.732** | **13.480** | **657.190** |
| 5e−4 | 2.5e−5 | 16.600 | 11.560 | 526.500 |

The results show that: 1) the learning process is **not overly sensitive to moderate changes in the learning rates**, indicating stable behavior within a reasonable range; 2) a **straightforward and coarse-grained grid search suffices** to identify appropriate values for $\alpha$ and $\beta$; 3) learning rates substantially larger than the tested range lead to instability under our batch size, hence we intentionally avoid configurations known to diverge; 4) as commonly acknowledged in MatNet (Kwon et al., 2021) and general ML practice, learning rates should be adjusted when batch size changes. Our experiments therefore provide a clear reference for stable configurations, while allowing readers with different settings to adjust accordingly.

### G.9. Supplementary Comparison of Computational Resources for Training

Table 37 presents a comprehensive comparison of the computational resources and the training durations needed for M²GenCO in both the multi-task meta-pretraining and OOD finetuning phases. It is important to note that the time cost can vary depending on factors such as different GPU types, CPU types, memory capacity, software versions, and other arbitrary system-related elements. Additionally, we have excluded the time required by validation during training, because it may differ based on specific configurations. Despite these potential variations, the reference results clearly demonstrate a substantial acceleration and a reduction in the training costs for the diffusion model within the realm of NCO. This enhancement can be primarily attributed to the lightweight nature of our model architecture and the integration of the meta-learning mechanism that is less data-demanding.

*Table 37.* Comparison of the training resources required. † denotes data quoted from the original papers. * denotes training with a sparsification factor $K = 50$ while all our models for edge-tasks are trained using dense matrices. The training time is in accord with the device type, the batch size (with a single GPU), and the total iterations given in Table 15. The time for validation and data loading do not count towards the training time. For edge-tasks we report TSP/ATSP, and for edge-tasks we report MIS/MCut, respectively, while neglecting the variance induced by data distributions.

| Method | Problem Scale | Batch size | GPU Memory | Train time | GPU Device |
|---|---|---|---|---|---|
| M$^2$GenCO (pretrain) | Node-small | 4 | 2.01 GB | 6h25m | NVIDIA 3090 (24GB) |
| | Node-large | 2 | 33.64 GB | 12h37m | NVIDIA H100 (80GB) |
| | Edge-50 | 4 | 2.20 GB | 17h28m | NVIDIA 3090 (24GB) |
| | Edge-100 | 4 | 7.09 GB | 24h10m | NVIDIA 3090 (24GB) |
| | Edge-500 | 2 | 64.39 GB | 47h35m | NVIDIA H100 (80GB) |
| M$^2$GenCO (finetune) | Node-small | 32 | 8.65/5.07 GB | 2h6m/16m | NVIDIA 3090 (24GB) |
| | Node-large | 16 | 14.07/5.42 GB | 28m/37m | NVIDIA 3090 (24GB) |
| | Edge-50 | 32 | 2.21/2.54 GB | 1h10m/1h20m | NVIDIA 3090 (24GB) |
| | Edge-100 | 12 | 2.92/3.09 GB | 1h30m/1h29m | NVIDIA 3090 (24GB) |
| | Edge-200 | 8 | 7.13/7.13 GB | 1h52m/1h48m | NVIDIA 3090 (24GB) |
| | Edge-500 | 8 | 41.80/42.96 GB | 8h35m/7h28m | NVIDIA H100 (80GB) |
| T2TCO† (Li et al., 2023) | Edge-50 | 32 | 13.2 GB | 56h24m | NVIDIA A100 |
| | Edge-100 | 12 | 24.0 GB | 206h20m | NVIDIA A100 |
| | Edge-500 | 6 | 19.1 GB* | 64h26m | NVIDIA A100 |
| Fast-T2T† (Li et al., 2024) | Edge-50 | 32 | 16.5 GB | 112h45m | NVIDIA A100 |
| | Edge-100 | 12 | 23.2 GB | 448h12m | NVIDIA A100 |
| | Edge-500 | 6 | 37.8 GB* | 142h17m | NVIDIA A100 |

# H. Extended Discussions

## H.1. On Synergetic Patterns and Techniques

**1. Shared structure across graph-based COPs.** We respectfully clarify that our work explicitly targets the family of graph-based COPs (as stated early in the abstract, line 16), a domain broad enough to encompass a wide spectrum of NCO studies. Within this scope, many canonical tasks, including MIS, MVC, TSP, and MCl, can be interpreted under a unified formulation of binary structural decisions on graphs. Although these COPs differ in their high-level objectives, they share fundamental low-level regularities such as node–edge interaction patterns, local feasibility constraints. Through multi-task meta-pretraining, M$^2$GenCO is expected to capture as much these reusable and problem-agnostic structural priors, which naturally facilitates transferability across heterogeneous graph-based COPs.

**2. Meta-learning aligns adaptation dynamics across tasks.** The outer-loop meta optimization explicitly encourages updates that are useful for multiple problem types. As highlighted in the initial submission, this produces a parameter initialization that is particularly toward features capturing invariances across COPs rather than per-problem heuristics. This also explains why our model exhibits strong few-shot adaptation under shifts in both problem type and distribution.

**3. Technical synergy: cross-task denoising aligns naturally with meta-learning.** Because many graph-based COPs have optimal solutions that lie on similarly structured discrete manifolds, training on multiple tasks exposes the model to a much richer set of valid solution transitions, which is a case traditional supervised learning (directly imitating the final optimal solution) cannot fully exploit. In contrast, the diffusion module learns denoising trajectories that capture the underlying geometry of these shared solution manifolds, making it naturally suitable for modeling cross-task variability. This broader and more structured exposure also, in turn, benefits the meta-learning process by: 1) shaping a smoother and more transferable loss landscape for inner-loop adaptation, 2) reducing overfitting to task-specific solution patterns, and 3) encouraging the model to acquire priors that are inherently cross-task.

**In general**, we expect the stability of cross-task meta-pretraining to be supported by the following mechanistic designs:

- **Data level.** Different COPs defined on the same or similar graph families share highly similar structural input distributions. Thus, the encoder consistently processes comparable node-edge topologies across tasks.

- **Model level.** Shared graph structure is captured by common GNN-based encoders, whereas task-specific differences are confined to lightweight, easily adaptable components (e.g., task embeddings and output heads). This enables a unified latent manifold with minimal interference across tasks.

- **Learning level.** Supervision is provided as homogeneous optimal signals (binary vectors/matrices describing the optimal assignment of decision variables). With only minimal task-specific prompts, the model effectively learns mappings from latent instance representations to these universal output formats, encouraging task-agnostic feature extraction.

## H.2. On Clarifying the Concept of Meta-Learning

**Remark on Conceptual Nuances.** First, when transplanting meta-learning from canonical domains into specialized areas such as graph-based NCO, **conceptual shifts are natural and widely accepted**. This mirrors well-established precedents in the NCO community itself: e.g., AM (Kool et al., 2018) introduced autoregressive policies for routing despite strong Markovian assumptions, PO4CO (Pan et al., 2025a), BOPO (Liao et al., 2025), and UCPO (Fang et al., 2026) adapted preference-optimization ideas from RLHF for combinatorial search. Such cross-domain methodological adaptation is both common and constructive, even when the mathematical correspondence is not strictly one-to-one. In this light, the nuanced semantics of "task" in graph-based COPs should be viewed as a reasonable abstraction rather than a flaw, especially given the strong empirical validation presented in our work.

**Technical Design for Multi-task Meta-NCO.** Classical meta-learning assumes that tasks are drawn from a *task family*, not that meta-train and meta-test tasks must follow *identical* distributions. In our work, each task is formally defined as a tuple (COP type, graph distribution, scale), i.e., $\mathcal{T}_{i,j}$, and the task distribution $p(\mathcal{T})$ is explicitly constructed over this tri-level space. Our meta-learner is therefore trained exactly to enable fast adaptation across such tasks, consistent with the core objective of task-level meta-learning.

**Differences from Transfer Learning.** M$^2$GenCO is fundamentally distinct from standard transfer learning. Unlike conventional pretrain-finetune pipelines (e.g., ImageNet pretraining), our framework explicitly performs **i) sampling over heterogeneous COP tasks**, **ii) a bi-level MAML-style inner-outer optimization procedure**, and **iii) optimization *for* fast adaptation** rather than for source-task performance alone. The empirical gap between "w/ meta" and "w/o meta" ablations further confirms that our framework cannot be reduced to vanilla transfer learning or simple multi-task learning.

Finally, our experiments intentionally examine realistic cross-distribution settings (e.g., MIS→MVC, RB→BA→WS, Uniform→Gaussian→Cluster), as these constitute practically relevant shifts in graph-based combinatorial optimization. Studying such shifts *strengthens* rather than weakens the meta-learning framing in this domain. For clarity, we explicitly position our method in the revised submission as a **multi-task, cross-distribution meta-learning inspired** framework while **maintaining the essential and fully consistent MAML-style bi-level optimization** at its core.

## H.3. Open Discussion on the Single-step Inner-Update

We find the observation that a single inner-loop gradient step ($t_{in} = 1$) outperforms multi-step updates ($t_{in} = 5, 10$) may appear counter-intuitive at first glance. We offer a brief speculation/clarification/analysis regarding this phenomenon:

First, in our setting, the backbone is a discrete diffusion–based generative model, whose parameters are sensitive to local perturbations, and each denoising step could rely on a carefully shaped latent trajectory, and aggressive multi-step adaptation may degrade the quality of the learned manifold over the multi-task data. Also, because the model is pretrained jointly across heterogeneous COP tasks toward a learned initialization that encodes a well-balanced and broadly useful latent structure, performing large inner-loop updates on a single meta-batch (each corresponding to a specific task type and distribution), might possibly "overfit" toward that task and disrupt the goal targeting shared cross-task initialization, leading to instability or underperformance. This is consistent with the empirical findings that additional gradient steps yield minimal or even negative returns while incurring non-trivial computational overhead. A single inner-loop update thus probably acts as a stable and effective compromise, gently guiding the model toward task-specific signals during multi-task meta-pretraining without distorting the generative dynamics that the outer loop aims to shape.

**Remark.** We would emphasize that the explanation above is intended as a *reasonable and empirically grounded hypothesis, rather than a complete theoretical proof.* A deeper mechanistic or theoretical investigation, while certainly interesting, lies beyond the scope of this work. Hopefully our analysis encourages further exploration of this scientifically appealing question in future work.

# I. Broader Impact

The proposal of M²GenCO, a multi-task meta-learning framework for diffusion-based combinatorial optimization, enables cross-problem generalization over unseen distributions with demonstrated efficiency and effectiveness. To our best knowledge, this framework represents one of the earliest efforts to join the three mainstream research objectives in the realm of NCO solving, i.e., the advanced neural backbone and training mechanism (diffusion), the pursuit of universal learning and solving pipeline capable of tackling diverse problem categories concurrently, and the importance attached to the OOD generalizability with efficient model adaptation via meta-learning. Meanwhile, the proposed standardized benchmark is expected to foster comprehensive and transparent comparison in terms of the generalization performance on distinctively distributed datasets with real-world interpretations. We believe the tri-leveled integration of *paradigmatic framework* (multi-task meta-learning), *generative modeling* (diffusion model with optimization consistency), and *benchmarking protocol* (multi-distribution datasets), well position this work as a foundational step toward  joint-pretraining and robust adaptation of neural combinatorial solvers with potentially broader industrial implications.

# J. Limitation and Outlook: A Further Discussion

**On the Problem Scope and Evaluation.** In this part, we provide a detailed discussion on the current limitations and the possible future improvements. First, the current framework focuses on classic graph-based COPs (e.g., TSP, MIS) and has not yet applied on non-graph structured problems. We note that this limitation is inherent to the *global-prediction* (GP) nature of generative, heatmap-guided solvers, and is therefore shared across the broader family of such approaches. In parallel, the equally mainstream *local-construction* (LC) paradigm in NCO (represented by influential works like AM (Kool et al., 2018), POMO (Kwon et al., 2020), and GOAL (Drakulic et al., 2025)) offers substantially greater flexibility for handling complex constraints through autoregressive decision making, albeit at the cost of reasonably slower inference. We believe that the choice between GP and LC solvers is ultimately an engineering decision, depending on the scope of problem types, available computational budget, time-efficiency requirements, and practitioner preferences over different evaluation metrics. Our reported results using a *vanilla greedy decoder* (without sampling or post-processing) are intended to provide a clear and meaningful reference for such decisions. **We remain fully open to continued innovation within both paradigms, and emphasize that advantages or weaknesses observed in any single metric should neither be overstated nor unduly criticized; rather, the speed-accuracy-flexibility trade-off across paradigms is what holds primary significance.**

**On Scalability and Benchmarks.** We acknowledge that ultra-large instances (e.g., graphs with 10k and even more nodes) have been unevaluated at present. We openly recommend full-parameter, single-task generative solvers for engineering-level large-scale deployment where scalability is the primary objective. Regarding the benchmarking effort, while the benchmark includes 38 datasets, real-world applications may encounter more complex distributions (e.g., dynamic networks with time-varying constraints) that require further adaptation.

**Future Works.** In the future, promising research directions include: 1) extend the framework to a wider spectrum of COPs, e.g., the KP, CVRP, etc., and if seek further, to tackle non-graph structured COPs via problem embedding innovations; 2) develop hierarchical (e.g., through D&C schemes) meta-learning strategies to tackle large-scale instances and compare different neural backbones (e.g., novel graph Transformers, etc.) for potential performance gain in the task-specific solving scenarios; 3) incorporate more neural approaches (Ma et al., 2025a; Ahn et al., 2020; Sanokowski et al., 2023) to report comparable performance on our novel benchmark datasets.

**Remarks.** While we regard the work presented thus far as substantial, we offer the above outlook for further insights aligned with a shared aspiration of the vibrant neural combinatorial optimization (NCO) community: developing a universal neural solver capable of addressing the diverse, complex, and real-world landscape of discrete optimization tasks, one that leverages data-driven paradigms to deliver enhanced efficiency.

