# OpenReview forum: "Problem Distributions as Tasks: Repurposing Meta Learning for Generative Combinatorial Optimization towards Multi-task Pretraining and Adaptation"
_ICML.cc/2026/Conference — ICML 2026 regular_

### Official Review · Reviewer_ZoYS · 2026-03-08

**Soundness:** 3
**Presentation:** 3
**Significance:** 4
**Originality:** 2
**Overall Recommendation:** 4
**Confidence:** 3

**Summary:**

This paper proposes a multi-task meta-learning framework named M²GenCO, which innovatively defines different combinatorial optimization problems and their data distributions as “tasks” in meta-learning. The framework integrates an advanced discrete diffusion-based generative model and performs multi-task pretraining under a unified graph neural network architecture. Through fine-tuning with only a small number of samples, it achieves fast and efficient adaptation to OOD graph instances. In addition, the authors construct a comprehensive benchmark consisting of five classical graph-based COPs and 38 datasets with diverse distributions, aiming to standardize the evaluation of generalization ability for NCO solvers. Extensive experiments demonstrate that M²GenCO significantly outperforms existing generative and meta-learning baselines in terms of inference speed, training efficiency, and solution quality.

**Compliance With Llm Reviewing Policy:**

Affirmed.

**Key Questions For Authors:**

Q1: The abstract states that “M²GenCO with greedy decoder yields …”. What exactly does the “greedy decoder” refer to in this context? Could you provide a more detailed explanation of its mechanism and implementation?

Q2: In the introduction, the statement “treating different instances of the same problem class (rather than distinct problem types) as meta-tasks, which departs from the original multi-task principle as in Finn et al. (2017)” is relatively brief. Could you elaborate on this point in greater detail? Additionally, please analyze the effectiveness of the proposed cross-problem shared structure compared to task-specific optimization approaches.

Q3: Lack of Theoretical Analysis in Experiments: In the Main Results and Analyses section, the discussion primarily focuses on quantitative metrics, without deeply examining the underlying reasons for the observed performance gains. Could you provide a more detailed analysis of the sources of the performance improvements?

**Limitations:**

Yes

**Strengths And Weaknesses:**

Strengths:

S1: New Perspective and Framework: The paper breaks the conventional limitation of meta-learning in NCO, which has typically focused on adaptation across instances of a single problem. It combines multi-task meta-learning with a powerful diffusion-based generative model, effectively enhancing generalization across different COP types and data distributions.

S2: Valuable Benchmark Contribution: The authors construct a comprehensive multi-distribution benchmark comprising 38 datasets, addressing the long-standing lack of standardized OOD generalization benchmarks in the NCO field. This contribution is highly valuable and is expected to significantly promote future research within the community.

S3: Extensive Experiments and Strong Empirical Performance: The main paper and appendix present remarkably comprehensive experimental results. While achieving high-quality solutions (SOTA-level optimality gaps), the proposed method substantially alleviates the slow inference and high training cost issues commonly associated with traditional diffusion models in combinatorial optimization.

Weaknesses:

W1: Limited Architectural Novelty: Although the overall framework is highly novel at a macro level, the methodological contribution primarily lies in the integration and incremental extension of existing techniques rather than fundamental algorithmic innovation. The degree of theoretical and micro-level novelty is relatively limited.

W2: Scalability Bottleneck for Ultra-Large Graphs: While the proposed method improves upon prior diffusion-based models, it still relies on dense distance matrix inputs (e.g., for TSP/ATSP) to maintain representational consistency. For real-world scenarios involving ultra-large graphs with tens of thousands of nodes (>10,000), this design will inevitably encounter GPU memory bottlenecks. What specific strategies do you propose for sparsifying or restructuring the architecture to address this limitation? Clarifying this issue would help define the practical industrial applicability boundary of the method.

W3: Relatively Concise Methodological Exposition: Although additional methodological details are provided in the appendix, key components of the approach would benefit from more thorough and explicit explanation in the main text to improve clarity and completeness.

---

> ### Author Rebuttal · Authors · 2026-03-31
>
> Dear Reviewer ZoYS,
>
> Thanks for your valuable recognition and helpful review!
>
> ---
> >**W1: Micro-level novelty.**
>
> **A:** Our novelty is not in inventing wholly new neural components, but in meaningfully and practically broadening the meta-learning paradigm in CO, directly motivated by limitations of prior formulations[17,21]. Notably, instantiating, coordinating, and making such a design in NCO is itself technically nontrivial. The gains in performance and efficiency suggest the contribution is not merely conceptual, but also a solid methodological and empirical advance.
>
> ---
> > **W2: Representation for Ultra-large graph.**
>
> **A:**
> * We use dense matrices for TSP/ATSP to follow precedent, ease reproduction, and remain effective at the scales currently tractable for neural solvers. Table 37 also shows our lightweight backbone is markedly more memory-efficient than baselines.
> * For larger-scale extension, we see two practical directions: **graph sparsification** (e.g., k-NN), as in [2,6,21-24,26,27], and orthogonal integration with **divide-and-conquer** methods[3,5,14], where our model serves as an efficient local solver. Neither requires major architectural refactoring.
> * Yet, fully validating these directions still needs industrial-scale compute, and ultra-large instances remain beyond the practical scope of this paper (and most current NCO works). We thus present them cautiously as feasible future directions.
>
> ---
> > **W3: Detailed method description.**
>
> **A:** Algorithm 1 already gives the full pipeline in detail. We'll further refine the paper and fully open-source the code & data upon acceptance.
>
> ---
> > **Q1: Greedy decoder.**
>
> **A:** The greedy decoder is a simple feasibility-enforcing procedure. Given a predicted heatmap $\mathcal{H}\in[0,1]^N$ (with $N$ decision variables), it sorts variables by score and iteratively adds the highest-scoring valid candidate until a feasible solution is completed. This follows common practice in prior NCO works [1-4,6,13-19,21-27] and avoids confounding effects from more complex decoders. In implementation, we use the greedy-decoding interface in open-source package ML4CO-Kit for reproducibility.
>
> ---
> > **Q2: Ours vs. prior meta NCO vs. task-specific NCO.**
>
> **A:**
> 1. Compared with prior single-task meta-learners[17,21], the key difference is **how the task distribution is defined** in the objective: $$\min_{\theta} \mathbb{E}\_{G \sim p(\mathcal{T})}[ \mathcal{L}(U( \theta;G_{\text{sup}}),G_{ \text{qry}})],$$
>    where $\theta$ is the meta-initialization, $\mathcal{T}$ is a **single fixed CO problem under a fixed distribution**, and the variation comes mainly from different **instances**. In contrast, our objective expands the task family:
>    $$\min_{\theta} \mathbb{E}\_{(i,j,s) \sim p(\mathcal{T})} \mathbb{E}\_{G \sim p(\mathcal{T}\_{i,j,s})}[\mathcal{L}(U(\theta;G_{\text{sup}}),G_{\text{qry}})],$$
>    where $\mathcal{T}_{i,j,s}$ denotes the meta-task induced by **problem type** $i$, **distribution** $j$, and **scale** $s$. Thus, our outer objective is optimized over multiple COPs and shifted distributions, rather than one fixed task family.
>
> 2. While task-specific models learn a dedicated $\theta_i$ for one task, our goal is to learn an **adaptation-friendly shared initializer** $\theta_{\text{meta}}$. A cross-problem shared structure is therefore natural: it exposes the model to more task-agnostic information, enables coordinated optimization across multiple task distributions, and improves parameter sharing, supervision reuse across related COPs, and few-shot adaptation under task shift.
>
> ---
> > **Q3: Analysis for improvement sources.**
>
> **A:** In fact, we provide targeted ablations showing the gains come from **3 coordinated sources**:
>
> 1. **Cross-task meta-learning** improves initialization. Meta-pretraining over expanded task families learns a more adaptation-friendly $\theta_{\text{meta}}$ across COPs/distributions. Removing it (“w/o meta”) consistently hurts performance (Table 19), esp. on OOD data. Strong transfer to unseen MVC (<2% gaps) further suggests meta-learning mainly provides a better adaptation-oriented shared prior.
>
> 2. **Generative diffusion modeling** improves expressivity. Our model learns the solution distribution via the diffusion-consistency [24] objective (Eq. 4-5), rather than prior weaker point-wise SL [2-4]. Replacing the generative backbone with a vanilla GNN (“w/o model”) degrades performance on all tasks, showing the gain comes not only from cross-task training, but also from better modeling of the solution manifold.
>
> 3. **Few-shot finetuning** turns the shared prior into target-distribution performance. Starting from $\theta_{\text{meta}}$, we adapt with a small support set to obtain $\theta_{\mathcal{T}_{i,j}}$ for each target distribution. Table 20 and its analysis show the lightweight adaptation consistently improves test performance.
>
> ---
> **References: https://anonymous.4open.science/r/paper-12521-1C3E/assets/rebuttal-table-2.png.**

---

> > ### Author Rebuttal · Reviewer_ZoYS · 2026-04-04
> >
> > Thank the authors for the rebuttal. I'll remain the positive score.

---

> > > ### Author Response · Authors · 2026-04-04
> > >
> > > Thanks for your acknowledgement and for maintaining your positive recommendation. Glad to learn our clarifications have fully resolved your concerns and questions. We'll continue refining the paper so that the final version can best reflect the additions and clarifications made during the rebuttal phase.
> > >
> > > Sincerely,
> > >
> > > Authors of Submission 12521

---

### Official Review · Reviewer_Vept · 2026-03-11

**Soundness:** 3
**Presentation:** 3
**Significance:** 3
**Originality:** 2
**Overall Recommendation:** 4
**Confidence:** 4

**Summary:**

This paper proposes a multi-task meta-learning framework M2GenCO for neural combinatorial optimization (NCO) across multiple graph-based combinatorial optimization problems. The key idea is to define meta-tasks at the level of problem types and distributions rather than instances, and to combine meta-learning with a diffusion-based generative solver. The paper also introduces a benchmark covering five classic combinatorial optimization problems under multiple distributions. Overall, the work addresses an important challenge in NCO-generalization across heterogeneous tasks and distributions, and presents a reasonably coherent framework integrating meta-learning, generative modeling, and graph neural networks. The experimental results suggest promising performance improvements in both solution quality and inference efficiency.

**Compliance With Llm Reviewing Policy:**

Affirmed.

**Key Questions For Authors:**

Q1: The framework models combinatorial solutions as Bernoulli variables and optimizes them using binary cross-entropy. How does the model ensure feasibility of solutions for problems with strong structural constraints (e.g., TSP tours)? Is any decoding or constraint enforcement applied after prediction? Additionally, are the inputs to all problems of node-oriented or edge-oriented problems represented in a unified format, or does each problem type require a separate input representation?

Q2: The paper emphasizes meta-learning for fast adaptation, yet the final adaptation procedure is performed via offline few-shot finetuning. Could the authors clarify how this differs from standard transfer learning, and whether test-time meta-adaptation was considered?

Q3: The architecture shares a common GNN encoder across different problem types, including both node-oriented and edge-oriented tasks. How does the model reconcile the differences in variable representations and graph structures across these tasks?

Q4: During finetuning, some tasks use supervised BCE loss while MCut uses an unsupervised energy-based objective. What is the rationale behind using different training objectives across tasks, and how sensitive is the performance to this choice?

Q5: In Table 2, the reported results for LKH3 on the ATSP instances are extremely small, with some values even equal to zero. Could the authors clarify whether these results are correct, and if so, explain why LKH3 achieves such unusually low values on ATSP compared to other solvers?

**Limitations:**

Yes

**Strengths And Weaknesses:**

Strengths: The paper tackles a meaningful and underexplored problem in neural combinatorial optimization, namely cross-problem generalization. The idea of defining meta-learning tasks at the level of problem types and graph distributions is interesting and potentially useful for building more general NCO solvers. The integration of diffusion-based generative modeling with meta-learning is also technically appealing and aligns with recent trends in generative optimization. Another notable contribution is the construction of a multi-task benchmark covering five classical graph combinatorial optimization problems with multiple distributions and scales, which could facilitate more standardized evaluation of generalization in NCO. In addition, the framework is designed with practical considerations, such as reducing inference cost via consistency models and enabling efficient adaptation through few-shot finetuning.

Weaknesses: Despite the interesting direction, the methodological novelty is somewhat limited because the main components, including meta-learning, diffusion-based generative modeling, and GNN-based encoders, are largely adopted from existing works and combined in a relatively straightforward way. The role of meta-learning in the final framework is also somewhat weakened, as adaptation is performed via offline finetuning rather than true test-time meta-adaptation, making the approach closer to transfer learning than classical meta-learning. In addition, several design choices are insufficiently justified, such as the use of independent Bernoulli modeling for structured combinatorial solutions, the difference between supervised and unsupervised finetuning objectives across tasks, and the use of shared encoders for both node-oriented and edge-oriented problems. Finally, although the paper evaluates multiple combinatorial optimization tasks, the choice of non-learning baselines appears somewhat inconsistent across problems. In particular, classical exact solvers and traditional heuristic methods are not always aligned across the evaluated tasks.

---

> ### Author Rebuttal · Authors · 2026-03-31
>
> Dear Reviewer Vept,
>
> Thanks for your review and the recognition of our work! Below are our detailed response to your concerns and questions.
>
> ---
> > **W1: Methodological novelty.**
>
> **A:** Please refer to our responses to Reviewer rqQV’s **W6** & ZoYS's **W1**.
>
> ---
> > **W2/Q2: Meta vs transfer learning; test-time adaptation.**
>
> **A:** For the former, Appendix H.2 already discusses the conceptual relation and nuances between our method and conventional transfer learning/multi-task learning. For the latter,
>
> * We agree that our setting is not classical online test-time meta-adaptation. Rather, it is a practically motivated meta-learning-inspired adaptation framework, where the learned initialization is explicitly optimized for efficient few-shot adaptation. This is relevant when the target distribution shifts but remains stable for some period, so that occasional offline adaptation can improve many subsequent instances.
> * Table 10 shows that performance is not strongly affected by the number of inner steps, whereas learning time increases much more with larger $t_{in}$. This is also consistent with prior studies, e.g. in DIMES[21], increasing 10 test-time meta-updates enhances the objective by less than 3%, but raises TSP-500 solving time from 1.06m to 2.11h.
>
> Thus, while we agree that online adaptation is an important future direction, in current NCO practice it is often more costly than effective.
>
> ---
> > **W3-Q1: Bernoulli modeling? feasibility decoding? unified input representation?**
>
> **A:**
> 1. Modeling graph-based CO solutions as Bernoulli variables and optimizing them with binary cross-entropy is a well-established convention[2-4,6,18,22-24,26-27], which we follow for fair comparison. Note graph-CO solutions can be naturally represented as binary selections over nodes or edges, so BCE provides a simple and effective supervised objective without requiring the raw output to be immediately feasible.
>
> 2. Yes, feasibility is enforced by greedy decoders. Please see details in our answer to Reviewer ZoYS's **Q1**.
>
> 3. Yes, all instances are stored in unified `.txt` format, and we use the open-source ML4CO-Kit package for standardized I/O. During training and inference, instances from different tasks are consistently represented as PyTorch tensors with shared fields (node features, edge features, and connectivity), enabling unified batched computation for GCN layers across diverse tasks.
>
> ---
> > **W3-Q3: How to reconcile across tasks?**
>
> **A:** As clarified above, structural heterogeneity among graphs is handled through a unified representation space. During joint pre-training, the encoder is learned mainly in an objective-driven manner via a unified BCE loss and is largely agnostic to task-specific semantics beyond the shared graph representation. From the model’s perspective, it receives diverse graph instances in a common format and learns a transferable shared initialization over their solution patterns. Therefore, we do not introduce hand-crafted mechanisms to explicitly reconcile structural diversity across task types; instead, this is learned data-driven as part of the shared meta-initialization $\theta_{meta}$.
>
> ---
> > **W3-Q4: Choice of fine-tuning objectives.**
>
> **A:** This design choice follows prior findings: DiffUCO[19] introduced unsupervised energy-based objectives for diffusion learning on node tasks, and COExpander[26] found them more effective for MCut than for other tasks. A plausible reason is that MCut is unconstrained: its objective can be directly written as $\mathcal{L}\propto\sum_{(i,j)\in\mathcal{E}}(2\mathbf{x}_i-1)(2\mathbf{x}_j-1),$ without additional penalty terms for feasibility. This avoids conflicting gradients for more constrained problems, better fitting energy-based exploration. Below is a minimal test on HK-small data, aligning these insights.
>
> |Loss|MIS↑|MCut↑|
> |-|-|-|
> |BCE|**77.40**|1505.24|
> |Energy|77.03|**1510.60**|
>
> ---
> > **W4/Q5: Inconsistent oracle solvers? LKH's low obj. on ATSP?**
>
> **A:**
> 1. Our oracle setup is not inconsistent, but chosen to match solver expertise and established benchmarking practice: KaMIS is specialized for MIS and often stronger than generic Gurobi; Concorde is specialized for 2D symmetric TSP with coordinates and is not applicable to general ATSP; while LKH is a strong heuristic for both TSP and general ATSPs.
>
> 2. The LKH results on ATSP are correct. As detailed in Appendix D.2, two challenging ATSP distributions in our benchmark are reduced from HCP and 3-SAT (inspired by [15]). In these constructions, the optimal ATSP tour length is exactly 0, indicating guaranteed satisfiability for SAT or the existence of a Hamiltonian cycle for HCP. Hence, near-zero objectives are expected rather than any anomaly in LKH, and such gaps also show our benchmarks remain challenging for neural solvers.
>
> ---
> **Note: A shared reference table is given via this anonymous link: https://anonymous.4open.science/r/paper-12521-1C3E/assets/rebuttal-table-2.png**

---

> > ### Author Rebuttal · Reviewer_Vept · 2026-04-03
> >
> > Thank you for the rebuttal. The authors have addressed my earlier concerns. I still find the methodological novelty somewhat limited, as the contribution appears closer to a practical extension of prior meta-learning formulations.
> >
> > My score remains unchanged.

---

> > > ### Author Response · Authors · 2026-04-03
> > >
> > > We are more than grateful to learn that our detailed responses have **fully resolved** your earlier concerns. Thank you for **maintaining your positive recommendation with solid confidence**.
> > >
> > > As this is the **last chance** we could communicate with you directly, which we sincerely cherish, we would like to offer a brief final clarification regarding the minor novelty and positioning issue you mentioned.
> > >
> > > ---
> > > ## **A. What motivates us and why it is fundamentally nontrivial relative to prior works**
> > >
> > > We emphasize that our claims are intended to address a concrete gap that, in our view, remains insufficiently resolved in prior work. As stated in the introduction, the three major research lines in NCO exhibit ***complementary strengths but orthogonal limitations***:
> > >
> > > * **1) Task-specific generative solvers.**
> > >   Diffusion-based solvers achieve state-of-the-art performance on *single* COPs, especially in-distribution, but typically require substantial computational budgets and show limited transferability beyond their designated tasks (e.g., Fast-T2T, DiffUCO, DIFUSCO, VAG-CO).
> > >
> > > * **2) Cross-task / multi-task frameworks.**
> > >   These methods pursue broader generality via unified representations or MoE architectures, but often introduce considerable task-reformulation overhead (e.g., UniCO), remain restricted to narrower COP families (e.g., MVMoE, MTNCO, UnifyML4TSP), or rely on sequential decoding architectures (e.g., UniteFormer, GOAL) that limit scalability and compatibility with more expressive backbones.
> > >
> > > * **3) Meta-learning approaches.**
> > >   Existing meta-learners rely mainly on RL/UL paradigms, and more importantly, formulate meta-tasks **only at the instance level within a single COP type** (e.g., DIMES, Meta-EGN), rather than across distinct problem types in the broader spirit of meta-learning.
> > >
> > > In this sense, our work is not meant to “repackage” prior ideas, but to take a concrete step toward a setting that, to our knowledge, prior methods have not jointly realized: **combining cross-problem adaptation, expressive generative modeling, and practical efficiency within one unified framework**.
> > >
> > > ---
> > > ## **B. What we contribute**
> > >
> > > * **1)** we **define meta-learning tasks across distinct COP types**, enabling task-level adaptation and cross-distribution pretraining, rather than only instance-level optimization within a fixed problem family;
> > >
> > > * **2)** we **integrate a supervised diffusion-based generative backbone** into a unified multi-task meta-solver, leveraging both the expressivity of diffusion models and the adaptability of meta-learning to achieve robust cross-distribution results with largely reduced training cost;
> > >
> > > * **3)** we establish a **multi-distribution benchmark** covering **5 COPs** across **38 datasets** with diverse distributions and scales, enabling more systematic evaluation of generalizability and adaptability;
> > >
> > > * **4)** we show **strong empirical performance**, including a **9.16% mean improvement** over prior best learning-based results with fast adaptation to unseen scales and distributions;
> > >
> > > * **5)** we demonstrate significant **efficiency gains**, including **95.6×** faster inference on average, up to **82%** memory reduction, and about **91%** training-time reduction compared to single-task full-parameter diffusion-based solvers.
> > >
> > > ---
> > > ## **C. What we have done during rebuttal**
> > >
> > > * we further clarified that training separate models for node/edge-oriented tasks is a **standard and principled design choice**, rather than an arbitrary weakening of the multi-task claim;
> > >
> > > * we explained carefully the conceptual distinction between our setting and conventional transfer learning/multi-task learning, and positioned our method modestly and precisely as a **multi-task, cross-distribution, meta-learning-inspired** framework;
> > >
> > > * we provided more **aggregate evidence**, e.g. **Mean Rank** metrics for both w/ and w/o finetuning, to summarize the competitiveness of our method across tasks and datasets;
> > >
> > > * we further clarified the **practicality of test-time finetuning**, emphasizing that it is a lightweight few-shot adaptation step with modest overhead, rather than expensive retraining;
> > >
> > > * we gave more explicit justification for several design choices, e.g. the rationale for the **single-step inner update**, whose quality–efficiency trade-off is supported both by our ablations and by prior meta-NCO studies.
> > >
> > > In short, we genuinely tried **not only to defend the method, but also to improve the transparency of its scope, assumptions, and design choices**.
> > >
> > > ---
> > > Finally, while already fully grateful for your positive assessment, we do hope that this final remark may **further alleviate your minor concern regarding the positioning of this work**, and help all reviewers and the chairs **better understand our motivation, concrete innovations, empirical contributions, and rebuttal efforts**.
> > >
> > > Thank you again for your time, encouragement, and thoughtful consideration.
> > >
> > > Best regards,
> > >
> > > Authors of Submission 12521

---

### Official Review · Reviewer_EGFY · 2026-03-11

**Soundness:** 3
**Presentation:** 3
**Significance:** 3
**Originality:** 2
**Overall Recommendation:** 4
**Confidence:** 2

**Summary:**

The paper proposes M²GenCO, a multi-task meta-learning framework for neural combinatorial optimization on graphs that combines a discrete diffusion-style generative solver with a MAML-like bi-level training procedure. Meta-tasks are defined at the level of problem type and graph distribution, and a lightweight GCN-based backbone is meta-pretrained across several COPs, then finetuned with few-shot supervision to new distributions. The authors also assemble a benchmark of 5 classic graph COPs with multiple distributions and scales, and empirically show strong performance and speed improvements over prior neural baselines, including task-specific diffusion models, meta-learners, and multi-task frameworks.

**Compliance With Llm Reviewing Policy:**

Affirmed.

**Final Justification:**

The authors have fully addressed all my concerns through the rebuttal and additional experiments. I think this paper is technically sound and well-positioned within the literature. I maintain my positive rating and strongly recommend the paper for acceptance.

**Key Questions For Authors:**

Q1. As aforementioned, could you provide the Mean Rank metric (either overall or per-task) for the results in Tables 2 and 3?

Q2. Table 4 shows a significant performance margin between results with and without fine-tuning. Could you also share the Mean Rank metric for cases where fine-tuning was not applied?

Q3. Since I am not an expert in this specific domain, I am curious about the practicality of performing fine-tuning at test-time. To what extent does this process impact real-world deployment or operational efficiency?

Minor question: In Table 4, the value for MCI-HK w/ f-t under Node-200-300 (Obj.) appears to differ from the corresponding value in Table 2. Could you clarify the reason for this discrepancy?

**Limitations:**

yes

**Strengths And Weaknesses:**

>1. Soundness
>
>The technical components—MAML-style meta-learning, discrete diffusion with consistency-style supervision, and the GCN-based architecture—are standard but smartly combined. The experimental methodology is solid overall, featuring strong baselines and relevant ablations. However, some details regarding the results are missing from the main text and require clarification.

>2. Presentation
>
>The paper is very polished and exhibits excellent readability. The limitations of prior arts, the reasoning behind the proposed methodology, the distinctions from existing work, and the contributions are all clearly stated. However, certain details in the Experiments section are insufficiently described, and there are minor issues with the presentation of results. Specifically, Tables 2 and 3 do not explicitly state whether fine-tuning was performed. Additionally, bolding only the "Ours" results in the tables somewhat hinders the intuitive comparison of performance. Given the variety of metrics, adding aggregate indicators such as Mean Rank or relative improvement/decrement compared to per-task SOTA would enhance the overall interpretability. (Note: As I am not deeply specialized in this specific field, the mathematical consistency has not been exhaustively verified.)

>3. Significance & Originality
>
>The combination of multi-task meta-learning across different COP types with a diffusion-style generative solver, along with a broad benchmark and strong empirical gains in both quality and speed, represents a meaningful contribution to the NCO community. While the conceptual novelty is moderate—largely integrating existing ideas—the execution and empirical validation are substantial.

---

> ### Author Rebuttal · Authors · 2026-03-31
>
> Dear Reviewer EGFY,
>
> Thank you for your dedicated review, especially for recognizing the overall contribution of our work and for the positive recommendation. Below, we provide detailed responses to your questions and concerns.
>
> ---
> ### **1. Responses to the Questions Raised**
>
> > **Q1/Q2: Mean-rank metrics.**
>
> **A:** Based on the **full results (Tables 17/18)**, we additionally report **Mean Rank (MR)** using a normalized ranking scheme to account for different numbers of comparable baselines across datasets. For each dataset, all **reported learning-based methods** are ranked by the corresponding metric, using higher-is-better for MIS/MCI/MCut objectives and lower-is-better for TSP/ATSP objectives and inference time. The rank is normalized as $(\mathrm{rank}-1)/(n-1)\times100\\%\in[0\\%,100\\%]$, where $n$ is the number of comparable methods, so lower is better. Ties use average rank, and missing results are excluded. We then average normalized ranks over datasets to obtain **per-task MR**, and over all comparable datasets to obtain **overall MR**. Each entry is reported as **[MR(obj) w/ ft, MR(obj) w/o ft, MR(time)]**, all in percentage. We **bold** the best MR and *italicize* the second-best in each column.
>
> |Method|TSP|ATSP|Overall|
> |-|-|-|-|
> |GNNGLS|83.33/83.33/30.56|-|83.33/83.33/30.56|
> |DIMES|70.00/70.00/*18.33*|*45.83*/*41.67*/*25.00*|59.64/55.83/21.19|
> |UTSP|91.67/91.67/**8.33**|-|91.67/91.67/**8.33**|
> |SymNCO|*18.33*/**15.00**/64.44|-|*18.33*/*15.00*/64.44|
> |DIFUSCO|65.00/65.00/86.67|-|65.00/65.00/86.67|
> |GOAL|18.75/18.75/93.75|53.12/43.75/100.00|35.94/31.25/96.88|
> |UniCO|-|83.33/83.33/75.00|83.33/83.33/75.00|
> |MatNet|-|79.17/79.17/50.00|79.17/79.17/50.00|
> |Ours|**13.75**/*16.25*/36.25|**3.12**/**3.12**/**0.00**|**8.44**/**9.69**/*18.12*|
>
> |Method|MIS|MCl|MCut|Overall|
> |-|-|-|-|-|
> |GCN4CO|46.43/48.21/33.93|60.71/60.71/45.54|85.37/83.67/28.57|64.02/64.04/36.44|
> |VAG-CO|80.36/82.14/**6.25**|83.93/83.93/*14.29*|97.62/97.96/**10.20**|86.65/87.82/**10.43**|
> |Meta-EGN|92.86/92.86/*16.07*|66.07/64.29/**8.04**|75.51/73.47/34.69|77.60/76.30/*19.07*|
> |Meta-EGN†|78.57/78.57/50.00|*17.86*/*16.07*/76.79|54.42/53.06/63.27|48.81/47.73/63.96|
> |DIFUSCO|50.00/48.21/87.50|62.50/66.07/100.00|47.96/48.98/100.00|53.90/54.95/96.02|
> |Fast-T2T|*21.43*/*21.43*/67.86|78.57/78.57/47.32|**10.88**/*12.24*/69.39|38.85/39.29/60.88|
> |DiffUCO|41.07/39.29/69.64|-|20.41/20.41/81.63|30.74/29.85/75.64|
> |GOAL|35.71/35.71/100.00|*17.86*/17.86/80.36|-|*26.19*/*26.19*/89.52|
> |Ours|**3.57**/**3.57**/18.75|**12.50**/**12.50**/27.68|*14.63*/**10.20**/*12.24*|**10.34**/**8.93**/19.93|
>
> ---
> > **Q3: Practicality of test-time fine-tuning.**
>
> **A:** In our setting, test-time fine-tuning is a **lightweight few-shot adaptation** step rather than expensive retraining: it uses only a small support set and very few update steps, so the added cost is modest. Empirically, a small adaptation budget already recovers most of the gain while preserving our framework’s memory and inference efficiency.
>
> From an application perspective, this is most relevant when the target distribution shifts but remains stable for some period, so that a short one-time or occasional adaptation can improve many subsequent instances. By contrast, in fully real-time or highly latency-sensitive scenarios, directly using the pretrained model may be preferable, and we do not claim that test-time fine-tuning is always necessary. Our point is only that it is a practical and beneficial option in cross-distribution CO settings when a small adaptation overhead is acceptable.
>
> That said, we agree that more robust online adaptive frameworks are also important and remain a promising future direction for the NCO community.
>
> ---
> > **Q4: Minor value inconsistency.**
>
> **A:** This is a typographical error, and we will correct it in the revision. The value for **MCI-HK w/ f-t** under **Node-200-300 (Obj.)** should be **6.42** (more precisely, **6.418**, as reported in Table 17), while **6.38** is the result for Node-800-1200 data.
>
> ---
> ### **2. Clarifications on Presentation-related Issues**
>
> > **I1: Fine-tuning status.**
>
> **A:** Yes, fine-tuning is consistently conducted for both our method and the compared neural baselines, on our unified few-shot support sets. We will update the main text to make it clearer.
>
> ---
> > **I2: Result bolding.**
>
> **A:** We would like to clarify that we are not bolding only `Ours`. As stated in the captions of Tables 2/3/17/18, we bold the **competitive learning-based methods relative to ours in both solution quality and solving time**. Our intention is for the bolded entries to reflect practically competitive (Pareto-optimal) results under the accuracy-efficiency trade-off, thus facilitating future model selection.
>
> ---
> > **I3: Aggregate metrics.**
>
> **A:** The **Mean Rank** metric is provided above. In addition, **Table 16 (Appendix G.1, p. 25)** gives a detailed leaderboard-style comparison summarizing relative gains over prior SOTA across benchmarks.

---

> > ### Author Rebuttal · Reviewer_EGFY · 2026-04-03
> >
> > See final justification.

---

> > > ### Author Response · Authors · 2026-04-03
> > >
> > > Thank you very much for your acknowledgement! We are truly pleased to know that our clarifications during the rebuttal have adequately resolved your concerns. We also sincerely appreciate your maintaining a positive recommendation of our work (although as authors we are not able to view your final justification text for now).
> > >
> > > Best regards,
> > >
> > > Authors of Submission 12521

---

### Official Review · Reviewer_rqQV · 2026-03-13

**Soundness:** 2
**Presentation:** 3
**Significance:** 3
**Originality:** 3
**Overall Recommendation:** 4
**Confidence:** 4

**Summary:**

This paper presents a multi-task meta-learning framework for graph-based combinatorial optimization. The approach integrates diffusion models with graph neural networks, pre-training across different problem types before fine-tuning on specific distributions. The authors also contribute a multi-distribution benchmark dataset and evaluate the framework against various solvers.

**Compliance With Llm Reviewing Policy:**

Affirmed.

**Final Justification:**

My concerns have been adequately addressed

**Key Questions For Authors:**

See weaknesses.

**Limitations:**

Yes

**Strengths And Weaknesses:**

Strengths

1. The authors connect generative modeling with a meta-learning paradigm mitigates some of the high training costs associated with standard diffusion models

2. The constructed benchmark dataset includes various graph structures and scales, facilitating the evaluation of out-of-distribution capabilities

3. Experiments cover both classical and neural solvers, demonstrating efficiency improvements on specific metrics

Weaknesses

1. The framework trains separate meta-models for edge-selection and node-selection tasks. It is not a unified cross-problem meta-learner

2. The reliance on a single-step inner loop update due to performance degradation with multiple steps indicates potential instability in the meta-optimization landscape. The model might be converging to a generic smoothed prior rather than developing true fast-adaptation dynamics

3. Many compared baselines were strictly designed for in-distribution solving. Comparing a distribution-fine-tuned model against these zero-shot baselines on shifted data does not constitute a strictly balanced evaluation

4. Evaluations for node-oriented tasks are restricted to relatively small graphs. The scalability claims lack empirical backing

5. The efficiency claims are less compelling when benchmarked against fast constructive neural methods or highly optimized heuristic operations

6. In the field of ML, the practice of conducting joint training on different types of tasks to obtain a good initialization is essentially the classic multi-task pre-training. The paper applies the two-level optimization framework of MAML. Referring to it as "defining a new task for meta-learning" is more of a conceptual repackaging rather than an innovation

---

> ### Author Rebuttal · Authors · 2026-03-31
>
> Dear Reviewer rqQV,
>
> Thank you for your dedicated review!
>
> ---
> > **W1: Separate training for edge/node tasks.**
>
> **A:** We sincerely clarify that **we do not intend to overclaim a fully "unified cross-problem meta-learner".**
>
> * In the submission, we have positioned our method cautiously as a modest but concrete step toward multi-task training, especially multi-distribution adaptation, rather than a universally cross-problem learner, which remains beyond the current scope of the broader NCO field.
> * Training separate models for node/edge-selection tasks is not arbitrary, but an established practice[1-34], because they involve fundamentally different action spaces and represent distinct task families. We also note that prior works covering even narrower task spectra, or conducting more isolated model training, have claimed to be “unified,” “universal,” or “generic” [8-12,15] without being regarded as conceptually inappropriate.
> * Even within either node/edge tasks, there are already multiple problem types and diverse distributions, which broaden the setting emphasized in recent NCO research.
> * **Summarized related works in this regard: https://anonymous.4open.science/r/paper-12521-1C3E**
>
> ---
> > **W2: Model stability with the single-step inner update.**
>
> **A:** First, additional inner steps tend to overfit limited support data and hurt both generalization and efficiency for training and inference. Using a single inner step should not be interpreted as instability in optimization, but rather grounded in **precedented, empirical, and practical considerations**.
>
> * The gains over baselines suggest our learned initialization, even with a single step, is sufficient for effective adaptation under sparse supervision, meanwhile, selecting the best-performing setting and reporting corresponding ablations (as we did) is common practice in ML.
> * Table 10 shows that performance is not quite sensitive to the number of inner steps, whereas learning time increases much more with larger $t_{in}$. This is largely consistent with evidence in former meta-NCO works. E.g., in DIMES[21] (Table 2c, p. 7), increasing inner steps from 0 to 14 changes the obj. by less than 3%, but raises TSP-500 solving time from 1.06m to 2.11h. Likewise, Meta-EGN[17] also deliberately adopts a single-step inner update and provides both empirical and theoretical support (p. 5 and Appendix A). Similar trade-offs are also reported in [6].
>
> ---
> > **W3: Evaluation fairness (finetune).**
>
> **A:** Compared baselines were also finetuned to ensure fair comparison. Our response to Reviewer EGFY's Q1/Q2 further reports Mean Rank metrics for both w/ and w/o finetuning.
>
> ---
> > **W4: Results on node tasks beyond small graphs.**
>
> **A:** Thank you for noting our oversight not to explicitly highlight the larger-case results in the main text. Please refer to Table 17 for these results (800-1200 nodes, multi-distributions). The scale choices follow prior works[16-27], and our model remains stable and consistently competitive.
>
> ---
> > **W5: Efficiency vs. constructive/heuristic methods.**
>
> **A:** To clarify, as shown in Tables 2/3, our solving time is already broadly competitive with (often better than) mainstream neural and exact/heuristic solvers, and Table 16 further provides a leaderboard regarding both performance and efficiency.
>
> * Meanwhile, efficiency shouldn't be viewed in isolation, since trade-offs among solution quality, efficiency, and applicability inherently exist. We agree on the importance of constructive methods, e.g., faster inference on smaller instances and stronger constraint enforcement via autoregressive decision. However, they face scalability limits (e.g. costly RL rollouts for $N>100$) and are less suited to node-oriented tasks, while one-shot generative methods offer **complementary advantages** in these respects.
> * We already discussed in Appendix J that our efficiency claim is not to overstate one paradigm over another, but to fairly characterize our practical competitiveness.
>
> ---
> > **W6: Conceptual repackaging?**
>
> **A:** First, in Appendix H.2, we've clarified the conceptual nuances of introducing a higher-level technique into the specific NCO application, and its relation to conventional multi-task/transfer learning.
>
> * To reiterate, our point is more modest: each problem-distribution-scale tuple defines a concrete adaptation episode, which naturally admits a new MAML-style perspective, rather than "repackaging as defining a new task for meta-learning".
> * Moreover, our contribution is better viewed as a natural extension of [17,21], which treat a *single CO problem under a fixed distribution* as meta-tasks in CO learning. We regard these works as important pioneers of cross-instance adaptation within a specific CO task, and our work builds on this foundation by moving to a more practical adaptation space.
>
> ---
> We hope our clarifications have adequately addressed your concerns, and we sincerely look forward to your further consideration!

---

> > ### Author Rebuttal · Reviewer_rqQV · 2026-04-03
> >
> > N/A

---

> > > ### Author Response · Authors · 2026-04-03
> > >
> > > Many thanks for your acknowledgement, and especially for your valuable decision to raise your score above the acceptance threshold. We are truly grateful for your recognition and support! We will continue refining the paper so that the final version can best reflect the additions and clarifications made during the rebuttal phase.
> > >
> > > Best regards,
> > >
> > > Authors of Submission 12521

---

### Decision · Program_Chairs · 2026-04-30

**Decision:**

Accept (regular)

**Comment:**

The author introduces M2GenCO, a multi-task meta-learning framework for neural combinatorial optimization that combines a diffusion-based generative solver with cross-problem, cross-distribution adaptation. All reviewers concluded that the studied problem is interesting and that the benchmark is a valuable contribution. Some reviewers also like the extensive empirical study.

Most reviewers criticized the paper for having limited methodological novelty, mostly by combining existing ideas, i.e., meta-learning, diffusion-style generative modeling, and GNNs. There were also concerns about whether the method should be viewed as true meta-learning or as a strong transfer-learning-style framework with lightweight adaptation, and about the extent to which the framework is genuinely unified across problem types.

However, these points were largely clarified in the rebuttal. Overall, all reviewers leaned towards accepting the paper and I follow this assessment.